# Evaluating the Evaluators: Which UDA validation methods are most effective? Can they be improved?

## Abstract

This paper compares and ranks 8 UDA validation methods. Validators estimate model accuracy, which makes them an essential component of any UDA train-test pipeline. We rank these validators to indicate which of them are most useful for the purpose of selecting optimal model checkpoints and hyperparameters. To the best of our knowledge, this large-scale benchmark study is the first of its kind in the UDA field. In addition, we propose 3 new validators that outperform existing validators. When paired with one particular UDA algorithm, one of our new validators achieves state-of-the-art performance.

## 1   Introduction

Unsupervised domain adaptation (UDA) is a machine-learning technique for training models. It allows the categorization of unlabeled target-data by a model that is trained on labeled source-data. To date, much UDA research has been focused on the algorithms that train models. This paper focuses instead on validation methods, also known as validators.

Validators generate validation scores, to estimate the accuracy of a model in categorizing unlabeled data. Validation scores are needed because, in the absence of labels, it is impossible to directly compute model accuracy in the target domain. However, model accuracy and validation scores are not always highly correlated. A poor validator can yield misleading estimates of model accuracy, resulting in the selection of sub-optimal model checkpoints and hyperparameters. To avoid this, researchers and engineers need to use validators that are the most reliable at estimating model accuracy.

Existing papers evaluate validators on checkpoint sets that are too small and homogeneous to accurately reveal which validators are most reliable. That is why this large-scale benchmark is needed. In this paper, we compare the effectiveness of 8 UDA validation methods on a large dataset of checkpoints generated by a variety of UDA algorithms and hyperparameter settings.

We consider the following questions:

1. Which validation methods generate validation scores that are highly correlated with model accuracy? In other words, which validators provide the most reliable guidance for researchers and engineers who need to select optimal model checkpoints and hyperparameters?

2. Can some existing validators be improved to yield more reliable estimates of model accuracy?

In summary, in this paper we have:

- Benchmarked and ranked 8 validators on a large dataset of 760,000 checkpoints generated by 10 UDA algorithms and 100 hyperparameter settings per algorithm.

- Significantly improved 2 existing validators.

- Developed a new validator that, when paired with an existing algorithm, achieves state-of-the-art performance.

The paper is organized as follows:

- Sections 1.1 and 1.2 provide an overview of existing validation methods, and the 3 new validation methods that we propose.

- Section 2 explains our experiment methodology.

- Section 3 shows which validators are most effective.

- The Appendix provides additional explanations, experiment details, tables of results, and an overview of UDA algorithms.

## 1.1 Existing validation methods

**Source accuracy**

This is simply the model's accuracy on the source domain:

$$\texttt{Accuracy} = \frac{1}{N} \sum_{i=1}^{N} \mathbb{1}(\arg\max(p_i) = y_i) \tag{1}$$

where $\mathbb{1}$ is the indicator function, $N$ is the size of the source dataset, $p_i$ is the $i$th prediction vector, and $y_i$ is the label for the $i$th dataset sample. The assumption here is that the source and target domains are similar enough that high source-accuracy implies high target-accuracy.

**Reverse validation (Zhong et al., 2010; Ganin et al., 2016)**

This method consists of two steps. First it trains a model via UDA on the source ($S$) and target ($T$) data, and uses this model to create pseudo-labels for $T$. Next, it trains a reverse model via UDA on $T$ and $S$, where $T$ is the pseudo-labeled target data, and $S$ is the "unlabeled" source data. The final score is the accuracy of the reverse model on $S$. One disadvantage of this approach is that it trains two models, doubling the required training time, but still producing only a single usable model. Furthermore, all it does is make it easier to choose between training runs (i.e. for tuning hyperparameters). So selecting the best forward-model checkpoint requires using another validator that can compute scores per checkpoint.

**Entropy**

This measures the "confidence" of the model:

$$\texttt{Entropy} = \frac{1}{N} \sum_{i=1}^{N} H(p_i) \tag{2}$$

$$H(p_i) = -\sum_{j=1}^{C} p_{ij} \log p_{ij} \tag{3}$$

where $H(p_i)$ is the entropy of the $i$th prediction vector, $C$ is the number of classes, and $N$ is the size of the target dataset. An accurate model will output prediction vectors that have a single large value corresponding with the correct class for each sample. This produces a low entropy score, indicating high confidence. However, this method fails if a model is incorrectly confident. For example, the model might incorrectly classify all samples in the dataset as belonging to the same class.

**Deep embedded validation (DEV) (You et al., 2019b)**

This computes a classification loss for every source validation sample, and weights each loss based on the probability that the sample belongs to the target domain. The probability comes from a domain classifier trained on source and target data.

$$\text{DEV} = \text{mean}(L) + \eta\text{mean}(W) - \eta \tag{4}$$

$$\eta = \frac{Cov(L, W)}{Var(W)} \tag{5}$$

where $L$ is the source validation losses, and $W$ is the weight of each loss. One practical issue with DEV is that its scores are unbounded. Very large values can occur if $W$ has low variance, or if $L$ and $W$ have high covariance.

**Proxy risk (Chuang et al., 2020)**

This method evaluates checkpoints using a "check" model. The check model predicts both class label and domain, and is trained on the transfer task using an algorithm such as DANN (Ganin et al., 2016), with an additional "disagreement" loss term on the target samples:

$$D = \frac{1}{B} \sum_{i=1}^{B} -||x_i - m_i||_2 \tag{6}$$

where $B$ is the batch size, $x_i$ and $m_i$ are the $i$th prediction vector of the checkpoint and the check model respectively, and $||.||_2$ is the L2 norm function. If the check model maintains a low DANN loss, but obtains outputs that differ from the checkpoint, then the checkpoint likely has low accuracy on the target domain. The disadvantage of this method is that it requires training a DANN-like model for every checkpoint, increasing total training time from $O(\text{epochs})$, to $O(\text{epochs}^2)$.

**Ensemble-based model selection (EMS) (Robbiano et al., 2021)**

This uses a linear regressor trained on 5 signals: target entropy, target diversity, silhoutte & Calinski-Harabasz scores on the target features, source accuracy, and time-consistent pseudo-labels. EMS differs from other methods because it requires a dataset of {signal, ground truth accuracy} pairs to train the regressor. These pairs have to be collected by training a model on a domain adaptation task that has labeled target data. A drawback of this method is that the regressor may overfit and not generalize to our actual UDA task.

**Soft neighborhood density (SND) (Saito et al., 2021)**

This computes the entropy of the softmaxed target similarity matrix:

$$\text{SND} = H(\text{softmax}_\tau(\widehat{X})) \tag{7}$$

$$X = F^T F \tag{8}$$

where $H$ is the entropy function, $\text{softmax}_\tau$ is the softmax function with temperature $\tau$, $X$ is the similarity matrix, $F$ is the set of L2 normalized target feature vectors, and $\widehat{X}$ is $X$ with the diagonal entries removed. A high SND score means that each feature is close to many other features, which can indicate good clustering. The caveat of SND is that it assumes the model has not mapped all target features into a single cluster. A single cluster would result in a high SND score, but low accuracy.

## 1.2 New validation methods

Here we explain our proposed validators, which include modifications of existing methods.

**Batch nuclear-norm maximization (BNM) (Cui et al., 2020a)**

BNM is a UDA algorithm which aims to generate predictions that are both diverse and confident. It approaches this via singular value decomposition:

$$\text{BNM} = ||P||_* \tag{9}$$

where $P$ is the $N \times C$ prediction matrix ($N$ is the dataset size and $C$ is the number of classes), and $||P||_*$ is the nuclear norm (the sum of the singular values) of $P$. This simple loss function is highly effective at training UDA models, which leads us to wonder if its numerical value is a proxy for target domain accuracy. We propose using BNM as a validator by applying the BNM loss function to all of the prediction vectors of the source and/or target domain. A drawback of BNM is that the computation can be expensive for large datasets with many classes, though fast approximations do exist (Cui et al., 2021).

**ClassAMI**

Robbiano et al. (2021) proposed using the silhouette score of the target features clustered with k-means (we call this "ClassSS"). However, the silhouette score rewards tightly-bound clusters that are far apart from each other, which is not strictly necessary for high accuracy. We propose replacing the silhouette score with a less stringent, but more direct approach: the Adjusted Mutual Information (AMI) between cluster labels and the predicted target labels.

$$\text{ClassAMI} = \text{AMI}(X, \text{kmeans}(F).\text{labels}) \tag{10}$$

$$X_i = \arg\max p_i \tag{11}$$

where $X$ is the predicted labels for the target data, $p_i$ is the $i$th prediction vector, and $F$ is the set of target features.

**DEV with normalization (DEVN)**

One practical concern with DEV is that $\eta$ can become very large if $W$ has low variance, or if $L$ and $W$ have high covariance. To avoid this, we propose normalizing the weights by either max normalization or standardization.

Max normalization:

```
weights /= max(weights)  # normalize between 0 and 1
weights -= mean(weights) - 1  # shift to have mean of 1
```

Standardization:

```
weights = (weights - mean(weights)) / std(weights)  # standardize
weights += 1  # shift to have mean of 1
```

### 1.3  How the new and existing validators are related

Table 1: How our proposed validators relate to existing validators and UDA algorithms.

| Existing validator or algorithm | Our proposed validator |
|---|---|
| BNM (used as an algorithm) | BNM (used as a validator) |
| ClassSS (validator) | ClassAMI |
| DEV (validator) | DEVN |

## 2 Experiment Methodology

To allow for efficient benchmarking, we created a dataset of feature vectors that could be easily loaded and used as input to all validation methods.

### 2.1 Creating the dataset of feature vectors

We ran experiments on 19 transfer tasks:

- **MNIST**: 1 task between MNIST and MNISTM (Ganin et al., 2016).

- **Office31** (Saenko et al., 2010): 6 tasks between 3 domains (Amazon, DSLR, Webcam).

- **OfficeHome** (Venkateswara et al., 2017): 12 tasks between 4 domains (Art, Clipart, Product, Real).

For the MNIST→MNISTM task, each training run used a LeNet-like model as the trunk, pretrained on MNIST. For Office31 and OfficeHome, we used a ResNet50 (He et al., 2016) pretrained (Wightman, 2019) on ImageNet (Russakovsky et al., 2015), and finetuned this model on every domain. Then for every task, we started each training run using the model finetuned on the source domain (i.e. the source-only model). We followed this procedure using 10 UDA algorithms (see Table 2), all implemented in PyTorch (Paszke et al., 2019).

For each UDA algorithm/task pair, we ran 100 steps of random hyperparameter search using Optuna (Akiba et al., 2019). This full search was run using two different feature layers: the 2nd-to-last layer (256-dim) and the penultimate layer (128-dim). Each training run lasted for a fixed number of epochs. Features and logits for both source and target datasets were saved at regular intervals, 20 times per training run. The final result was 760,000 datapoints: 10 algorithms * 100 steps of hyperparameter search * 20 checkpoints per training run * 19 tasks * 2 feature layers.

Table 2: The 10 UDA algorithms used to create the dataset of feature vectors.

| Algorithms | Category |
| --- | --- |
| ATDOC (Liang et al., 2021) | Pseudo labeling |
| BNM (Cui et al., 2020a) BSP (Chen et al., 2019) | SVD loss |
| CDAN (Long et al., 2017a) DANN (Ganin et al., 2016) GVB (Cui et al., 2020b) | Adversarial |
| IM (Shi & Sha, 2012) MCC (Jin et al., 2020) | Info max |
| MCD (Saito et al., 2018a) | Multiple classifier discrepancy |
| MMD (Long et al., 2015) | Feature distance |

### 2.2 Benchmarking the validation methods

We benchmarked the validators described in Sections 1.1 and 1.2, excluding those that are impractical to apply on a per-checkpoint basis. Computing scores per-checkpoint is preferred because it allows for faster feedback during training, and a greater likelihood of finding the optimal model. As well, it is how checkpoint selection is usually done in the supervised setting. The validation methods we excluded are:

- Reverse validation, which is typically applied per training run rather than per checkpoint.

- Proxy risk, which requires training a full UDA model per checkpoint. This increases total training-time complexity to $O(\texttt{epochs}^2)$, and is therefore not practical to use on a large scale.

- EMS, which requires access to a separate dataset with ground-truth target-labels.

We wanted all validators to give higher scores for better models, so we multiplied the scores of DEV, DEVN, and Entropy by $-1$. Each validator can have multiple variants by changing its parameters (see appendix section E).

The ideal validator ranks model checkpoints in the order of target-domain accuracy. Thus, a suitable evaluation metric is the Spearman ranking correlation between target-domain accuracy and validation scores. However, the Spearman correlation treats all samples equally, whereas we are more interested in the samples with high validation scores.

For example, consider a hypothetical set of validation scores that are perfectly correlated with accuracy, with the exception of the highest score that breaks the trend and returns a model with 0% accuracy. The set of scores with perfect correlation is useless, because ultimately, only the model with the highest validation score is selected. In this example, that model has 0% accuracy.

Thus, to account for this type of scenario, we use the weighted Spearman correlation (Bailey et al., 2018) to give more weight to the samples with high validation scores (see Figure 1). We set the weight of sample $x_i$ as:

$$\texttt{weight}(x_i) = \left( \frac{\texttt{rank}(v(x_i))}{\max\limits_{1 \leq k \leq N} \texttt{rank}(v(x_k))} \right)^2 \tag{12}$$

where

- $v(x_i)$ is the validation score of sample $x_i$.

- $\texttt{rank}(v(x_i))$ is the integer rank obtained by ranking all validation scores in ascending order, such that the lowest validation score has a rank of 1.

- $\max\limits_{1 \leq k \leq N} \texttt{rank}(v(x_k))$ is the maximum rank of all $N$ samples.

See appendix section D for details on how the weighted Spearman correlation is calculated.

### 2.3 How we define the average weighted Spearman correlation and top 5 training runs

In the next section, some figures and tables make use of the "average weighted Spearman correlation (WSC) across tasks", and the "top 5 training runs per algorithm/task pair".

A validator's average WSC across tasks is defined as:

$$\texttt{Average WSC across tasks} = \frac{1}{T} \sum_{i=1}^{T} \texttt{WSC}(A_i, V_i) \tag{13}$$

where $A_i$ is the set of target domain accuracies for task $i$, $V_i$ is the set of validator scores for task $i$, and $T$ is the number of tasks.

The top 5 training runs for an algorithm/task pair is defined as:

$$R_i = \max_{V \in r_i} V \tag{14}$$

$$\texttt{Top 5 Training Runs} = \texttt{sorted}(R)\texttt{[:5]} \tag{15}$$

where $R_i$ is the maximum validation score obtained in training run $i$, and $\texttt{sorted}(R)$ is $R$ sorted in descending order.

## 2.4 Advantage of the weighted Spearman correlation

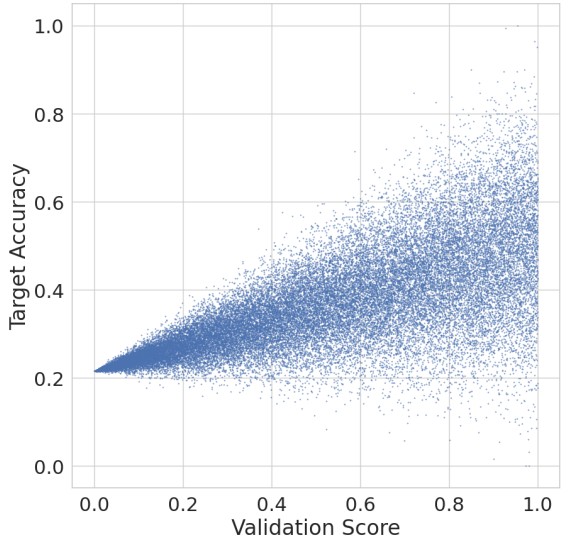
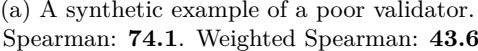

(a) A synthetic example of a poor validator. Spearman: **74.1**. Weighted Spearman: **43.6**.

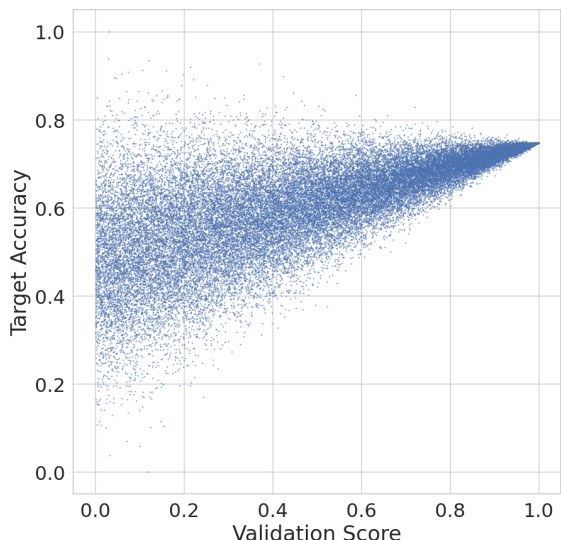

(b) A synthetic example of a better validator. Spearman: **74.2**. Weighted Spearman: **81.1**.

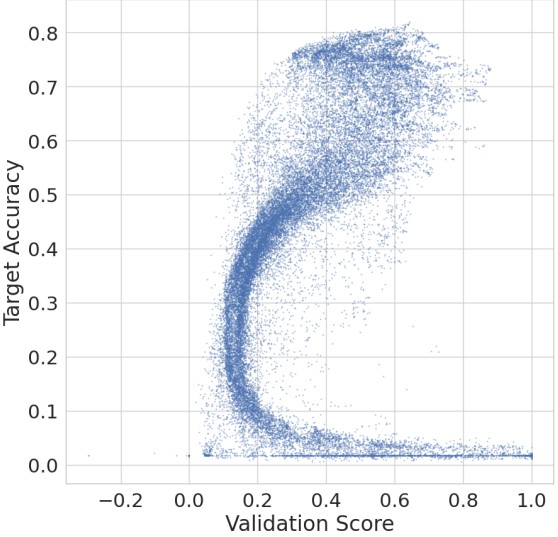

(c) ClassSS validator, OfficeHome Art → Real. Spearman: **60.2**. Weighted Spearman: **-2.7**.

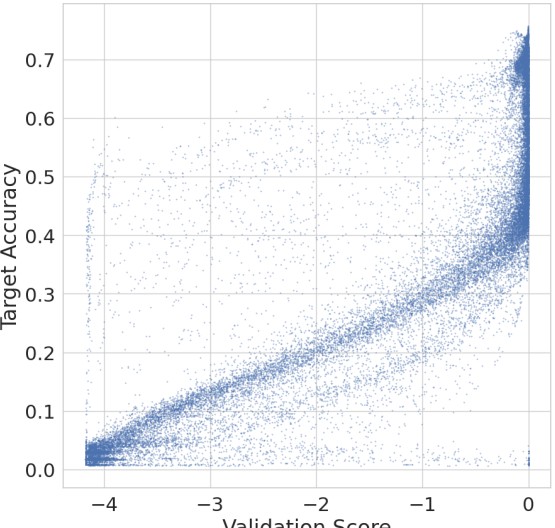

(d) Entropy validator, OfficeHome Real → Art. Spearman: **87.0**. Weighted Spearman: **43.2**.

Figure 1: These scatter plots show the advantage of the *weighted* Spearman correlation over the Spearman correlation. The Spearman correlation gives roughly the same score for Figures 1a and 1b. In contrast, our *weighted* Spearman correlation gives Figure 1a a much lower score, because there are many high validation scores corresponding with low accuracies. This means that during checkpoint selection, there is a high chance of selecting a low-accuracy checkpoint. In Figure 1c, the worst accuracies correspond with the highest validation scores, and in Figure 1d, accuracies ranging from 40% to 70% all have roughly the same validation score. The Spearman correlation treats all points equally and produces misleading scores for our purposes. In contrast, our *weighted* Spearman correlation emphasizes the samples with high validation scores, and heavily penalizes these two examples.

# 3 Results

Here are our main findings:

- Source validation accuracy (the baseline method) outperforms all other validators on Office31 and OfficeHome when applied to the checkpoints of all UDA algorithms simultaneously (see Figure 2). For all but one UDA algorithm, source validation accuracy is also the best validator on a per-algorithm basis (see Table 4).

- Our proposed BNM validator combined with the ATDOC algorithm comprises the best algorithm/validator pair, outperforming all other pairs, including those that use the baseline validation method (source validation accuracy). See Figure 3 and Table 4.

- Our proposed method DEVN vastly outperforms DEV (see Figure 4). This shows the importance of weight normalization for making DEV usable in actual applications, rather than in theory.

- Our proposed method ClassAMI significantly outperforms the original ClassSS, when applied to both source and target features (see Figure 5).

- SND consistently underperforms (see Figure 2 and Figure 6).

- When using an oracle validator (i.e. a validator that can directly compute target domain accuracy), most UDA algorithms outperform the source-only model (see Table 5a). However, when using non-oracle validators, UDA algorithms actually degrade accuracy in many cases (see Table 5b).

Table 3: The contributions of our proposed validators.

| Existing validator or algorithm | Our proposed validator | What our proposed validator achieves |
|---|---|---|
| BNM (used as an algorithm) | BNM (used as a validator) | State-of-the-art performance when paired with ATDOC |
| ClassSS (validator) | ClassAMI | Significantly better performance than ClassSS |
| DEV (validator) | DEVN | Significantly better performance than DEV |

# 4 Conclusion

Our benchmark comparison of validation methods reveals that:

- Our proposed validators significantly outperform the existing methods. However, the baseline (source validation accuracy) still leads the pack overall.

- Our BNM validator achieves state-of-the-art performance when used with the ATDOC algorithm.

- Even the best validators tend to pick sub-optimal checkpoints, which in many cases causes UDA algorithms to perform worse than untrained models (see Table 5). Thus, there is much room for improvement in the effectiveness of existing validation methods.

To unlock the full potential of UDA algorithms and models, more research is needed to improve validator accuracy and consistency. We hope our large-scale benchmark study, and our three new validators, will serve as a useful reference for future research in this area.

**Figures and tables**

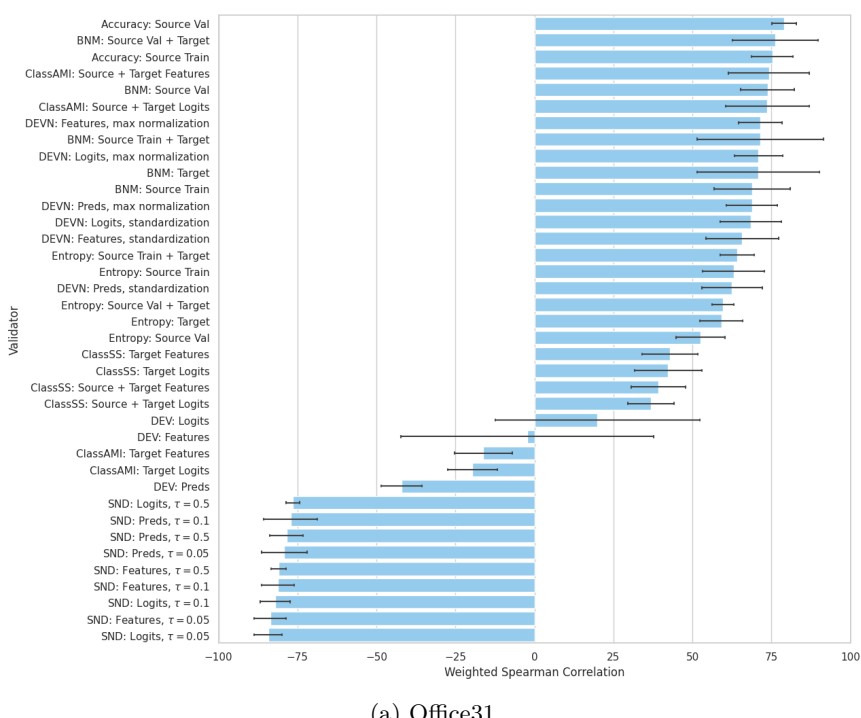

(a) Office31

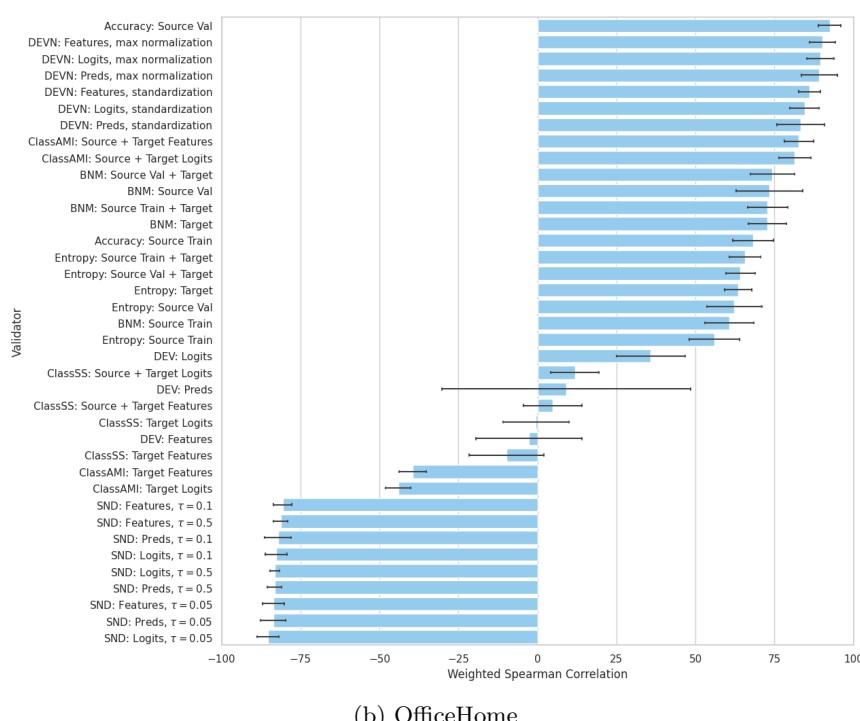

(b) OfficeHome

Figure 2: The weighted Spearman correlation per validator, averaged across the transfer tasks within Office31 and OfficeHome (see equation 13). The error bars represent the standard deviation across transfer tasks.

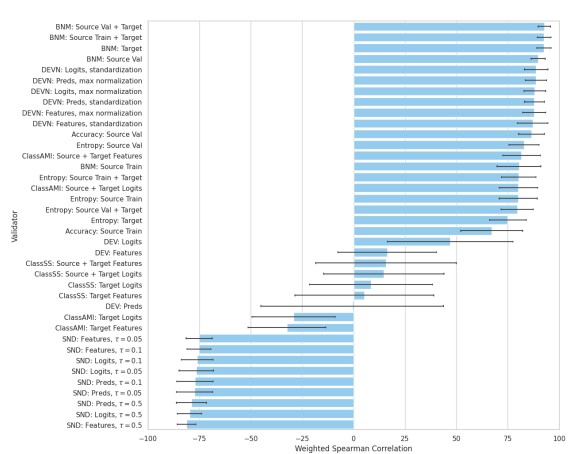

(a) The weighted Spearman correlation for validators paired with ATDOC, averaged across the Office31 and OfficeHome tasks (see equation 13). The BNM validator performs the best for this algorithm.

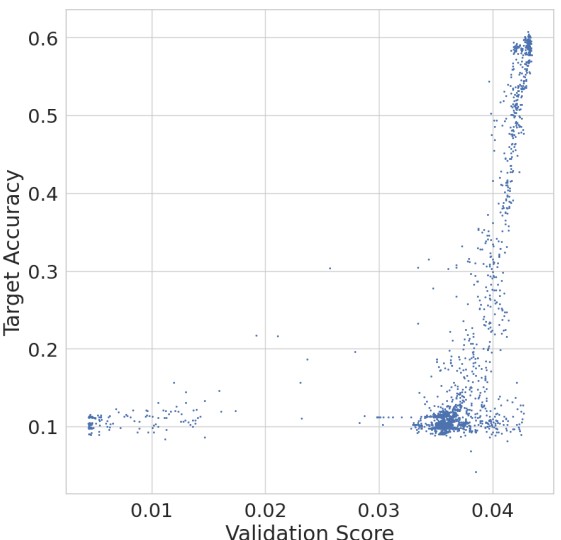

(b) BNM validation scores for the ATDOC algorithm on the MNIST→MNISTM task.

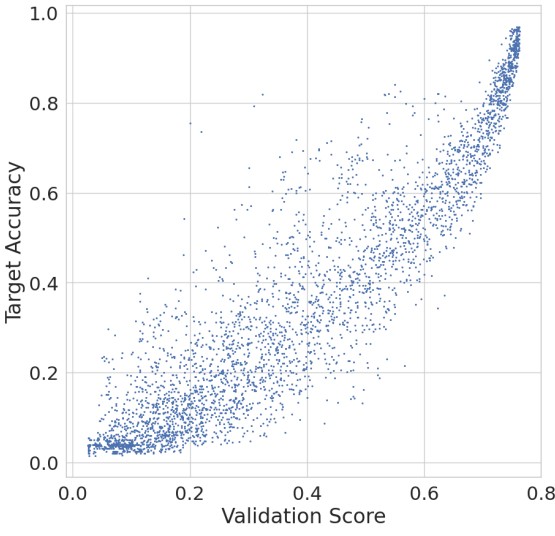

(c) BNM validation scores for the ATDOC algorithm on the Office31 DSLR → Webcam task.

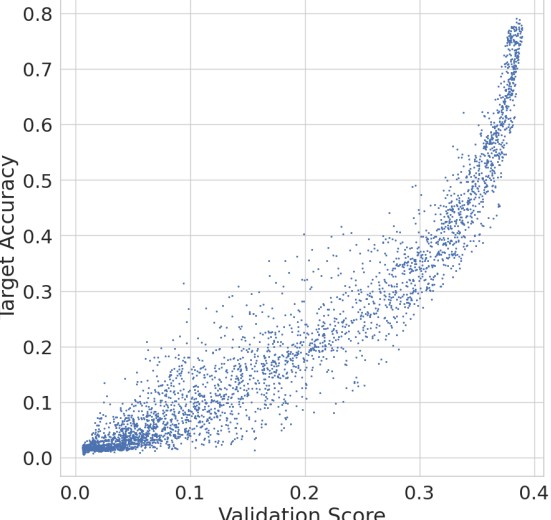

(d) BNM validation scores for the ATDOC algorithm on the OfficeHome Product → Real task.

Figure 3: The BNM validator is effective when paired with the ATDOC algorithm. Figures 3b-3d show the high correlation this pair achieves across MNIST, Office31, and OfficeHome. The notation X → Y means that X is the source domain, and Y is the target domain.

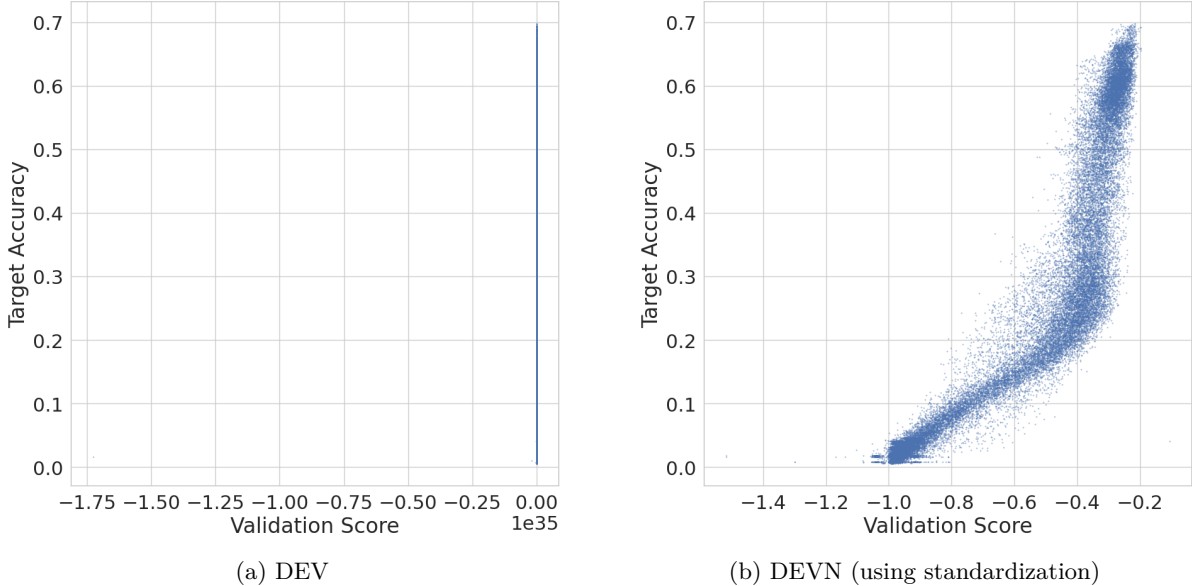

(a) DEV

(b) DEVN (using standardization)

Figure 4: DEV can produce scores approaching infinity (Figure 4a). Our proposed method, DEVN, fixes this problem by normalizing the sample weights. (Figure 4b). These plots are for the OfficeHome Clipart → Art task.

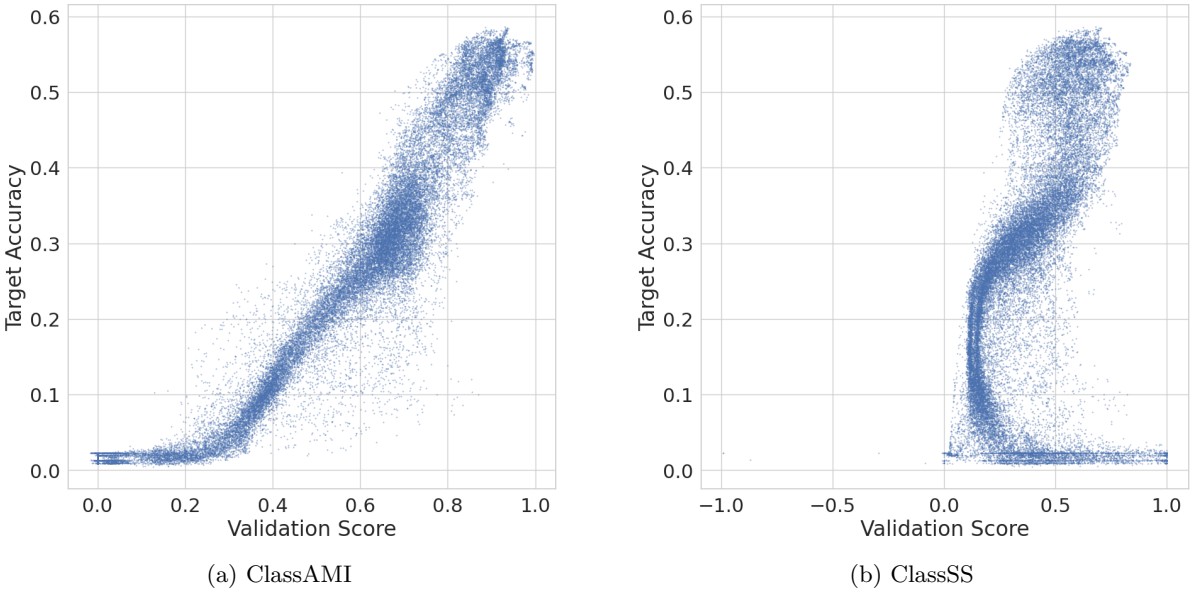

(a) ClassAMI

(b) ClassSS

Figure 5: We used the Adjusted Mutual Information (ClassAMI) instead of the Silhouette Score (ClassSS) to achieve a significant improvement for the class clustering validation method. These plots are for the OfficeHome Real → Clipart task.

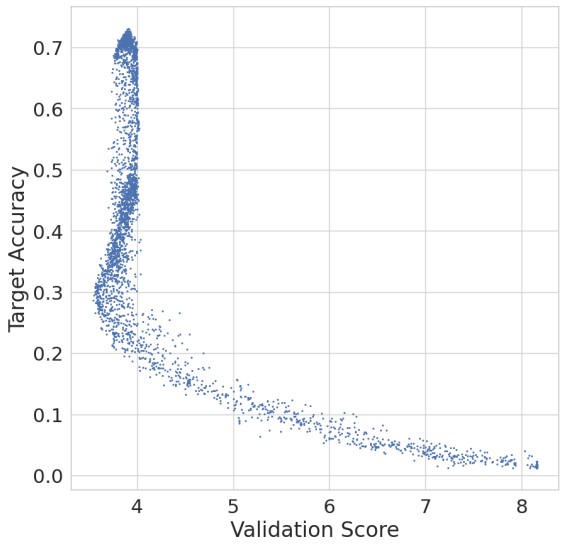 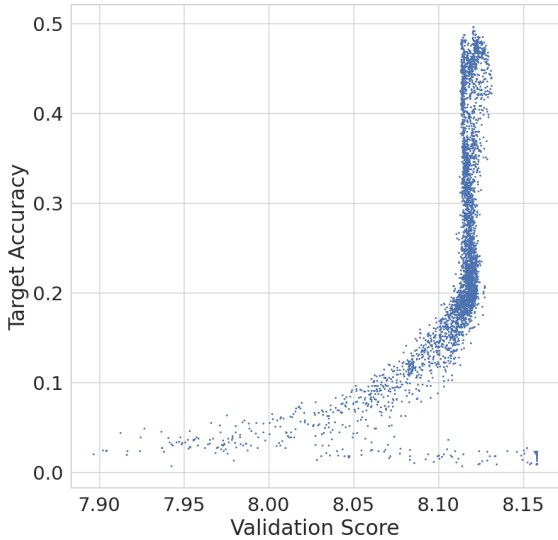

(a) SND applied to the checkpoints of the CDAN algorithm, on the OfficeHome Clipart → Real task. The weighted Spearman correlation is -93.1. This is SND applied to the prediction vectors, with $\tau = 0.05$.

(b) SND applied to the checkpoints of the MMD algorithm, on the OfficeHome Product → Clipart task. The weighted Spearman correlation is 4.4. This is SND applied to the feature vectors, with $\tau = 0.5$.

Figure 6: Examples of the SND validator being a poor predictor of accuracy.

Table 4: The best validator to use for each UDA algorithm. By "best", we mean that the validator scores the highest average weighted Spearman correlation (WSC) across Office31 and OfficeHome tasks (equation 13), for the checkpoints of a particular UDA algorithm.

| Algorithm | Validator | Parameters | Average WSC |
|-----------|-----------|------------|-------------|
| ATDOC | BNM | Source Val + Target | $92.7 \pm 3.0$ |
| BNM | Accuracy | Source Val | $89.4 \pm 9.0$ |
| BSP | Accuracy | Source Val | $83.6 \pm 11.2$ |
| CDAN | Accuracy | Source Val | $89.0 \pm 9.8$ |
| DANN | Accuracy | Source Val | $88.4 \pm 8.0$ |
| GVB | Accuracy | Source Val | $89.1 \pm 6.3$ |
| IM | Accuracy | Source Val | $89.1 \pm 7.2$ |
| MCC | Accuracy | Source Val | $89.5 \pm 5.4$ |
| MCD | Accuracy | Source Val | $90.9 \pm 6.5$ |
| MMD | Accuracy | Source Val | $88.9 \pm 8.1$ |

Table 5: Even the best validators tend to pick sub-optimal checkpoints, which in many cases causes UDA algorithms to perform worse than untrained models. These tables show the performance of UDA algorithms when using an oracle validator (Table 5a) and non-oracle validators (Table 5b). Each value is the average target-domain accuracy of the top 5 training runs, as determined by the validator (see equation 14). Green cells have an average accuracy greater than the source-only model. A stronger green color indicates higher accuracy. The color scheme is shared between the two tables, i.e. the colors in Table 5b are on the same scale as in Table 5a. Bold indicates the highest value per column, per table. Bolding in Table 5b is independent of the bolding in Table 5a.

(a) The performance of UDA algorithms when using an oracle validator.

| | MM | AD | AW | DA | DW | WA | WD | AC | AP | AR | CA | CP | CR | PA | PC | PR | RA | RC | RP |
|---|---|---|---|---|---|---|---|---|---|---|---|---|---|---|---|---|---|---|---|
| | | | | Office31 | | | | | | | | | OfficeHome | | | | | | |
| Source only | 57.6 | 82.7 | 80.4 | 69.6 | 94.3 | 71.5 | 99.0 | 41.2 | 68.6 | 76.7 | 60.2 | 67.6 | 70.5 | 60.0 | 42.8 | 76.2 | 68.8 | 44.7 | 79.1 |
| ATDOC | 60.4 | 88.8 | 84.4 | 72.1 | 96.6 | 72.3 | 99.6 | 47.5 | 72.3 | 76.8 | 63.9 | 74.6 | 74.0 | 65.9 | 47.9 | 78.4 | 73.6 | 52.2 | 80.9 |
| BNM | 63.3 | 89.3 | 91.3 | **73.5** | 97.4 | 74.8 | **100.0** | 52.3 | 74.2 | 79.9 | 67.0 | 74.2 | 76.7 | 66.8 | 51.9 | 80.4 | 72.5 | 56.8 | 81.9 |
| BSP | 57.1 | 85.3 | 78.8 | 69.3 | 96.4 | 69.9 | 99.8 | 43.9 | 67.8 | 76.2 | 60.1 | 64.9 | 70.0 | 60.2 | 42.1 | 76.0 | 69.6 | 45.6 | 77.5 |
| CDAN | 91.6 | 88.2 | 91.3 | 72.7 | 96.6 | 74.1 | 99.8 | 52.3 | 70.9 | 77.6 | 62.6 | 69.0 | 72.5 | 64.0 | 53.2 | 79.9 | 72.0 | 57.1 | 81.3 |
| DANN | **92.2** | 89.6 | 91.9 | 72.7 | 97.1 | 74.2 | 99.8 | 53.2 | 71.3 | 78.0 | 63.0 | 69.5 | 72.6 | 64.6 | 52.6 | 79.7 | 73.1 | **58.1** | 81.7 |
| GVB | 78.8 | 90.2 | 90.9 | 71.5 | 95.8 | 74.6 | **100.0** | 52.9 | 70.4 | 78.3 | 65.4 | 71.2 | 74.5 | 64.9 | 53.4 | 81.1 | 74.2 | 56.9 | 82.3 |
| IM | 63.2 | 89.1 | 91.3 | 72.7 | 96.6 | 75.0 | 99.9 | 52.7 | 73.4 | 80.3 | 67.0 | 74.0 | 76.4 | 66.2 | 51.1 | 80.9 | 73.1 | 56.5 | 81.9 |
| MCC | 68.9 | **93.2** | **92.8** | 73.2 | 97.6 | **75.2** | **100.0** | **55.1** | **74.7** | **81.1** | **69.7** | **75.9** | **77.6** | **68.3** | **54.5** | **82.5** | **75.3** | 58.0 | **83.5** |
| MCD | 68.1 | 87.8 | 83.3 | 67.0 | **98.7** | 66.4 | **100.0** | 43.3 | 67.7 | 75.2 | 59.8 | 68.5 | 70.8 | 61.8 | 44.9 | 77.3 | 72.0 | 49.4 | 80.6 |
| MMD | 71.6 | 87.8 | 88.0 | 72.1 | 97.2 | 72.1 | **100.0** | 50.5 | 71.1 | 77.4 | 64.2 | 69.7 | 72.1 | 64.9 | 48.7 | 78.8 | 72.2 | 52.8 | 80.5 |

(b) The performance of UDA algorithms when using the algorithm/validator pairs shown in Table 4.

| | MM | AD | AW | DA | DW | WA | WD | AC | AP | AR | CA | CP | CR | PA | PC | PR | RA | RC | RP |
|---|---|---|---|---|---|---|---|---|---|---|---|---|---|---|---|---|---|---|---|
| | | | | Office31 | | | | | | | | | OfficeHome | | | | | | |
| Source only | 57.6 | 82.7 | 80.4 | **69.6** | 94.3 | **71.5** | 99.0 | 41.2 | 68.6 | 76.7 | 60.2 | 67.6 | 70.5 | 60.0 | 42.8 | 76.2 | 68.8 | 44.7 | 79.1 |
| ATDOC | 58.7 | 82.5 | 83.5 | 68.1 | 95.2 | 67.9 | 97.1 | 42.5 | 69.7 | 71.9 | 54.7 | 68.3 | 66.3 | 64.3 | 38.4 | 76.8 | 71.1 | 48.3 | 79.1 |
| BNM | 52.2 | 83.8 | 85.1 | 63.2 | 93.1 | 66.3 | 96.1 | 50.6 | 72.0 | 78.2 | 62.2 | 69.6 | **73.7** | **65.6** | 46.3 | 79.0 | 71.0 | 54.0 | 80.4 |
| BSP | 18.2 | 80.4 | 74.4 | 44.1 | 84.0 | 43.7 | 96.8 | 41.1 | 66.6 | 74.9 | 56.7 | 58.5 | 65.2 | 54.7 | 35.7 | 74.8 | 67.3 | 44.1 | 75.8 |
| CDAN | 63.9 | 82.2 | 87.0 | 63.8 | 90.1 | 68.5 | 95.1 | 50.8 | 67.9 | 76.4 | 56.4 | 67.9 | 68.3 | 62.2 | **50.0** | 78.8 | 70.2 | 52.7 | 79.5 |
| DANN | **67.7** | 84.1 | **88.1** | 64.8 | 92.1 | 68.4 | 97.6 | 50.2 | 68.3 | 76.2 | 56.7 | 66.2 | 70.4 | 60.8 | 49.9 | 77.9 | 69.8 | 52.2 | 80.2 |
| GVB | 46.7 | 82.3 | 87.0 | 55.4 | 88.5 | 54.9 | 94.9 | 51.3 | 67.2 | 76.7 | 60.4 | 64.2 | 70.6 | 61.9 | **50.0** | 78.5 | 71.3 | 53.6 | 80.0 |
| IM | 47.4 | 81.3 | 85.5 | 65.4 | 90.9 | 63.9 | 96.2 | 51.5 | 71.9 | 78.9 | 62.7 | 71.8 | 72.9 | 63.8 | 49.1 | 79.9 | **71.8** | **55.6** | 79.8 |
| MCC | 50.2 | 85.6 | 86.8 | 58.0 | 94.0 | 70.4 | 98.2 | **51.8** | **72.8** | **79.9** | **64.4** | **73.3** | 73.2 | 63.3 | 48.9 | **80.1** | 71.5 | 53.7 | **81.5** |
| MCD | 13.2 | **86.7** | 79.4 | 44.0 | **96.6** | 45.9 | **100.0** | 42.3 | 65.3 | 73.7 | 56.6 | 66.3 | 68.6 | 60.6 | 41.6 | 74.4 | 68.8 | 47.1 | 79.7 |
| MMD | 57.9 | 79.7 | 79.9 | 53.3 | 88.7 | 63.3 | 96.4 | 47.6 | 69.0 | 76.8 | 58.0 | 65.4 | 69.2 | 62.6 | 45.8 | 76.8 | 70.7 | 51.4 | 78.9 |

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

# A    Glossary

- **Model**: a function that receives some input (e.g. photographic images), and returns a label for each item in that input.
- **Domain adaptation**: a type of machine-learning algorithm that repurposes existing models to work in new domains (a.k.a. target domains). For example, the existing model might work on photographs of food, whereas the target domain contains drawings of food.
- **Unsupervised domain adaptation (UDA)**: a type of domain adaptation where the target-domain does not have any existing class labels.
- **Validation method (validator)**: a function that evaluates how closely a model's output reflects certain attributes of the dataset, such as labels. The validator will return a quality score. Ideally, the quality score will indicate how similar the model's output is to the dataset attributes.
- **UDA validation method**: a validation method that estimates target domain accuracy, without having access to target labels. An effective UDA validation method is one that reliably estimates target-domain accuracy. For example, a higher score returned by an effective UDA validation method will reliably indicate that the target-domain accuracy is high.
- **Target-domain accuracy**: a model's accuracy in the target domain.
- **Oracle validation method**: a validation method that has access to existing target labels and is therefore able to directly compute target-domain accuracy. This contrasts with UDA, in which no target labels are available, and consequently, accuracy can only be estimated. When target labels are available, they should be used during training, as this will improve the model's target-domain accuracy. This type of training is known as semi-supervised or supervised domain adaptation. On the other hand, when target labels are not available, UDA is the only training method possible.

# B    What are validation methods, and why are they important?

In this paper, we compare the accuracy of various validation methods. Validation methods are functions that are used to evaluate the accuracy of machine-learning models, or in the case of this paper, unsupervised domain-adaptation models. This kind of research is essential, for the following reasons.

To date, most UDA papers have focused on improvements to the training procedure (algorithm), with the goal of maximizing target-domain accuracy. These papers tend to use the oracle validation method to evaluate their models, which is useful only when target labels are available. In contrast, when target labels are not available, the oracle validation method cannot be used. In that case, UDA validation methods are the only viable choice.

Unfortunately, without access to target labels, UDA validation methods tend to produce scores that are not 100% correlated with target-domain accuracy. For example, in an extreme case, the UDA validation method could return a high score for a low-accuracy model, and a low score for a high-accuracy model. Even in less extreme scenarios, the score might mislead the user into selecting a model that is not the most accurate one available. Yet achieving the highest possible accuracy is crucial in most application scenarios.

## B.1    How validation methods are used

Here is what a typical model-selection workflow looks like:

1. Select a UDA algorithm that you think will train your model effectively.
2. Set the hyperparameters either arbitrarily, or by using a hyperparameter optimizer. The UDA algorithm and hyperparameters will determine how your model is trained.
3. Use the UDA algorithm to train your model for an arbitrary amount of time, and save a version (checkpoint) of the model at arbitrary regular intervals.
4. To evaluate each checkpoint, employ whatever UDA validation method you think will correlate well with target-domain accuracy. The goal is to obtain validation scores that are as accurate as possible.
5. Keep the checkpoint with the highest validation score. Discard all other checkpoints.
6. Repeat steps 2-5 an arbitrary number of times, or until the best validation scores start to plateau.

At the end of this procedure, the model with the highest validation score will typically be deployed in some application. (For example, the model might be used on a smartphone to classify images of food.)

### B.2 Why validation methods are an important area of research

The above model-selection workflow optimizes for a high validation score, but the goal is to have a model with high target-domain accuracy. UDA validation scores are not perfectly correlated with target-domain accuracy. Thus, the model with the highest validation score might have sub-optimal target-domain accuracy. The lower the correlation, the more likely that a sub-optimal model will be selected inadvertently.

Our research shows that existing UDA validation methods have considerable room for improvement. For example, the model with the best validation score often has low target-domain accuracy. In other words, the model that gets chosen for deployment actually performs poorly, even though the validation method indicates that it performs well. Until UDA validation methods are able to produce more reliable results, it will be difficult to determine which models have the highest target-domain accuracy. As long as this is the case, the full potential of UDA algorithms will be unrealized.

Despite this fact, there are far more papers on UDA algorithms than on UDA validation methods. Yet validation methods have much more room for improvement. A UDA algorithm paper might improve target domain accuracy (as computed by an oracle) by a single percentage point, from 89% to 90% for example. But the checkpoint with the highest validation score might have only 70% target-domain accuracy. Thus, the validation method has a much larger effect on accuracy than the choice of UDA algorithm. Hence, research into UDA validation methods is crucial.

## C   Methodology summary

Our benchmarking process consisted of the following steps:

1. Create a dataset of model outputs:

   - Select ten UDA algorithms that represent a variety of algorithm types.

   - For each UDA algorithm, randomly create 100 hyperparameter settings.

   - For each hyperparameter setting, train a model on a UDA task for a fixed number of iterations, and save a model checkpoint at regular intervals. Each checkpoint consists of only the outputs of the model.

2. Calculate how accurate various validation methods are at ranking model checkpoints:

   - Select eight UDA validation methods.

   - Using each validation method in turn, compute a validation score for every checkpoint. (Remember that validation scores are computed using a UDA validation method. In contrast, target-domain accuracies are computed using the oracle validation method.)

   - For the results of each validation method, compute the weighted Spearman correlation.

## D   Calculating the weighted Spearman correlation

Our evaluation of validation methods is based on the weighted Spearman correlation described in Bailey et al. (2018) and implemented by van der Zee (2022). Here we show a summary of the calculations described in Bailey et al. (2018), with a few changes in notation.

The Pearson correlation is:

$$r_{\text{Pearson}} = \frac{\sum_{i=1}^{N}(x_i - \overline{x})(y_i - \overline{y})}{\sqrt{\sum_{i=1}^{N}(x_i - \overline{x})^2 \sum_{i=1}^{N}(y_i - \overline{y})^2}} \tag{16}$$

The weighted Pearson correlation scales the terms by a weight for each $(x, y)$ pair:

$$r_{\texttt{Weighted Pearson}} = \frac{\sum_{i=1}^{N} w_i(x_i - \widehat{x})(y_i - \widehat{y})}{\sqrt{\sum_{i=1}^{N} w_i(x_i - \widehat{x})^2 \sum_{i=1}^{N} w_i(y_i - \widehat{y})^2}} \tag{17}$$

$$\widehat{x} = \frac{\sum_{i=1}^{N} w_i x_i}{\sum_{i=1}^{N} w_i} \tag{18}$$

$$\widehat{y} = \frac{\sum_{i=1}^{N} w_i y_i}{\sum_{i=1}^{N} w_i} \tag{19}$$

where $w_i$ is the weight of each pair, and $N$ is the number of pairs.

The Spearman correlation is the Pearson correlation between rankings, i.e. $x_i$ and $y_i$ represent the rank of sample $i$ in the $x$ and $y$ variables. To compute the weighted Spearman correlation, let $\gamma_j$ be the weighted rank of the $j$th sample:

$$\gamma_j = a_j + b_j \tag{20}$$

$$a_j = \sum_{i=1}^{N} w_i \mathbb{1}(\texttt{rank}_i < \texttt{rank}_j) \tag{21}$$

$$b_j = \frac{t+1}{2} \overline{w_j} \tag{22}$$

$$\overline{w_j} = \frac{1}{t} \sum_{i=1}^{N} w_i \mathbb{1}(\texttt{rank}_i = \texttt{rank}_j) \tag{23}$$

$\mathbb{1}$ is a function that equals 1 if the expression inside is true, and 0 otherwise, and $t$ is the number of samples that have the same rank as sample $j$.

The $\gamma_j$ is computed for all $x$ and $y$ samples to obtain $\gamma_x$ and $\gamma_y$. Then $\gamma_x$ and $\gamma_y$ are plugged into equations 17-19, taking the place of the $x$ and $y$ variables.

# E  Validator parameters explained

Table 6: Each validator has its own settings. In the other tables and figures in this paper, we use short descriptions to indicate what settings are used. This table explains what those short descriptions mean. Note that we L2-normalized the inputs to ClassSS because it performed worse with un-normalized inputs.

| Validator | Parameters | Explanation |
|---|---|---|
| Accuracy | Source Train | `Accuracy(Source train predictions)` |
| | Source Val | `Accuracy(Source validation predictions)` |
| BNM | Source Train | `BNM(Source train predictions)` |
| | Source Train + Target | `BNM(Source train predictions) + BNM(Target predictions)` |
| | Source Val | `BNM(Source validation predictions)` |
| | Source Val + Target | `BNM(Source validation predictions) + BNM(Target predictions)` |
| | Target | `BNM(Target predictions)` |
| ClassAMI | Source + Target Features | `ClassAMI(concat(Source train features, Target features))` |
| | Source + Target Logits | `ClassAMI(concat(Source train logits, Target logits))` |
| | Target Features | `ClassAMI(Target features)` |
| | Target Logits | `ClassAMI(Target logits)` |
| ClassSS | Source + Target Features | `ClassSS(concat(Source train normalized features, Target normalized features))` |
| | Source + Target Logits | `ClassSS(concat(Source train normalized logits, Target normalized logits))` |
| | Target Features | `ClassSS(Target normalized features)` |
| | Target Logits | `ClassSS(Target normalized logits)` |
| DEV | Features | The discriminator is trained on feature vectors. |
| | Logits | The discriminator is trained on logits. |
| | Preds | The discriminator is trained on prediction vectors. |
| DEVN | Features, max normalization | The discriminator is trained on feature vectors. The sample weights are max-normalized. |
| | Features, standardization | The discriminator is trained on feature vectors. The sample weights are standardized. |
| | Logits, max normalization | The discriminator is trained on logits. The sample weights are max-normalized. |
| | Logits, standardization | The discriminator is trained on logits. The sample weights are standardized. |
| | Preds, max normalization | The discriminator is trained on prediction vectors. The sample weights are max-normalized. |
| | Preds, standardization | The discriminator is trained on prediction vectors. The sample weights are standardized. |
| Entropy | Source Train | `Entropy(Source train predictions)` |
| | Source Train + Target | `Entropy(Source train predictions) + Entropy(Target predictions)` |
| | Source Val | `Entropy(Source validation predictions)` |
| | Source Val + Target | `Entropy(Source validation predictions) + Entropy(Target predictions)` |
| | Target | `Entropy(Target predictions)` |
| SND | Features, $\tau = 0.05$ | The similarity matrix is derived from target features. Softmax temperature is 0.05. |
| | Features, $\tau = 0.1$ | The similarity matrix is derived from target features. Softmax temperature is 0.1. |
| | Features, $\tau = 0.5$ | The similarity matrix is derived from target features. Softmax temperature is 0.5. |
| | Logits, $\tau = 0.05$ | The similarity matrix is derived from target logits. Softmax temperature is 0.05. |
| | Logits, $\tau = 0.1$ | The similarity matrix is derived from target logits. Softmax temperature is 0.1 |
| | Logits, $\tau = 0.5$ | The similarity matrix is derived from target logits. Softmax temperature is 0.5 |
| | Preds, $\tau = 0.05$ | The similarity matrix is derived from target predictions. Softmax temperature is 0.05 |
| | Preds, $\tau = 0.1$ | The similarity matrix is derived from target predictions. Softmax temperature is 0.1 |
| | Preds, $\tau = 0.5$ | The similarity matrix is derived from target predictions. Softmax temperature is 0.5 |

Table 7: The validator parameters categorized by function.

| Validators | Parameters |
|---|---|
| Accuracy, BNM, Entropy | Dataset splits |
| ClassAMI, ClassSS | Dataset splits, Choice of feature vector |
| DEV | Choice of feature vector |
| DEVN | Choice of feature vector, type of normalization |
| SND | Choice of feature vector, softmax temperature |

## F   Training methodology details

Table 8: A list of the models used in our experiments. We created the dataset of model outputs using two feature layers: FL3 and FL6. Every checkpoint contains the feature layer (FL3 or FL6) and the logits. The discriminator is used only for adversarial methods. It receives the feature layer as input, but keeps the same depth regardless of feature layer.

|  | Layers | Feature name |
|---|---|---|
| Trunk | `LeNet` or `ResNet50` | |
| Classifier | `Linear(256)` `ReLU()` `Dropout(0.5)` `Linear(128)` `ReLU()` `Dropout(0.5)` `Linear(num_cls)` `Softmax()` | FL3 FL6 `Logits` `Preds` |
| Discriminator | `Linear(2048)` `ReLU()` `Linear(2048)` `ReLU()` `Linear(1)` | |

Table 9: A list of various experiment settings. The learning rate (lr) is one of the hyperparameters. The same lr is used by trunk, classifier, and discriminator.

| Category | Settings |
|---|---|
| Optimizer | Adam (Kingma & Ba, 2014) Weight decay of `1e-4` lr $\in$ `log([1e-5,0.1])` |
| LR scheduler | One Cycle (Smith & Topin, 2019) 5% warmup period $\text{lr}_{init} = \text{lr}_{max}/100$ $\text{lr}_{final} = 0$ Cosine annealing |
| Batch size | 64 source + 64 target |
| Epochs / checkpoint interval | Digits: 100 / 5 Office31 (W and D as target): 2000 / 100 Office31 (A as target): 200 / 10 OfficeHome: 200 / 10 |
| Training image transforms | `Resize(256)` `RandomCrop(224)` `RandomHorizontalFlip()` `Normalize()` |
| Validation image transforms | `Resize(256)` `CenterCrop(224)` `Normalize()` |
| MNIST image transforms | `Resize(32)` `GrayscaleToRGB()` `Normalize()` |

Table 10: A list of the hyperparameter search settings used in the experiment.

| Algorithm | Hyperparameter | Search space |
|---|---|---|
| ATDOC | $\lambda_{atdoc}$ | [0,1] |
| | $k_{atdoc}$ | int([5, 25], step=5) |
| | $\lambda_L$ | [0,1] |
| BNM | $\lambda_{bnm}$ | [0,1] |
| | $\lambda_L$ | [0,1] |
| BSP | $\lambda_{bsp}$ | log([1e-6,1]) |
| | $\lambda_L$ | [0,1] |
| CDAN | $\lambda_D$ | [0,1] |
| | $\lambda_G$ | [0,1] |
| | $\lambda_L$ | [0,1] |
| DANN | $\lambda_D$ | [0,1] |
| | $\lambda_{grl}$ | log([0.1,10]) |
| | $\lambda_L$ | [0,1] |
| GVB | $\lambda_D$ | [0,1] |
| | $\lambda_{B_G}$ | [0,1] |
| | $\lambda_{B_D}$ | [0,1] |
| | $\lambda_{grl}$ | log([0.1,10]) |
| IM | $\lambda_{imax}$ | [0,1] |
| | $\lambda_L$ | [0,1] |
| MMD | $\lambda_F$ | [0,1] |
| | $\lambda_L$ | [0,1] |
| | $\gamma_{exp}$ | int([1,8]) |
| MCC | $\lambda_{mcc}$ | [0,1] |
| | $T_{mcc}$ | [0.2,5] |
| | $\lambda_L$ | [0,1] |
| MCD | $N_{mcd}$ | int([1,10]) |
| | $\lambda_L$ | [0,1] |
| | $\lambda_{disc}$ | [0,1] |

Table 11: Description of every hyperparameter that is mentioned in Table 10.

| Hyperparameter | Description |
|---|---|
| $\lambda_{atdoc}$ | ATDOC loss weight |
| $\lambda_{bnm}$ | BNM loss weight |
| $\lambda_{bsp}$ | BSP loss weight |
| $\lambda_{disc}$ | Classifier discrepancy loss weight for MCD |
| $\lambda_{grl}$ | Gradient reversal weight, i.e. gradients are multiplied by $-\lambda_{grl}$ |
| $\lambda_{imax}$ | Information maximization loss weight |
| $\lambda_{mcc}$ | MCC loss weight |
| $\lambda_{B_G}$ | Generator bridge loss weight for GVB |
| $\lambda_{B_D}$ | Discriminator bridge loss weight for GVB |
| $\lambda_D$ | Discriminator loss weight |
| $\lambda_F$ | Feature distance loss weight |
| $\lambda_G$ | Generator loss weight |
| $\lambda_L$ | Source classification loss weight |
| $\gamma_{exp}$ | Exponent of the bandwidth multiplier for MMD. For example, if $\gamma_{exp} = 2$, then the bandwidths used will be $\{2^{-2}x, 2^{-1}x, 2^0x, 2^1x, 2^2x\}$, where $x$ is the base bandwidth. |
| $k_{atdoc}$ | Number of nearest neighbors to retrieve for computing pseudolabels in ATDOC |
| $N_{mcd}$ | Number of times the MCD generator is updated per batch |
| $T_{mcc}$ | Softmax temperature used by MCC |

## G    Correlation tables

Table 12: Summary of tables that show the weighted Spearman correlation of the benchmarked validators.

| Algorithm | Office31 & OfficeHome | MNIST→MNISTM (MM) |
|---|---|---|
| All combined | Table 13 | Table 24 |
| ATDOC | Table 14 | Table 25 |
| BNM | Table 15 | Table 25 |
| BSP | Table 16 | Table 25 |
| CDAN | Table 17 | Table 25 |
| DANN | Table 18 | Table 25 |
| GVB | Table 19 | Table 25 |
| IM | Table 20 | Table 25 |
| MCC | Table 21 | Table 25 |
| MCD | Table 22 | Table 25 |
| MMD | Table 23 | Table 25 |

**What the green coloring means**

For all tables, the green coloring indicates better performance. The greener the cell color, the better the performance, compared to the Source Val Accuracy validator. The best value per column is bolded. The Mean and Std columns are the mean and standard deviation of all task or algorithm columns. A high mean and low standard deviation reflect good performance.

### G.1    Weighted Spearman Correlation for Office31 and OfficeHome

Table 13: The weighted Spearman correlation of each validator/task pair, using the checkpoints of all algorithms simultaneously.

| | | AD | AW | DA | DW | WA | WD | AC | AP | AR | CA | CP | CR | PA | PC | PR | RA | RC | RP | Mean | Std |
|---|---|---|---|---|---|---|---|---|---|---|---|---|---|---|---|---|---|---|---|---|---|
| Accuracy | Source Train | 74.4 | 75.5 | 85.2 | 72.7 | 80.1 | 63.7 | 61.3 | 65.3 | 63.4 | 59.3 | 63.7 | 64.5 | 77.4 | 72.6 | 79.2 | 63.9 | 73.7 | 73.9 | 70.5 | 7.3 |
| | Source Val | 81.6 | 84.2 | 77.0 | 77.2 | 81.7 | 72.4 | 95.9 | 95.5 | 96.9 | 84.2 | 88.9 | 89.0 | 92.9 | 90.9 | 95.1 | 94.1 | 92.6 | 94.3 | 88.0 | 7.4 |
| BNM | Source Train | 74.4 | 76.8 | 80.1 | 59.4 | 76.1 | 46.4 | 60.5 | 65.8 | 53.7 | 51.4 | 55.4 | 53.9 | 71.3 | 68.3 | 72.1 | 48.3 | 67.0 | 60.6 | 63.4 | 10.2 |
| | Source Train + Target | 74.1 | 83.4 | 86.2 | 70.4 | 85.8 | 28.9 | 76.8 | 79.2 | 72.7 | 66.1 | 74.6 | 71.4 | 72.2 | 78.0 | 77.2 | 56.5 | 79.9 | 69.5 | 72.4 | 12.6 |
| | Source Val | 75.7 | 77.6 | 79.3 | 73.3 | 81.2 | 55.6 | 83.2 | 84.0 | 76.5 | 53.2 | 60.1 | 56.3 | 79.9 | 79.0 | 78.6 | 69.1 | 83.4 | 76.9 | 73.5 | 9.9 |
| | Source Val + Target | 76.8 | 83.0 | 85.6 | 77.4 | 87.1 | 47.3 | 81.4 | 83.1 | 75.9 | 63.6 | 70.5 | 66.7 | 74.3 | 79.3 | 78.3 | 61.5 | 82.4 | 74.4 | 74.9 | 9.7 |
| | Target | 73.1 | 81.6 | 84.8 | 71.2 | 85.4 | 29.4 | 75.5 | 78.4 | 72.7 | 66.8 | 75.6 | 72.8 | 71.6 | 77.1 | 76.9 | 56.7 | 79.3 | 69.1 | 72.1 | 12.2 |
| ClassAMI | Source + Target Features | 75.9 | 83.8 | 85.3 | 60.7 | 86.2 | 53.4 | 84.5 | 83.6 | 81.1 | 80.0 | 83.7 | 85.7 | 83.4 | 86.8 | 85.5 | 69.8 | 89.1 | 79.7 | 79.9 | 9.2 |
| | Source + Target Logits | 74.9 | 84.1 | 85.1 | 59.4 | 86.2 | 52.5 | 83.9 | 82.7 | 79.9 | 77.4 | 82.1 | 83.5 | 81.4 | 86.7 | 84.2 | 68.2 | 89.0 | 78.4 | 78.9 | 9.4 |
| | Target Features | -14.9 | -35.4 | -14.2 | -9.1 | -16.4 | -7.5 | -45.5 | -40.9 | -41.6 | -42.2 | -43.7 | -42.1 | -34.1 | -30.8 | -34.6 | -40.2 | -35.7 | -42.2 | -31.7 | 12.7 |
| | Target Logits | -18.5 | -35.6 | -17.1 | -14.9 | -21.2 | -10.7 | -55.0 | -43.9 | -42.7 | -45.6 | -46.9 | -44.0 | -39.0 | -40.8 | -39.1 | -44.1 | -44.0 | -44.1 | -36.0 | 12.8 |
| ClassSS | Source + Target Features | 34.4 | 48.5 | 43.0 | 41.9 | 44.6 | 22.4 | -7.7 | 0.2 | 8.8 | -14.3 | 6.2 | 1.1 | 0.9 | 17.0 | 15.1 | 2.8 | 12.4 | 15.2 | 16.3 | 18.5 |
| | Source + Target Logits | 39.2 | 41.6 | 39.4 | 36.4 | 43.2 | 21.2 | 1.9 | 10.8 | 14.0 | -4.9 | 17.1 | 11.8 | 5.5 | 21.1 | 19.8 | 10.2 | 12.8 | 22.0 | 20.2 | 14.0 |
| | Target Features | 44.4 | 58.6 | 39.6 | 47.6 | 36.1 | 30.9 | -30.4 | -10.3 | -2.7 | -28.7 | -0.3 | -5.7 | -23.8 | -9.5 | 0.9 | -9.2 | -10.0 | 10.9 | 7.7 | 27.2 |
| | Target Logits | 45.5 | 60.3 | 34.1 | 49.8 | 34.2 | 29.7 | -12.1 | -0.6 | 3.3 | -20.0 | 9.9 | 3.7 | -17.8 | 6.4 | 5.5 | -2.7 | 2.5 | 15.8 | 13.7 | 22.8 |
| DEV | Features | -52.0 | -59.1 | 19.8 | 33.3 | 43.8 | 0.8 | 26.3 | 30.6 | 7.9 | -22.4 | -15.0 | -6.3 | -11.2 | -0.0 | -5.5 | -27.5 | -0.4 | -8.9 | -2.6 | 26.8 |
| | Logits | -19.3 | -22.3 | 43.9 | 45.9 | 60.5 | 11.1 | 48.2 | 48.3 | 37.1 | 27.1 | 30.0 | 41.5 | 24.9 | 39.5 | 40.8 | 9.6 | 47.2 | 36.2 | 30.6 | 22.0 |
| | Preds | -36.4 | -45.5 | -35.9 | -40.2 | -40.1 | -54.7 | -17.0 | -12.6 | -19.4 | -33.0 | 48.1 | 55.3 | -41.9 | 41.9 | 47.6 | -49.7 | 49.5 | 40.2 | -8.0 | 40.4 |
| DEVN | Features, max normalization | 77.4 | 77.6 | 74.7 | 62.7 | 75.7 | 61.2 | 94.5 | 94.7 | 95.6 | 81.9 | 84.9 | 85.5 | 91.8 | 87.9 | 92.6 | 90.4 | 90.4 | 91.1 | 83.9 | 10.2 |
| | Features, standardization | 61.9 | 53.8 | 82.8 | 56.1 | 80.4 | 59.3 | 89.7 | 91.6 | 89.5 | 83.6 | 82.5 | 81.4 | 90.0 | 81.6 | 87.0 | 85.1 | 86.4 | 85.0 | 79.3 | 12.0 |
| | Logits, max normalization | 77.2 | 80.7 | 70.6 | 62.3 | 74.9 | 59.9 | 94.5 | 94.5 | 95.4 | 81.0 | 84.5 | 85.0 | 90.9 | 86.6 | 91.6 | 89.7 | 89.8 | 91.2 | 83.4 | 10.4 |
| | Logits, standardization | 68.3 | 73.1 | 76.2 | 58.6 | 80.9 | 53.5 | 90.7 | 91.1 | 90.2 | 80.3 | 79.9 | 78.2 | 87.8 | 78.0 | 86.1 | 81.9 | 84.8 | 85.6 | 79.2 | 10.1 |
| | Preds, max normalization | 77.5 | 81.1 | 64.9 | 60.7 | 69.0 | 59.8 | 94.7 | 94.2 | 95.3 | 76.8 | 82.5 | 81.0 | 90.1 | 89.7 | 92.2 | 89.1 | 92.2 | 92.4 | 82.4 | 11.6 |
| | Preds, standardization | 70.4 | 75.4 | 63.3 | 49.4 | 65.6 | 51.2 | 89.9 | 90.2 | 90.3 | 63.9 | 77.6 | 74.4 | 83.9 | 85.0 | 86.4 | 82.2 | 88.0 | 87.4 | 76.4 | 12.8 |
| Entropy | Source Train | 69.2 | 69.1 | 71.8 | 55.6 | 68.0 | 44.4 | 57.7 | 62.3 | 48.4 | 46.3 | 50.6 | 49.9 | 63.6 | 65.0 | 67.7 | 43.2 | 63.6 | 53.6 | 58.3 | 9.2 |
| | Source Train + Target | 71.3 | 71.3 | 58.5 | 64.0 | 61.0 | 58.6 | 66.7 | 72.8 | 68.4 | 58.9 | 66.8 | 63.7 | 63.2 | 67.7 | 71.1 | 55.3 | 70.5 | 63.4 | 65.2 | 5.1 |
| | Source Val | 41.4 | 45.1 | 59.1 | 58.7 | 61.7 | 49.5 | 61.6 | 65.0 | 67.9 | 43.8 | 51.8 | 49.2 | 67.6 | 70.1 | 71.1 | 63.0 | 70.6 | 65.3 | 59.0 | 9.5 |
| | Source Val + Target | 63.9 | 58.2 | 56.4 | 64.5 | 59.9 | 55.2 | 62.9 | 69.1 | 69.0 | 55.3 | 62.8 | 60.0 | 63.3 | 67.0 | 70.6 | 58.5 | 68.8 | 63.8 | 62.7 | 4.8 |
| | Target | 65.9 | 62.7 | 46.2 | 63.6 | 54.1 | 62.2 | 63.2 | 69.9 | 67.3 | 58.6 | 66.7 | 64.1 | 59.8 | 64.0 | 67.9 | 54.4 | 66.2 | 61.0 | 62.1 | 5.7 |
| SND | Features, $\tau = 0.05$ | -82.4 | -82.1 | -77.9 | -91.6 | -78.8 | -89.1 | -80.4 | -82.9 | -90.3 | -77.7 | -85.5 | -85.3 | -77.9 | -83.3 | -86.1 | -85.6 | -81.7 | -85.9 | -83.6 | 4.1 |
| | Features, $\tau = 0.1$ | -81.2 | -80.6 | -74.8 | -87.5 | -75.0 | -87.6 | -79.1 | -80.3 | -86.2 | -75.9 | -82.1 | -82.7 | -75.7 | -80.1 | -82.8 | -80.7 | -78.5 | -82.8 | -80.7 | 3.8 |
| | Features, $\tau = 0.5$ | -83.3 | -84.7 | -80.7 | -77.9 | -81.0 | -78.6 | -82.6 | -82.9 | -79.9 | -81.6 | -83.8 | -84.5 | -82.2 | -80.1 | -82.2 | -75.6 | -79.3 | -81.2 | -81.2 | 2.3 |
| | Logits, $\tau = 0.05$ | -83.7 | -82.1 | -78.1 | -90.8 | -81.6 | -89.6 | -81.9 | -85.8 | -92.0 | -80.1 | -87.6 | -87.9 | -80.2 | -83.9 | -87.8 | -87.3 | -82.9 | -87.0 | -85.0 | 3.9 |
| | Logits, $\tau = 0.1$ | -82.7 | -80.2 | -75.1 | -87.4 | -78.7 | -88.6 | -80.3 | -83.6 | -89.6 | -78.4 | -85.2 | -85.7 | -77.3 | -79.6 | -85.3 | -83.9 | -79.4 | -83.9 | -82.5 | 4.0 |
| | Logits, $\tau = 0.5$ | -79.2 | -78.7 | -73.3 | -75.9 | -74.2 | -77.3 | -82.7 | -82.7 | -86.2 | -79.8 | -84.2 | -84.6 | -82.8 | -82.8 | -84.6 | -82.0 | -83.3 | -82.8 | -80.9 | 3.6 |
| | Preds, $\tau = 0.05$ | -80.4 | -79.1 | -68.4 | -88.8 | -71.8 | -86.5 | -79.7 | -84.8 | -90.8 | -78.4 | -87.0 | -86.2 | -76.4 | -81.3 | -86.0 | -85.7 | -81.7 | -86.1 | -82.2 | 5.7 |
| | Preds, $\tau = 0.1$ | -78.7 | -76.7 | -64.9 | -87.1 | -68.7 | -87.2 | -78.6 | -83.0 | -89.9 | -77.4 | -85.7 | -84.9 | -74.1 | -78.5 | -84.4 | -84.8 | -79.7 | -84.9 | -80.5 | 6.4 |
| | Preds, $\tau = 0.5$ | -79.2 | -78.8 | -71.2 | -85.4 | -72.6 | -83.9 | -82.1 | -84.1 | -86.9 | -80.7 | -85.5 | -85.5 | -78.8 | -81.0 | -84.4 | -83.4 | -82.4 | -84.0 | -81.6 | 4.2 |

Table 14: The weighted Spearman correlation of each validator/task pair for **ATDOC**.

| | | AD | AW | DA | DW | WA | WD | AC | AP | AR | CA | CP | CR | PA | PC | PR | RA | RC | RP | Mean | Std |
|---|---|---|---|---|---|---|---|---|---|---|---|---|---|---|---|---|---|---|---|---|---|
| Accuracy | Source Train | 79.7 | 84.4 | 81.4 | 82.5 | 72.0 | 88.7 | 49.1 | 54.6 | 46.7 | 72.0 | 42.7 | 56.4 | 86.8 | 56.6 | 80.0 | 49.9 | 58.4 | 68.5 | 67.2 | 15.0 |
| | Source Val | 82.3 | 88.3 | 85.3 | 89.5 | 74.1 | 90.5 | 90.5 | 92.0 | 91.4 | 86.8 | 79.7 | 80.1 | 90.7 | 72.3 | 89.1 | 93.4 | 87.1 | 93.8 | 86.5 | 6.2 |
| BNM | Source Train | 86.0 | 90.0 | 89.5 | 90.9 | 81.7 | 88.6 | 74.6 | 81.3 | 66.8 | 85.1 | 63.9 | 79.9 | 93.0 | 76.8 | 91.4 | 51.7 | 75.9 | 81.8 | 80.5 | 10.7 |
| | Source Train + Target | 92.5 | 93.9 | 94.6 | 95.3 | 85.2 | 94.6 | 86.6 | 92.7 | 89.7 | 94.6 | 94.0 | 94.0 | 95.8 | 90.8 | 97.7 | 91.6 | 87.4 | 95.3 | 92.6 | 3.3 |
| | Source Val | 87.3 | 91.0 | 90.8 | 92.7 | 82.4 | 89.3 | 90.3 | 93.6 | 91.0 | 89.2 | 83.7 | 88.9 | 95.0 | 87.2 | 95.6 | 89.9 | 85.8 | 92.0 | 89.8 | 3.5 |
| | Source Val + Target | 91.9 | 93.5 | 93.6 | 94.6 | 84.2 | 94.1 | 89.8 | 93.8 | 92.4 | 93.7 | 92.6 | 93.4 | 95.8 | 91.2 | 97.5 | 94.0 | 87.2 | 94.9 | 92.7 | 3.0 |
| | Target | 91.5 | 92.7 | 93.2 | 94.5 | 83.4 | 94.4 | 86.8 | 92.5 | 91.7 | 95.1 | 96.3 | 94.4 | 94.4 | 91.2 | 97.9 | 92.6 | 87.8 | 96.2 | 92.6 | 3.5 |
| ClassAMI | Source + Target Features | 81.5 | 88.6 | 88.2 | 86.8 | 86.5 | 86.2 | 79.7 | 87.9 | 70.2 | 87.9 | 82.4 | 85.6 | 91.2 | 76.3 | 91.0 | 55.4 | 66.2 | 78.7 | 81.7 | 9.2 |
| | Source + Target Logits | 81.6 | 89.0 | 87.2 | 87.0 | 82.9 | 87.6 | 75.2 | 86.9 | 70.6 | 85.4 | 80.6 | 78.7 | 90.5 | 69.4 | 90.3 | 58.3 | 62.0 | 80.1 | 80.2 | 9.3 |
| | Target Features | -27.2 | -42.5 | -16.0 | 4.5 | -19.1 | 7.7 | -54.1 | -46.9 | -49.6 | -47.7 | -31.2 | -49.0 | -23.6 | -23.2 | -24.8 | -36.7 | -65.6 | -38.1 | -32.4 | 18.8 |
| | Target Logits | -26.7 | -43.8 | -5.9 | 4.6 | -16.5 | 14.5 | -58.3 | -39.0 | -46.4 | -45.9 | -25.0 | -44.0 | -21.6 | -28.6 | -22.2 | -24.8 | -68.1 | -27.5 | -29.2 | 20.2 |
| ClassSS | Source + Target Features | 61.2 | 63.4 | 38.6 | 65.1 | 26.7 | 77.5 | -27.5 | -2.3 | -1.4 | -24.7 | -4.8 | -25.5 | 16.9 | -9.2 | 29.9 | 16.3 | -36.5 | 21.9 | 15.9 | 34.2 |
| | Source + Target Logits | 59.6 | 31.7 | 25.9 | 63.0 | 25.4 | 64.9 | -29.8 | -7.5 | 5.2 | -14.4 | 15.8 | -9.7 | 17.2 | -3.9 | 36.3 | 5.2 | -40.6 | 20.0 | 14.7 | 29.3 |
| | Target Features | 25.0 | 37.2 | 42.7 | 61.0 | 26.0 | 69.1 | -43.9 | -8.7 | -17.4 | -39.1 | 33.9 | -21.2 | -13.6 | -18.6 | -1.1 | -3.0 | -44.1 | 10.3 | 5.2 | 33.8 |
| | Target Logits | 33.8 | 51.6 | 27.3 | 65.9 | 24.6 | 58.8 | -27.6 | -11.1 | -8.4 | -33.6 | 27.3 | -16.3 | -10.0 | -6.5 | 9.7 | -9.2 | -30.6 | 7.2 | 8.5 | 29.9 |
| DEV | Features | -16.2 | -7.1 | 36.9 | 70.1 | 8.9 | 34.5 | 30.9 | 42.1 | 22.5 | -0.7 | 2.4 | 15.5 | 25.3 | 32.2 | 35.9 | -27.4 | -8.9 | -2.3 | 16.4 | 23.9 |
| | Logits | -11.9 | -2.2 | 48.8 | 76.2 | 35.5 | -0.8 | 48.6 | 60.3 | 32.8 | 68.8 | 50.7 | 83.1 | 78.7 | 68.9 | 84.5 | 2.8 | 69.9 | 50.3 | 46.9 | 30.5 |
| | Preds | -25.6 | -33.8 | -25.2 | -24.7 | -26.5 | -60.0 | -26.4 | -15.5 | -19.5 | -27.4 | 56.5 | 75.7 | -26.0 | 61.1 | 64.0 | -56.4 | 66.5 | 32.4 | -0.6 | 44.4 |
| DEVN | Features, max normalization | 83.2 | 89.0 | 86.7 | 91.0 | 74.9 | 90.4 | 91.0 | 91.7 | 90.1 | 89.5 | 83.8 | 83.6 | 93.2 | 75.2 | 91.8 | 93.4 | 88.6 | 94.2 | 87.8 | 5.5 |
| | Features, standardization | 75.6 | 81.3 | 87.3 | 92.1 | 67.9 | 89.6 | 92.4 | 93.1 | 78.7 | 92.2 | 87.4 | 87.3 | 94.9 | 78.9 | 94.2 | 89.6 | 91.6 | 94.6 | 87.1 | 7.4 |
| | Logits, max normalization | 83.8 | 89.8 | 87.3 | 91.7 | 75.9 | 90.2 | 91.6 | 92.2 | 90.6 | 89.2 | 84.4 | 84.5 | 93.0 | 75.3 | 91.7 | 92.5 | 88.3 | 94.0 | 88.1 | 5.3 |
| | Logits, standardization | 78.5 | 87.7 | 90.1 | 93.0 | 75.6 | 88.9 | 93.9 | 93.5 | 84.4 | 92.2 | 89.9 | 92.7 | 94.5 | 80.2 | 94.9 | 84.5 | 91.4 | 92.2 | 88.8 | 5.7 |
| | Preds, max normalization | 84.5 | 89.7 | 85.9 | 89.6 | 75.3 | 90.0 | 93.5 | 93.2 | 92.5 | 88.0 | 88.3 | 86.8 | 92.6 | 77.1 | 92.8 | 93.2 | 88.8 | 94.5 | 88.7 | 5.2 |
| | Preds, standardization | 81.6 | 88.8 | 85.2 | 88.9 | 74.1 | 88.1 | 87.7 | 87.4 | 88.7 | 83.1 | 92.2 | 94.3 | 91.3 | 84.8 | 96.0 | 88.0 | 89.7 | 92.4 | 87.9 | 4.9 |
| Entropy | Source Train | 76.9 | 83.2 | 90.4 | 90.8 | 81.4 | 87.5 | 77.8 | 83.1 | 68.2 | 83.5 | 67.5 | 82.5 | 87.8 | 79.7 | 90.7 | 53.4 | 74.8 | 82.8 | 80.1 | 9.3 |
| | Source Train + Target | 81.6 | 79.9 | 83.1 | 89.9 | 80.7 | 89.6 | 78.7 | 86.6 | 76.8 | 86.1 | 85.6 | 86.5 | 83.5 | 77.5 | 85.0 | 60.2 | 58.3 | 76.2 | 80.3 | 8.5 |
| | Source Val | 64.6 | 77.7 | 89.1 | 90.4 | 80.9 | 86.7 | 78.8 | 83.9 | 80.3 | 85.3 | 84.1 | 87.5 | 87.7 | 86.6 | 92.1 | 86.8 | 65.4 | 82.8 | 82.8 | 7.3 |
| | Source Val + Target | 81.3 | 77.6 | 83.5 | 89.2 | 80.3 | 88.7 | 75.0 | 82.0 | 76.5 | 85.1 | 85.6 | 84.9 | 83.0 | 76.8 | 83.6 | 67.3 | 56.1 | 75.3 | 79.5 | 7.8 |
| | Target | 78.9 | 74.3 | 76.7 | 87.0 | 76.7 | 88.0 | 72.0 | 80.2 | 72.5 | 81.1 | 81.8 | 79.8 | 78.1 | 70.2 | 76.2 | 55.3 | 52.9 | 67.8 | 75.0 | 9.0 |
| SND | Features, $\tau = 0.05$ | -76.5 | -69.7 | -69.5 | -83.0 | -71.3 | -84.1 | -70.1 | -75.2 | -81.9 | -64.3 | -83.7 | -76.5 | -61.6 | -73.9 | -75.6 | -80.1 | -73.7 | -80.9 | -75.1 | 6.4 |
| | Features, $\tau = 0.1$ | -76.8 | -70.5 | -70.1 | -82.8 | -71.5 | -83.8 | -72.1 | -75.3 | -80.9 | -65.6 | -83.8 | -77.6 | -62.3 | -74.8 | -76.0 | -75.3 | -74.4 | -80.6 | -75.2 | 5.8 |
| | Features, $\tau = 0.5$ | -79.2 | -75.0 | -77.5 | -85.0 | -77.3 | -87.3 | -80.3 | -82.3 | -86.2 | -73.5 | -88.1 | -84.5 | -72.8 | -82.5 | -83.1 | -82.7 | -79.4 | -85.8 | -81.3 | 4.5 |
| | Logits, $\tau = 0.05$ | -78.1 | -68.2 | -67.8 | -84.4 | -72.8 | -87.0 | -66.4 | -76.8 | -87.7 | -64.6 | -87.7 | -78.0 | -60.7 | -76.6 | -76.0 | -87.3 | -74.0 | -83.6 | -76.5 | 8.3 |
| | Logits, $\tau = 0.1$ | -78.6 | -68.7 | -67.3 | -83.5 | -72.0 | -86.2 | -66.9 | -75.8 | -85.9 | -64.6 | -86.9 | -77.7 | -61.1 | -75.9 | -75.9 | -84.6 | -74.3 | -82.1 | -76.0 | 7.7 |
| | Logits, $\tau = 0.5$ | -79.7 | -71.2 | -72.9 | -85.5 | -76.7 | -87.6 | -75.5 | -79.1 | -86.2 | -69.7 | -89.0 | -82.8 | -70.1 | -82.6 | -80.5 | -84.0 | -78.4 | -84.9 | -79.8 | 5.9 |
| | Preds, $\tau = 0.05$ | -78.2 | -67.4 | -69.7 | -84.8 | -72.7 | -88.4 | -63.3 | -78.8 | -89.4 | -65.7 | -88.1 | -80.3 | -62.3 | -76.6 | -76.5 | -89.4 | -74.1 | -86.4 | -77.3 | 8.9 |
| | Preds, $\tau = 0.1$ | -78.1 | -67.7 | -69.0 | -84.4 | -71.8 | -87.8 | -63.5 | -78.1 | -89.1 | -65.8 | -87.1 | -80.3 | -62.2 | -75.8 | -76.3 | -90.3 | -73.6 | -85.9 | -77.0 | 8.8 |
| | Preds, $\tau = 0.5$ | -79.4 | -69.4 | -72.7 | -85.1 | -74.4 | -87.9 | -70.4 | -79.8 | -88.1 | -67.9 | -88.0 | -83.2 | -64.6 | -80.9 | -79.5 | -85.8 | -75.8 | -86.1 | -78.8 | 7.3 |

Table 15: The weighted Spearman correlation of each validator/task pair for **BNM**.

| | | AD | AW | DA | DW | WA | WD | AC | AP | AR | CA | CP | CR | PA | PC | PR | RA | RC | RP | Mean | Std |
|---|---|---|---|---|---|---|---|---|---|---|---|---|---|---|---|---|---|---|---|---|---|
| Accuracy | Source Train | 75.5 | 78.3 | 90.2 | 72.4 | 80.6 | 51.1 | 65.0 | 71.6 | 64.8 | 54.2 | 57.9 | 60.8 | 73.2 | 70.1 | 70.6 | 57.9 | 71.1 | 62.4 | 68.2 | 9.6 |
| | Source Val | 82.2 | 89.8 | 84.3 | 78.2 | 85.6 | 61.3 | 97.4 | 97.6 | 97.8 | 85.4 | 87.6 | 90.6 | 93.6 | 93.0 | 94.7 | 96.1 | 97.2 | 96.2 | 89.4 | 9.0 |
| BNM | Source Train | 65.6 | 72.4 | 68.6 | 34.0 | 73.3 | -5.4 | 53.1 | 56.4 | 36.8 | 30.8 | 20.7 | 41.7 | 61.7 | 59.2 | 55.2 | 27.8 | 53.0 | 44.8 | 47.2 | 19.9 |
| | Source Train + Target | 65.6 | 80.3 | 80.1 | 42.0 | 85.5 | -34.2 | 68.0 | 76.2 | 68.6 | 50.2 | 52.8 | 67.5 | 63.4 | 73.0 | 74.0 | 45.4 | 74.3 | 67.9 | 61.1 | 26.0 |
| | Source Val | 65.8 | 73.2 | 72.1 | 54.9 | 80.0 | 10.7 | 79.3 | 80.4 | 75.2 | 30.8 | 29.9 | 48.6 | 74.7 | 75.6 | 73.3 | 63.4 | 81.2 | 75.0 | 63.6 | 20.0 |
| | Source Val + Target | 65.9 | 78.2 | 79.6 | 55.6 | 86.5 | -10.8 | 75.3 | 79.6 | 73.8 | 45.9 | 44.3 | 61.4 | 66.5 | 74.6 | 74.7 | 52.0 | 78.1 | 72.9 | 64.1 | 21.7 |
| | Target | 65.7 | 79.8 | 79.5 | 42.0 | 84.8 | -34.3 | 66.5 | 76.0 | 68.8 | 51.8 | 54.5 | 69.9 | 63.0 | 73.1 | 74.6 | 45.9 | 74.5 | 68.2 | 61.3 | 25.9 |
| ClassAMI | Source + Target Features | 68.3 | 85.5 | 90.9 | 58.8 | 89.8 | 18.5 | 84.9 | 87.2 | 87.5 | 66.5 | 73.9 | 85.0 | 81.2 | 89.9 | 90.1 | 73.3 | 92.3 | 85.5 | 78.3 | 17.3 |
| | Source + Target Logits | 67.6 | 84.7 | 88.9 | 58.2 | 89.4 | 19.4 | 81.8 | 85.3 | 86.0 | 64.2 | 72.4 | 82.7 | 78.7 | 89.4 | 89.2 | 72.9 | 91.5 | 84.5 | 77.0 | 16.8 |
| | Target Features | -16.4 | -32.8 | 5.2 | -14.3 | -9.7 | -12.7 | -43.5 | -54.3 | -39.7 | -35.3 | -41.3 | -42.4 | -25.8 | -32.4 | -41.6 | -20.6 | -41.2 | -52.3 | -30.6 | 15.7 |
| | Target Logits | -12.4 | -35.2 | 11.5 | -16.4 | -8.0 | -6.5 | -48.5 | -48.4 | -35.1 | -33.4 | -42.4 | -42.2 | -29.6 | -36.6 | -34.9 | -19.9 | -43.4 | -50.3 | -29.5 | 16.8 |
| ClassSS | Source + Target Features | 26.8 | 42.8 | 61.2 | 39.5 | 59.8 | -8.0 | -19.4 | 7.7 | 16.4 | -10.9 | 1.2 | 8.3 | -6.7 | 17.2 | 19.4 | 4.2 | 5.4 | 16.6 | 15.6 | 22.4 |
| | Source + Target Logits | 7.8 | 43.4 | 49.0 | 35.7 | 55.8 | -10.1 | -12.0 | 5.4 | 12.7 | -5.7 | 8.4 | 19.2 | -3.7 | 5.1 | 9.9 | 4.1 | -0.4 | 10.5 | 13.1 | 19.5 |
| | Target Features | 43.4 | 61.8 | 59.8 | 46.6 | 55.7 | 3.0 | -49.6 | -3.1 | 10.4 | -17.2 | -0.1 | 4.3 | -24.6 | -17.9 | 5.6 | -10.1 | -23.6 | 14.3 | 8.8 | 31.5 |
| | Target Logits | 21.5 | 68.0 | 47.1 | 52.2 | 51.4 | 0.4 | -41.6 | -4.3 | 8.5 | -20.2 | 6.5 | 13.0 | -21.6 | -10.3 | -0.5 | -6.8 | -15.9 | 10.7 | 8.8 | 28.6 |
| DEV | Features | -33.0 | -33.3 | 15.4 | 32.4 | 51.3 | 25.6 | 29.4 | 40.2 | 24.7 | -16.6 | -8.3 | 3.3 | -4.9 | 7.9 | -14.2 | -5.9 | -0.9 | -8.3 | 5.8 | 23.6 |
| | Logits | 20.3 | 25.9 | 63.3 | 47.0 | 72.8 | 34.7 | 51.7 | 64.4 | 78.4 | 23.1 | 22.5 | 44.1 | 19.3 | 35.7 | 38.4 | 23.0 | 51.4 | 63.7 | 43.3 | 18.8 |
| | Preds | -24.1 | -38.6 | -42.7 | -36.7 | -47.4 | -45.8 | -29.0 | -10.0 | -25.5 | -30.4 | 65.3 | 78.9 | -35.3 | 66.0 | 75.1 | -53.5 | 72.3 | 79.1 | 1.0 | 51.7 |
| DEVN | Features, max normalization | 77.5 | 85.5 | 77.3 | 54.0 | 79.3 | 38.1 | 96.6 | 97.1 | 96.1 | 80.7 | 76.9 | 89.1 | 92.7 | 90.9 | 93.9 | 94.5 | 96.0 | 95.1 | 85.1 | 15.4 |
| | Features, standardization | 79.0 | 58.1 | 78.4 | 46.6 | 77.4 | 32.3 | 91.1 | 93.7 | 89.7 | 83.8 | 79.9 | 85.6 | 89.7 | 82.8 | 86.4 | 89.0 | 90.2 | 87.9 | 79.0 | 16.2 |
| | Logits, max normalization | 74.2 | 86.9 | 72.9 | 54.3 | 78.8 | 38.5 | 96.8 | 97.1 | 95.9 | 78.9 | 75.4 | 88.5 | 91.9 | 89.8 | 93.5 | 94.4 | 95.7 | 94.7 | 83.2 | 15.5 |
| | Logits, standardization | 64.7 | 73.7 | 73.4 | 48.0 | 81.7 | 33.0 | 92.8 | 93.6 | 91.4 | 79.7 | 74.5 | 81.4 | 87.7 | 80.8 | 86.7 | 87.5 | 90.2 | 86.2 | 78.2 | 15.5 |
| | Preds, max normalization | 72.7 | 87.4 | 70.8 | 54.1 | 75.8 | 38.2 | 96.5 | 97.0 | 96.0 | 76.2 | 73.5 | 87.6 | 91.0 | 90.0 | 92.8 | 93.6 | 96.2 | 94.4 | 82.4 | 15.8 |
| | Preds, standardization | 65.9 | 78.1 | 63.6 | 41.1 | 65.3 | 30.0 | 90.1 | 92.2 | 90.5 | 62.0 | 71.1 | 84.8 | 83.2 | 79.6 | 89.2 | 86.2 | 87.9 | 90.5 | 75.1 | 17.3 |
| Entropy | Source Train | 62.4 | 71.7 | 62.3 | 27.3 | 69.9 | -9.1 | 49.5 | 58.2 | 33.1 | 24.1 | 19.3 | 39.4 | 52.3 | 59.1 | 53.1 | 23.5 | 51.0 | 40.7 | 43.8 | 20.3 |
| | Source Train + Target | 69.0 | 80.7 | 74.7 | 58.6 | 80.6 | 16.3 | 58.4 | 73.8 | 70.4 | 43.8 | 50.1 | 64.5 | 51.5 | 64.6 | 73.7 | 54.6 | 69.4 | 66.6 | 62.3 | 15.1 |
| | Source Val | 42.1 | 64.5 | 67.6 | 47.5 | 75.8 | 7.1 | 55.6 | 62.8 | 72.1 | 16.6 | 26.2 | 43.1 | 56.4 | 65.4 | 69.0 | 59.0 | 71.0 | 64.0 | 53.7 | 19.2 |
| | Source Val + Target | 61.5 | 75.6 | 75.1 | 55.4 | 81.4 | 10.7 | 55.9 | 68.7 | 71.5 | 37.9 | 44.3 | 59.9 | 51.6 | 64.8 | 73.1 | 56.2 | 69.3 | 65.2 | 59.9 | 16.2 |
| | Target | 68.4 | 79.3 | 68.5 | 58.9 | 78.1 | 15.9 | 55.9 | 71.0 | 70.6 | 45.3 | 52.2 | 66.9 | 49.1 | 64.4 | 73.2 | 55.9 | 68.1 | 66.3 | 61.7 | 14.1 |
| SND | Features, $\tau = 0.05$ | -77.9 | -85.6 | -83.9 | -88.3 | -86.5 | -66.3 | -82.0 | -82.6 | -92.5 | -80.7 | -89.7 | -84.5 | -80.5 | -82.8 | -87.6 | -90.4 | -81.7 | -84.2 | -83.8 | 5.6 |
| | Features, $\tau = 0.1$ | -78.0 | -84.7 | -80.5 | -87.7 | -82.8 | -81.6 | -79.5 | -81.0 | -87.2 | -76.4 | -85.4 | -82.2 | -78.2 | -78.0 | -85.1 | -83.7 | -77.8 | -82.7 | -81.8 | 3.3 |
| | Features, $\tau = 0.5$ | -80.4 | -86.0 | -81.3 | -73.7 | -82.9 | -58.8 | -83.4 | -85.3 | -83.1 | -85.4 | -86.2 | -85.7 | -86.2 | -81.5 | -86.3 | -77.8 | -81.5 | -83.3 | -81.6 | 6.4 |
| | Logits, $\tau = 0.05$ | -78.8 | -87.3 | -86.7 | -86.3 | -89.4 | -75.0 | -82.9 | -85.2 | -96.2 | -86.4 | -94.5 | -87.7 | -85.9 | -86.8 | -90.2 | -92.2 | -84.0 | -85.1 | -86.7 | 4.9 |
| | Logits, $\tau = 0.1$ | -80.1 | -88.2 | -83.4 | -90.0 | -87.3 | -89.3 | -82.2 | -84.4 | -94.3 | -85.0 | -93.5 | -86.8 | -84.0 | -82.7 | -88.9 | -89.9 | -82.1 | -84.9 | -86.5 | 3.9 |
| | Logits, $\tau = 0.5$ | -78.9 | -85.3 | -77.9 | -68.0 | -78.5 | -50.7 | -83.3 | -82.5 | -87.0 | -84.9 | -87.5 | -83.8 | -85.9 | -84.5 | -85.6 | -84.4 | -82.1 | -83.6 | -80.8 | 8.5 |
| | Preds, $\tau = 0.05$ | -79.7 | -89.1 | -87.8 | -83.6 | -89.6 | -63.1 | -82.4 | -84.3 | -91.4 | -85.6 | -90.6 | -87.0 | -86.9 | -86.6 | -88.2 | -88.2 | -83.6 | -83.5 | -85.1 | 6.1 |
| | Preds, $\tau = 0.1$ | -80.2 | -89.6 | -89.5 | -96.1 | -90.4 | -91.0 | -82.9 | -84.5 | -96.2 | -86.1 | -94.5 | -87.6 | -87.3 | -88.2 | -91.5 | -92.3 | -84.3 | -85.3 | -88.8 | 4.4 |
| | Preds, $\tau = 0.5$ | -80.5 | -86.8 | -68.8 | -84.7 | -66.7 | -86.5 | -79.2 | -83.8 | -92.1 | -84.3 | -88.7 | -83.9 | -77.8 | -68.3 | -84.0 | -89.7 | -78.8 | -84.6 | -81.6 | 7.1 |

Table 16: The weighted Spearman correlation of each validator/task pair for **BSP**.

| | | AD | AW | DA | DW | WA | WD | AC | AP | AR | CA | CP | CR | PA | PC | PR | RA | RC | RP | Mean | Std |
|---|---|---|---|---|---|---|---|---|---|---|---|---|---|---|---|---|---|---|---|---|---|
| Accuracy | Source Train | 71.3 | 61.5 | **75.7** | 80.0 | **66.1** | 81.9 | 33.5 | 34.6 | 46.5 | 34.5 | 37.9 | 45.8 | 70.1 | 46.1 | 64.2 | 56.2 | 55.8 | 58.5 | 56.7 | 15.5 |
| | Source Val | **83.3** | **84.0** | 63.2 | 81.9 | 65.0 | **85.1** | **95.3** | 96.1 | 95.7 | 59.4 | 76.6 | 76.8 | 87.2 | 86.4 | 91.9 | 95.9 | 87.7 | 93.8 | **83.6** | 11.2 |
| BNM | Source Train | 33.0 | 11.1 | 22.4 | 27.9 | 1.7 | 23.3 | 29.2 | 34.7 | 36.3 | 51.2 | 54.3 | 53.1 | 52.5 | 36.6 | 50.9 | 58.4 | 48.0 | 50.0 | 37.5 | 15.7 |
| | Source Train + Target | 16.8 | -1.8 | 8.2 | 21.9 | -4.8 | 22.6 | 24.7 | 33.4 | 41.1 | 53.2 | 56.6 | 54.7 | 47.2 | 25.1 | 49.2 | 57.1 | 39.3 | 46.7 | 32.8 | 19.2 |
| | Source Val | 24.7 | 0.7 | 20.1 | 23.9 | -0.3 | 19.9 | 27.2 | 33.1 | 43.0 | 50.5 | 53.5 | 50.0 | 52.2 | 36.8 | 50.7 | 62.2 | 42.8 | 49.9 | 35.6 | 17.6 |
| | Source Val + Target | 16.9 | -2.7 | 13.4 | 22.2 | -3.7 | 20.8 | 23.6 | 31.4 | 41.5 | 51.9 | 54.5 | 51.9 | 48.5 | 28.5 | 49.2 | 58.3 | 37.8 | 46.7 | 32.8 | 18.6 |
| | Target | 9.7 | -3.6 | -20.0 | 17.0 | -13.9 | 21.5 | 14.2 | 26.5 | 38.6 | 51.8 | 52.3 | 51.4 | 39.6 | 16.8 | 46.1 | 53.2 | 27.2 | 41.0 | 26.1 | 22.2 |
| ClassAMI | Source + Target Features | 68.8 | 36.6 | 6.0 | 44.9 | 10.4 | 67.6 | 27.4 | 36.7 | 53.0 | 60.8 | 43.8 | 63.8 | 79.3 | 5.6 | 76.0 | 77.8 | 26.0 | 62.2 | 47.0 | 23.9 |
| | Source + Target Logits | 71.9 | 31.7 | 15.8 | 50.3 | 14.1 | 71.0 | 25.3 | 40.2 | 56.1 | 59.9 | 49.3 | 67.6 | 82.8 | 5.2 | 77.7 | 78.3 | 29.2 | 61.6 | 49.3 | 23.7 |
| | Target Features | -69.5 | -65.5 | -65.5 | -47.5 | -57.3 | -40.4 | -67.4 | -51.9 | -34.6 | -61.9 | -70.0 | -61.7 | -54.5 | -64.3 | -29.1 | -54.7 | -45.7 | -49.2 | -55.0 | 11.7 |
| | Target Logits | -65.4 | -64.6 | -65.9 | -51.2 | -59.1 | -40.1 | -81.7 | -58.7 | -39.8 | -68.8 | -70.9 | -64.6 | -61.5 | -76.0 | -40.1 | -57.7 | -67.0 | -60.7 | -60.8 | 11.4 |
| ClassSS | Source + Target Features | -64.9 | -27.2 | -46.3 | 1.2 | -56.4 | -15.2 | -79.2 | -80.8 | -76.0 | -72.6 | -74.3 | -75.8 | -66.8 | -78.2 | -73.1 | -70.9 | -70.7 | -73.4 | -61.2 | 23.2 |
| | Source + Target Logits | -44.3 | -25.0 | -40.8 | -10.4 | -43.5 | 7.0 | -76.9 | -74.7 | -57.5 | -67.5 | -68.7 | -64.8 | -63.3 | -76.1 | -67.7 | -59.2 | -67.5 | -62.5 | -53.5 | 22.8 |
| | Target Features | -48.5 | -20.1 | -47.3 | -4.8 | -56.9 | -3.7 | -80.3 | -80.0 | -73.5 | -78.7 | -69.6 | -74.1 | -75.0 | -83.0 | -70.1 | -69.4 | -78.6 | -71.9 | -60.3 | 24.9 |
| | Target Logits | -51.2 | -14.0 | -45.4 | 1.5 | -50.6 | 11.0 | -82.4 | -76.5 | -62.3 | -79.7 | -65.3 | -69.1 | -76.3 | -82.5 | -67.2 | -64.6 | -78.9 | -59.6 | -56.3 | 27.4 |
| DEV | Features | -62.9 | -47.2 | -19.8 | -15.0 | -31.4 | -60.3 | -31.3 | -29.0 | -29.7 | -63.8 | -45.9 | -41.3 | -59.9 | -45.1 | -45.7 | -59.9 | -25.5 | -43.9 | -42.1 | 14.9 |
| | Logits | -37.6 | -24.3 | -5.5 | -5.2 | -8.8 | -62.3 | -6.5 | -14.9 | -19.5 | -18.5 | 14.2 | 22.2 | -29.8 | 6.4 | 2.0 | -16.6 | 18.8 | 6.6 | -10.0 | 20.4 |
| | Preds | -42.8 | -39.7 | -29.0 | -37.9 | -35.7 | -60.6 | -28.4 | -10.9 | -25.3 | -35.3 | -1.0 | -2.4 | -44.2 | -23.4 | -33.1 | -43.1 | -15.4 | -10.6 | -28.8 | 15.5 |
| DEVN | Features, max normalization | 82.2 | 77.0 | 43.5 | 63.9 | 43.1 | 79.9 | 94.9 | **96.6** | **96.7** | 72.1 | **83.1** | **82.8** | **91.4** | **87.7** | **94.9** | **97.0** | 88.2 | **94.4** | 81.6 | 16.2 |
| | Features, standardization | 52.4 | 51.6 | 39.9 | 50.4 | 43.6 | 70.3 | 86.9 | 89.9 | 93.7 | **75.6** | 78.9 | 79.6 | 90.4 | 79.9 | 93.0 | 95.0 | 81.7 | 90.8 | 74.6 | 18.1 |
| | Logits, max normalization | 82.6 | 75.6 | 47.8 | 68.0 | 45.0 | 79.9 | 93.4 | 94.3 | 95.4 | 68.5 | 78.6 | 79.3 | 90.0 | 84.4 | 92.6 | **96.8** | **89.6** | **96.1** | 81.0 | 15.1 |
| | Logits, standardization | 67.5 | 53.6 | 52.8 | 56.4 | 52.3 | 72.7 | 82.5 | 84.3 | 91.5 | 70.2 | 67.8 | 71.0 | 87.5 | 73.8 | 89.1 | 94.0 | 85.5 | 93.2 | 74.8 | 14.0 |
| | Preds, max normalization | 82.2 | 76.5 | 42.2 | 68.0 | 41.0 | 79.1 | 92.5 | 90.1 | 92.2 | 64.2 | 77.5 | 74.9 | **89.4** | 82.4 | 90.9 | 94.6 | 78.1 | 89.4 | 78.1 | 15.4 |
| | Preds, standardization | 69.4 | 60.5 | 42.8 | 59.7 | 44.5 | 76.4 | 71.7 | 71.1 | 81.8 | 60.2 | 71.7 | 77.8 | 88.0 | 79.6 | 87.1 | 82.9 | 64.5 | 77.5 | 70.4 | 12.7 |
| Entropy | Source Train | 18.9 | 9.2 | 14.8 | 24.1 | 3.0 | 17.1 | 28.2 | 37.6 | 37.9 | 51.0 | 54.5 | 52.2 | 50.9 | 35.0 | 49.4 | 56.6 | 45.4 | 47.8 | 35.2 | 16.6 |
| | Source Train + Target | 8.9 | -4.4 | -13.5 | 14.7 | -14.4 | 14.7 | 22.2 | 32.9 | 38.8 | 50.9 | 53.4 | 51.4 | 44.7 | 21.1 | 45.2 | 52.5 | 33.3 | 40.5 | 27.4 | 21.8 |
| | Source Val | 3.8 | -4.2 | 3.5 | 17.7 | -2.9 | 11.5 | 20.1 | 29.1 | 38.7 | 48.8 | 51.1 | 47.2 | 48.1 | 32.4 | 46.8 | 55.8 | 34.4 | 41.0 | 29.1 | 19.4 |
| | Source Val + Target | 2.7 | -8.3 | -15.8 | 13.7 | -16.3 | 12.4 | 17.4 | 27.3 | 38.2 | 48.8 | 50.0 | 47.8 | 42.2 | 20.3 | 43.5 | 50.4 | 26.9 | 36.1 | 24.3 | 21.9 |
| | Target | 2.9 | -9.2 | -18.2 | 10.1 | -23.3 | 13.1 | 19.8 | 28.1 | 40.1 | 47.2 | 47.3 | 46.0 | 37.5 | 15.0 | 40.6 | 46.0 | 19.8 | 31.6 | 21.9 | 22.0 |
| SND | Features, $\tau = 0.05$ | -79.6 | -84.4 | -72.6 | -88.7 | -75.0 | -79.9 | -63.8 | -73.8 | -81.6 | -70.6 | -73.2 | -77.1 | -77.2 | -75.0 | -63.0 | -74.5 | -79.2 | -75.6 | 6.2 |
| | Features, $\tau = 0.1$ | -78.3 | -79.6 | -69.7 | -87.4 | -72.7 | -79.3 | -64.9 | -72.9 | -79.6 | -69.0 | -72.3 | -75.3 | -70.5 | -75.1 | -74.8 | -61.2 | -72.0 | -78.1 | -74.0 | 5.9 |
| | Features, $\tau = 0.5$ | -76.0 | -77.0 | -73.1 | -82.5 | -73.7 | -76.3 | -69.9 | -74.8 | -77.5 | -75.1 | -74.8 | -75.7 | -75.5 | -77.4 | -75.1 | -61.0 | -71.8 | -76.0 | -74.6 | 4.2 |
| | Logits, $\tau = 0.05$ | -76.9 | -70.3 | -71.0 | -79.3 | -73.2 | -78.8 | -68.9 | -80.9 | -83.9 | -72.2 | -78.2 | -82.4 | -73.8 | -79.3 | -76.3 | -63.9 | -77.0 | -81.2 | -76.0 | 5.1 |
| | Logits, $\tau = 0.1$ | -75.6 | -67.7 | -71.8 | -76.2 | -72.9 | -77.1 | -71.2 | -78.8 | -84.4 | -73.0 | -77.3 | -81.2 | -72.9 | -79.7 | -77.0 | -62.8 | -77.4 | -78.2 | -75.3 | 4.9 |
| | Logits, $\tau = 0.5$ | -76.0 | -66.8 | -76.0 | -75.6 | -77.2 | -79.4 | -74.3 | -79.8 | -86.6 | -76.7 | -79.8 | -81.7 | -78.4 | -81.0 | -78.7 | -67.3 | -78.7 | -79.8 | -77.4 | 4.6 |
| | Preds, $\tau = 0.05$ | -77.9 | -74.2 | -81.2 | -85.1 | -80.8 | -85.7 | -81.5 | -85.7 | -87.0 | -74.6 | -80.4 | -79.6 | -78.0 | -78.4 | -79.0 | -65.1 | -82.8 | -83.0 | -80.2 | 4.4 |
| | Preds, $\tau = 0.1$ | -77.9 | -73.4 | -79.7 | -84.3 | -78.2 | -85.9 | -85.9 | -82.9 | -85.8 | -74.4 | -80.3 | -79.1 | -78.2 | -78.0 | -80.0 | -72.0 | -82.1 | -81.6 | -80.0 | **4.0** |
| | Preds, $\tau = 0.5$ | -77.4 | -73.0 | -72.8 | -83.7 | -72.8 | -86.3 | -81.0 | -79.9 | -83.0 | -74.5 | -80.3 | -80.4 | -77.9 | -73.8 | -82.1 | -76.8 | -78.5 | -81.0 | -78.6 | **4.0** |

Table 17: The weighted Spearman correlation of each validator/task pair for **CDAN**.

| | | AD | AW | DA | DW | WA | WD | AC | AP | AR | CA | CP | CR | PA | PC | PR | RA | RC | RP | Mean | Std |
|---|---|---|---|---|---|---|---|---|---|---|---|---|---|---|---|---|---|---|---|---|---|
| Accuracy | Source Train | 74.6 | 79.5 | **88.5** | 69.1 | 85.2 | 44.8 | 65.8 | 73.0 | 71.6 | 49.0 | 71.7 | 71.3 | 73.2 | 78.0 | 83.0 | 63.8 | 76.6 | 72.2 | 71.7 | 10.7 |
| | Source Val | 85.1 | 92.2 | 77.6 | 75.2 | 86.2 | 61.1 | 96.4 | 97.7 | 94.2 | 77.5 | 93.0 | 91.2 | 95.2 | 95.9 | 97.0 | 95.0 | 95.5 | 95.9 | 89.0 | 9.8 |
| BNM | Source Train | 51.6 | 68.9 | 61.9 | 30.6 | 70.1 | 34.5 | 57.6 | 65.0 | 58.2 | 38.2 | 59.8 | 61.3 | 63.7 | 76.2 | 77.3 | 36.3 | 62.8 | 58.9 | 59.7 | 13.5 |
| | Source Train + Target | 38.0 | 54.0 | 74.7 | 53.9 | 81.3 | 12.3 | 77.8 | 73.0 | 56.4 | 39.8 | 64.7 | 65.5 | 57.4 | 80.3 | 73.7 | 41.8 | 78.1 | 62.6 | 60.3 | 17.8 |
| | Source Val | 60.8 | 79.0 | 60.0 | 65.3 | 73.2 | 41.9 | 89.9 | 87.8 | 77.0 | 40.2 | 63.0 | 64.2 | 75.1 | 89.5 | 85.3 | 71.4 | 91.5 | 85.5 | 72.2 | 15.1 |
| | Source Val + Target | 51.7 | 72.9 | 70.7 | 69.2 | 79.9 | 37.0 | 87.1 | 84.7 | 73.3 | 40.1 | 65.4 | 65.7 | 60.7 | 84.5 | 79.3 | 53.3 | 86.8 | 79.6 | 69.0 | 14.9 |
| | Target | 37.2 | 52.2 | 74.5 | 54.7 | 81.0 | 11.0 | 77.9 | 72.6 | 56.1 | 39.9 | 64.8 | 65.8 | 57.3 | 80.2 | 73.5 | 41.8 | 78.1 | 62.5 | 60.1 | 18.1 |
| ClassAMI | Source + Target Features | 32.7 | 46.8 | 69.2 | 26.3 | 79.5 | 31.2 | 76.2 | 74.3 | 45.6 | 48.0 | 63.9 | 68.9 | 65.9 | 82.5 | 77.8 | 57.4 | 87.8 | 70.1 | 61.3 | 18.2 |
| | Source + Target Logits | 28.8 | 43.7 | 66.3 | 24.7 | 75.4 | 29.7 | 73.8 | 72.2 | 42.3 | 43.7 | 61.2 | 65.5 | 61.6 | 81.5 | 76.3 | 55.7 | 86.4 | 68.5 | 58.7 | 18.4 |
| | Target Features | -32.3 | -41.5 | -23.1 | -47.9 | -26.2 | -37.5 | -63.4 | -13.1 | -56.1 | -63.8 | -59.6 | -60.1 | -55.3 | -48.3 | -24.2 | -53.1 | -34.0 | -57.3 | -44.3 | 15.3 |
| | Target Logits | -31.7 | -38.9 | -19.2 | -48.6 | -24.4 | -32.1 | -65.0 | -8.6 | -52.4 | -58.1 | -58.5 | -48.8 | -50.7 | -18.1 | -51.8 | -34.9 | -55.8 | -42.3 | 16.4 |
| ClassSS | Source + Target Features | 19.3 | 17.6 | 43.3 | -11.9 | 48.5 | -13.2 | 27.1 | 32.0 | 15.6 | -7.0 | 21.4 | 20.1 | -8.6 | 20.1 | 31.0 | -2.8 | 45.3 | 22.1 | 17.8 | 18.9 |
| | Source + Target Logits | 13.8 | -1.5 | 32.8 | -26.4 | 41.1 | -15.5 | 24.1 | 32.1 | 9.0 | -2.3 | 21.0 | 18.0 | -1.0 | 28.5 | 35.4 | 15.7 | 50.6 | 26.0 | 16.7 | 19.5 |
| | Target Features | 23.9 | 27.3 | 42.4 | -6.6 | 46.5 | -6.0 | 23.5 | 28.5 | 7.5 | -20.2 | 14.4 | 13.5 | -13.2 | 17.0 | 26.3 | -1.7 | 42.7 | 19.3 | 15.8 | 18.9 |
| | Target Logits | 19.7 | 26.3 | 30.7 | -5.1 | 37.2 | -9.5 | 30.0 | 29.3 | 3.5 | -15.3 | 15.6 | 11.2 | -15.6 | 26.7 | 27.4 | 10.1 | 49.8 | 21.3 | 16.3 | 18.0 |
| DEV | Features | -64.9 | -82.0 | -16.2 | 17.3 | 47.0 | 7.6 | 42.7 | 48.2 | 28.3 | -8.8 | -15.8 | -11.5 | -6.8 | 6.3 | 16.5 | -21.2 | 4.6 | 10.6 | 0.1 | 33.4 |
| | Logits | -20.4 | -20.9 | 26.8 | 39.5 | 64.1 | 29.2 | 66.5 | 72.3 | 74.9 | 26.4 | 32.8 | 45.1 | 22.9 | 41.0 | 44.2 | 7.7 | 40.1 | 49.3 | 35.6 | 26.4 |
| | Preds | -48.5 | -55.9 | -49.1 | -35.8 | -46.7 | -44.3 | -4.6 | -29.3 | -29.2 | -29.8 | 61.7 | 67.8 | -50.3 | 56.8 | 69.7 | -52.0 | 76.9 | 74.2 | -3.8 | 52.1 |
| DEVN | Features, max normalization | 69.1 | 80.8 | 55.0 | 50.8 | 71.3 | 49.7 | 93.0 | 96.7 | 88.6 | 70.7 | 89.1 | 85.0 | 93.8 | 91.5 | 94.5 | 90.0 | 92.9 | 90.7 | 80.7 | 15.3 |
| | Features, standardization | 73.7 | 80.7 | 61.2 | 29.1 | 76.6 | 50.4 | 86.7 | 92.3 | 84.1 | 75.0 | 78.2 | 77.7 | 92.3 | 81.8 | 85.7 | 85.7 | 86.4 | 88.5 | 77.0 | 15.4 |
| | Logits, max normalization | 64.5 | 79.2 | 52.0 | 51.7 | 68.1 | 47.6 | 93.0 | 96.2 | 87.7 | 68.3 | 88.0 | 84.8 | 92.4 | 89.7 | 92.5 | 87.4 | 89.3 | 91.2 | 79.1 | 15.6 |
| | Logits, standardization | 53.7 | 64.0 | 54.7 | 44.6 | 68.7 | 38.9 | 87.4 | 90.8 | 79.6 | 73.6 | 71.4 | 75.9 | 87.3 | 76.9 | 84.2 | 79.8 | 81.8 | 87.5 | 72.3 | 14.9 |
| | Preds, max normalization | 63.8 | 79.1 | 50.2 | 51.1 | 65.6 | 47.3 | 93.2 | 96.6 | 88.6 | 59.0 | 73.7 | 71.2 | 91.1 | 93.2 | 92.7 | 88.1 | 94.3 | 90.4 | 77.2 | 16.7 |
| | Preds, standardization | 54.1 | 71.0 | 48.1 | 37.0 | 57.1 | 36.0 | 85.6 | 91.6 | 82.0 | 47.4 | 59.0 | 59.6 | 80.1 | 86.0 | 82.8 | 76.9 | 90.0 | 81.7 | 68.1 | 17.9 |
| Entropy | Source Train | 47.8 | 55.3 | 53.5 | 18.8 | 57.3 | 30.9 | 59.8 | 63.9 | 55.7 | 38.2 | 55.6 | 61.9 | 49.7 | 74.8 | 71.9 | 32.1 | 63.7 | 51.9 | 52.4 | 14.1 |
| | Source Train + Target | 36.8 | 44.1 | 57.7 | 12.2 | 58.0 | 30.9 | 75.8 | 68.6 | 45.9 | 38.9 | 57.4 | 60.4 | 44.7 | 76.4 | 68.1 | 42.2 | 77.1 | 59.3 | 53.0 | 17.0 |
| | Source Val | 61.2 | 63.0 | 46.4 | 25.6 | 49.2 | 26.8 | 85.2 | 79.8 | 65.4 | 40.9 | 61.2 | 62.1 | 59.6 | 85.1 | 78.7 | 67.2 | 87.6 | 78.2 | 62.4 | 18.3 |
| | Source Val + Target | 55.9 | 59.2 | 55.3 | 17.2 | 56.6 | 29.6 | 80.3 | 74.8 | 57.6 | 39.6 | 59.9 | 60.4 | 46.4 | 78.6 | 71.3 | 50.2 | 81.8 | 70.8 | 58.1 | 16.9 |
| | Target | 34.9 | 41.2 | 55.2 | 10.8 | 56.8 | 29.1 | 75.5 | 68.2 | 45.5 | 38.8 | 57.3 | 59.1 | 43.2 | 75.8 | 67.4 | 42.0 | 76.8 | 58.9 | 52.0 | 17.3 |
| SND | Features, $\tau = 0.05$ | -94.5 | -92.8 | -85.4 | -92.2 | -87.5 | -89.1 | -94.4 | -92.8 | -95.6 | -88.6 | -92.4 | -92.9 | -86.8 | -89.8 | -92.4 | -93.0 | -92.5 | -93.0 | -91.4 | 2.8 |
| | Features, $\tau = 0.1$ | -93.6 | -90.6 | -80.0 | -87.4 | -81.6 | -89.8 | -88.5 | -79.6 | -91.3 | -84.3 | -87.6 | -88.2 | -79.3 | -79.4 | -80.2 | -86.1 | -79.6 | -86.8 | -85.2 | 4.6 |
| | Features, $\tau = 0.5$ | -79.7 | -84.1 | -79.7 | -68.3 | -80.1 | -75.0 | -81.2 | -64.9 | -78.4 | -70.3 | -76.7 | -75.6 | -70.9 | -73.8 | -64.4 | -68.8 | -67.6 | -76.6 | -74.2 | 5.7 |
| | Logits, $\tau = 0.05$ | -92.0 | -91.5 | -89.9 | -93.0 | -91.5 | -89.2 | -97.2 | -97.0 | -97.3 | -90.0 | -94.5 | -95.1 | -88.7 | -92.2 | -96.5 | -95.3 | -97.2 | -94.9 | -93.5 | 2.9 |
| | Logits, $\tau = 0.1$ | -95.3 | -92.9 | -84.3 | -92.6 | -87.3 | -93.1 | -95.1 | -93.7 | -96.6 | -89.3 | -93.7 | -94.3 | -86.3 | -85.8 | -93.3 | -92.8 | -92.5 | -93.8 | -91.8 | 3.5 |
| | Logits, $\tau = 0.5$ | -82.1 | -83.2 | -75.5 | -69.5 | -73.7 | -72.5 | -89.1 | -77.5 | -89.3 | -86.0 | -82.6 | -83.1 | -84.2 | -80.1 | -76.7 | -87.7 | -79.2 | -84.0 | -80.9 | 5.6 |
| | Preds, $\tau = 0.05$ | -89.6 | -89.5 | -91.8 | -89.2 | -91.5 | -81.5 | -93.1 | -92.5 | -95.0 | -89.2 | -91.3 | -93.1 | -87.8 | -89.1 | -93.2 | -92.5 | -93.6 | -92.2 | -90.9 | 3.0 |
| | Preds, $\tau = 0.1$ | -95.8 | -93.9 | -91.3 | -97.3 | -91.8 | -95.3 | -97.4 | -97.4 | -97.8 | -90.1 | -94.9 | -95.6 | -89.2 | -94.4 | -97.9 | -96.0 | -98.0 | -95.0 | -94.9 | **2.7** |
| | Preds, $\tau = 0.5$ | -93.6 | -93.3 | -82.2 | -89.3 | -84.4 | -90.1 | -88.6 | -87.2 | -91.0 | -87.6 | -90.5 | -90.9 | -81.9 | -82.8 | -85.9 | -91.0 | -84.5 | -90.1 | -88.1 | 3.6 |

Table 18: The weighted Spearman correlation of each validator/task pair for **DANN**.

| | | AD | AW | DA | DW | WA | WD | AC | AP | AR | CA | CP | CR | PA | PC | PR | RA | RC | RP | Mean | Std |
|---|---|---|---|---|---|---|---|---|---|---|---|---|---|---|---|---|---|---|---|---|---|
| Accuracy | Source Train | 77.3 | 80.8 | 88.2 | 69.0 | 82.2 | 45.4 | 67.2 | 71.4 | 70.2 | 46.9 | 68.5 | 57.0 | 65.3 | 76.6 | 76.9 | 57.3 | 75.1 | 67.5 | 69.1 | 11.2 |
| | Source Val | 88.2 | 93.7 | 77.7 | 76.5 | 82.7 | 68.0 | 94.4 | 97.1 | 97.0 | 80.6 | 91.7 | 85.1 | 92.5 | 93.0 | 95.0 | 95.3 | 88.7 | 94.3 | 88.4 | 8.0 |
| BNM | Source Train | 66.3 | 75.6 | 66.6 | 28.6 | 66.7 | 21.7 | 57.8 | 66.4 | 53.0 | 43.3 | 64.6 | 47.4 | 58.3 | 71.0 | 73.6 | 37.3 | 71.9 | 46.8 | 56.5 | 15.4 |
| | Source Train + Target | 40.4 | 58.2 | 72.8 | 41.4 | 76.8 | 3.4 | 79.5 | 63.7 | 56.3 | 42.1 | 65.5 | 41.1 | 49.0 | 78.9 | 63.0 | 39.6 | 79.6 | 44.9 | 55.3 | 19.2 |
| | Source Val | 74.3 | 83.9 | 71.7 | 70.3 | 71.0 | 57.8 | 93.9 | 87.5 | 84.7 | 49.6 | 71.3 | 54.7 | 77.7 | 85.9 | 85.7 | 73.1 | 89.3 | 84.5 | 75.9 | 12.1 |
| | Source Val + Target | 58.2 | 78.2 | 75.0 | 71.7 | 78.7 | 46.8 | 92.4 | 82.3 | 80.3 | 44.6 | 69.9 | 48.6 | 54.6 | 84.9 | 74.7 | 55.7 | 90.0 | 75.4 | 70.1 | 14.6 |
| | Target | 37.9 | 54.3 | 71.9 | 40.4 | 76.6 | 2.4 | 79.5 | 63.2 | 56.2 | 42.0 | 65.3 | 39.8 | 48.7 | 78.7 | 62.6 | 39.5 | 79.4 | 44.7 | 54.6 | 19.4 |
| ClassAMI | Source + Target Features | 51.3 | 57.2 | 82.0 | 19.1 | 83.9 | 44.5 | 89.7 | 78.4 | 74.7 | 83.4 | 78.0 | 73.6 | 82.0 | 90.7 | 84.6 | 77.9 | 92.6 | 72.3 | 73.1 | 18.3 |
| | Source + Target Logits | 33.3 | 52.1 | 77.9 | 11.7 | 79.1 | 37.2 | 87.4 | 66.3 | 60.7 | 76.1 | 66.4 | 63.4 | 73.9 | 88.1 | 78.5 | 68.1 | 90.3 | 60.7 | 65.1 | 20.1 |
| | Target Features | -27.7 | -54.6 | -2.5 | -23.3 | -0.1 | -13.4 | -17.1 | -31.3 | -5.4 | -27.5 | -35.6 | -28.6 | 1.6 | 1.8 | 1.3 | -12.6 | -8.3 | -12.9 | -16.5 | 15.2 |
| | Target Logits | -64.3 | -52.8 | -43.2 | -59.0 | -50.9 | -50.9 | -68.3 | -71.8 | -65.6 | -68.5 | -72.1 | -66.3 | -62.3 | -54.9 | -60.0 | -66.1 | -57.8 | -60.0 | -60.8 | 7.7 |
| ClassSS | Source + Target Features | 34.1 | 42.4 | 34.0 | 6.5 | 50.7 | 6.5 | 10.5 | 13.2 | 13.2 | -11.1 | 7.4 | -12.1 | 9.2 | 33.6 | 23.9 | -2.9 | 23.0 | 17.1 | 16.6 | 17.0 |
| | Source + Target Logits | 17.0 | 42.6 | 28.1 | -0.9 | 48.6 | -1.8 | 17.5 | 12.3 | 10.3 | -8.8 | 17.5 | -2.0 | 3.1 | 33.2 | 18.7 | -8.2 | 29.9 | 10.4 | 14.8 | 16.3 |
| | Target Features | 24.7 | 38.0 | 21.1 | 8.9 | 34.6 | 16.9 | -39.1 | -16.7 | -27.2 | -61.8 | -34.0 | -42.6 | -50.8 | -17.8 | -11.2 | -42.0 | -16.2 | -9.0 | -12.4 | 29.6 |
| | Target Logits | 11.1 | 35.5 | 25.5 | 2.0 | 47.4 | 8.6 | 4.6 | 4.5 | 5.3 | -24.8 | 4.8 | -8.3 | -19.3 | 26.4 | 13.0 | -21.3 | 16.5 | 7.9 | 7.7 | 18.4 |
| DEV | Features | -54.5 | -56.9 | 26.6 | 24.3 | 62.4 | 28.1 | 40.3 | 43.9 | 36.4 | -21.7 | -27.4 | -25.9 | -7.8 | -1.0 | -13.0 | -26.3 | -4.1 | -14.5 | 0.5 | 33.4 |
| | Logits | -9.2 | 15.2 | 49.6 | 41.6 | 52.5 | 49.2 | 71.2 | 77.8 | 78.7 | 31.2 | 35.2 | 23.9 | 25.6 | 27.2 | 40.9 | 5.6 | 38.4 | 33.3 | 38.2 | 22.6 |
| | Preds | -39.2 | -49.9 | -39.2 | -34.5 | -40.7 | -36.5 | -2.8 | -5.5 | -12.5 | -22.9 | 56.3 | 40.7 | -47.5 | 39.0 | 64.6 | -51.4 | 52.2 | 67.0 | -3.5 | 42.7 |
| DEVN | Features, max normalization | 75.1 | 89.3 | 59.6 | 52.6 | 56.1 | 57.4 | 87.4 | 94.7 | 94.0 | 75.4 | 84.8 | 74.0 | 87.7 | 81.2 | 85.1 | 90.0 | 77.2 | 83.5 | 78.0 | 13.0 |
| | Features, standardization | 71.4 | 87.9 | 72.1 | 23.1 | 76.8 | 3.4 | 79.0 | 88.1 | 86.5 | 73.4 | 73.5 | 67.2 | 85.1 | 69.9 | 71.9 | 80.8 | 74.4 | 71.6 | 72.4 | 14.7 |
| | Logits, max normalization | 70.2 | 85.2 | 53.3 | 53.7 | 48.8 | 57.2 | 86.4 | 94.0 | 93.2 | 72.0 | 81.6 | 76.0 | 85.1 | 76.8 | 81.8 | 86.9 | 71.6 | 82.0 | 75.3 | 13.5 |
| | Logits, standardization | 56.2 | 67.1 | 57.9 | 37.3 | 52.5 | 48.2 | 77.2 | 84.6 | 84.2 | 69.5 | 65.5 | 65.8 | 79.8 | 60.1 | 66.3 | 74.4 | 60.5 | 67.5 | 65.2 | 12.1 |
| | Preds, max normalization | 71.4 | 87.4 | 50.0 | 54.3 | 42.1 | 56.3 | 88.3 | 94.7 | 93.4 | 62.3 | 64.9 | 51.0 | 83.8 | 85.0 | 84.8 | 87.8 | 83.3 | 85.6 | 73.7 | 16.6 |
| | Preds, standardization | 60.9 | 75.2 | 45.2 | 35.3 | 37.6 | 42.4 | 81.3 | 84.9 | 82.4 | 44.4 | 51.8 | 37.6 | 71.1 | 81.7 | 73.4 | 73.3 | 85.6 | 78.1 | 63.5 | 18.2 |
| Entropy | Source Train | 52.2 | 29.2 | 47.8 | 8.3 | 41.1 | 9.7 | 48.3 | 56.2 | 38.1 | 31.2 | 47.6 | 36.3 | 38.3 | 60.2 | 68.0 | 21.6 | 62.5 | 28.7 | 40.3 | 16.5 |
| | Source Train + Target | 33.7 | 18.2 | 52.7 | 5.6 | 52.3 | 27.0 | 67.0 | 53.5 | 35.7 | 32.8 | 48.6 | 30.2 | 35.7 | 68.8 | 58.8 | 29.4 | 69.8 | 28.6 | 41.6 | 17.7 |
| | Source Val | 55.0 | 35.2 | 55.9 | 23.3 | 48.0 | 35.0 | 77.5 | 71.1 | 61.5 | 39.5 | 55.4 | 47.2 | 58.4 | 75.3 | 80.3 | 55.6 | 76.5 | 58.7 | 56.1 | 15.8 |
| | Source Val + Target | 51.0 | 32.4 | 54.1 | 12.5 | 53.4 | 35.4 | 74.0 | 63.7 | 53.8 | 35.4 | 52.3 | 37.1 | 39.4 | 72.0 | 66.1 | 40.9 | 75.9 | 47.0 | 49.8 | 16.2 |
| | Target | 29.1 | 14.8 | 49.1 | 3.9 | 51.4 | 26.5 | 66.3 | 52.7 | 35.5 | 33.1 | 48.3 | 28.9 | 35.2 | 68.4 | 58.2 | 29.2 | 69.4 | 28.3 | 40.5 | 18.0 |
| SND | Features, τ = 0.05 | -87.9 | -89.2 | -81.7 | -86.9 | -82.5 | -88.0 | -88.4 | -88.6 | -90.6 | -83.9 | -85.0 | -88.7 | -86.2 | -84.9 | -87.7 | -86.6 | -87.0 | -91.2 | -86.9 | 2.5 |
| | Features, τ = 0.1 | -87.4 | -85.5 | -78.1 | -82.2 | -78.6 | -86.6 | -84.6 | -84.9 | -84.4 | -83.3 | -82.3 | -86.7 | -82.2 | -78.1 | -81.4 | -82.7 | -79.9 | -87.9 | -83.2 | 3.0 |
| | Features, τ = 0.5 | -84.1 | -83.8 | -83.0 | -76.1 | -85.6 | -81.4 | -80.5 | -83.2 | -80.6 | -73.6 | -78.5 | -77.2 | -75.8 | -76.7 | -79.0 | -76.1 | -77.1 | -79.5 | -79.5 | 3.3 |
| | Logits, τ = 0.05 | -88.8 | -91.4 | -91.4 | -90.8 | -93.1 | -84.0 | -95.9 | -93.2 | -96.3 | -90.4 | -89.7 | -92.9 | -91.6 | -94.0 | -93.5 | -91.5 | -93.6 | -94.2 | -92.0 | 2.7 |
| | Logits, τ = 0.1 | -90.4 | -92.6 | -89.5 | -92.6 | -93.1 | -93.0 | -95.1 | -93.1 | -96.0 | -90.1 | -88.5 | -92.1 | -90.9 | -92.6 | -92.4 | -91.0 | -92.7 | -93.6 | -92.2 | 1.8 |
| | Logits, τ = 0.5 | -85.0 | -84.0 | -80.6 | -72.6 | -87.4 | -79.7 | -89.3 | -87.8 | -87.2 | -84.9 | -82.0 | -85.2 | -87.1 | -82.7 | -84.3 | -85.9 | -85.0 | -82.6 | -84.1 | 3.7 |
| | Preds, τ = 0.05 | -87.3 | -90.3 | -90.4 | -87.0 | -91.3 | -78.1 | -91.8 | -92.0 | -93.7 | -88.9 | -88.0 | -89.5 | -85.7 | -86.6 | -89.9 | -89.2 | -88.9 | -88.7 | -88.7 | 3.3 |
| | Preds, τ = 0.1 | -90.4 | -93.8 | -90.6 | -96.6 | -92.9 | -92.8 | -94.8 | -93.4 | -96.8 | -90.9 | -91.2 | -95.3 | -91.8 | -94.4 | -94.8 | -92.0 | -93.4 | -94.9 | -93.4 | 1.9 |
| | Preds, τ = 0.5 | -88.8 | -93.5 | -79.0 | -85.2 | -85.1 | -87.0 | -88.8 | -90.9 | -90.7 | -87.6 | -88.0 | -89.2 | -84.8 | -84.5 | -89.3 | -88.6 | -87.3 | -91.2 | -87.7 | 3.2 |

Table 19: The weighted Spearman correlation of each validator/task pair for **GVB**.

| | | AD | AW | DA | DW | WA | WD | AC | AP | AR | CA | CP | CR | PA | PC | PR | RA | RC | RP | Mean | Std |
|---|---|---|---|---|---|---|---|---|---|---|---|---|---|---|---|---|---|---|---|---|---|
| Accuracy | Source Train | 79.1 | 76.3 | 87.5 | 72.2 | 81.0 | 66.3 | 50.9 | 53.5 | 58.9 | 51.2 | 64.0 | 54.4 | 76.4 | 70.9 | 70.1 | 58.3 | 68.2 | 65.0 | 66.9 | 10.5 |
| | Source Val | 86.5 | 87.3 | 83.3 | 77.6 | 82.5 | 75.0 | 97.5 | 95.8 | 96.5 | 85.1 | 91.9 | 88.0 | 93.5 | 92.8 | 94.9 | 89.8 | 93.7 | 92.8 | 89.1 | 6.3 |
| BNM | Source Train | 57.8 | 59.1 | 61.7 | 28.2 | 50.0 | 25.1 | 36.6 | 27.2 | 28.8 | 28.9 | 42.0 | 24.4 | 54.1 | 58.4 | 41.9 | 29.5 | 44.9 | 29.4 | 40.5 | 13.1 |
| | Source Train + Target | 65.1 | 77.0 | 82.3 | 42.5 | 84.1 | 20.0 | 73.3 | 61.8 | 69.2 | 56.0 | 69.8 | 56.3 | 68.3 | 82.8 | 71.9 | 57.1 | 78.9 | 67.0 | 65.7 | 15.3 |
| | Source Val | 64.3 | 68.8 | 70.2 | 63.2 | 65.8 | 42.9 | 79.1 | 65.6 | 69.9 | 29.1 | 46.4 | 27.9 | 62.7 | 79.1 | 61.8 | 58.7 | 78.6 | 70.4 | 61.4 | 14.9 |
| | Source Val + Target | 67.5 | 81.5 | 79.9 | 62.9 | 81.2 | 37.6 | 78.8 | 64.9 | 70.1 | 48.7 | 60.7 | 44.4 | 67.5 | 83.4 | 68.7 | 58.4 | 80.2 | 70.2 | 67.0 | 13.0 |
| | Target | 64.4 | 72.8 | 82.5 | 44.6 | 86.0 | 20.2 | 75.2 | 63.4 | 70.5 | 57.9 | 72.6 | 59.7 | 68.7 | 83.3 | 72.5 | 58.5 | 80.2 | 68.2 | 66.7 | 15.2 |
| ClassAMI | Source + Target Features | 75.2 | 82.3 | 86.8 | 31.0 | 90.2 | 32.2 | 85.7 | 69.9 | 74.7 | 72.6 | 81.9 | 75.3 | 81.0 | 85.1 | 76.3 | 67.9 | 88.0 | 70.5 | 73.7 | 16.2 |
| | Source + Target Logits | 74.4 | 80.5 | 86.7 | 30.1 | 90.3 | 32.8 | 85.4 | 69.6 | 73.8 | 71.5 | 81.1 | 74.3 | 80.1 | 85.6 | 75.5 | 67.6 | 88.3 | 70.3 | 73.2 | 16.2 |
| | Target Features | 8.5 | -34.0 | -10.3 | -13.7 | 3.8 | -16.6 | -51.8 | -45.1 | -62.8 | -48.5 | -48.6 | -37.4 | -27.5 | -50.2 | -48.6 | -59.6 | -50.1 | -62.5 | -36.4 | 21.5 |
| | Target Logits | 9.7 | -33.9 | -10.2 | -13.6 | 1.8 | -15.4 | -52.0 | -45.3 | -62.7 | -48.5 | -48.6 | -37.3 | -50.3 | -48.7 | -59.7 | -50.1 | -61.9 | -36.3 | 21.5 |
| ClassSS | Source + Target Features | 49.5 | 71.1 | 62.3 | 17.8 | 74.0 | 18.4 | 48.8 | 50.2 | 43.5 | 21.8 | 49.7 | 31.9 | 23.2 | 66.1 | 47.5 | 30.3 | 60.5 | 57.5 | 45.8 | 17.6 |
| | Source + Target Logits | 59.8 | 71.5 | 61.6 | 17.0 | 77.6 | 29.1 | 58.2 | 56.6 | 53.6 | 40.7 | 61.8 | 49.2 | 32.5 | 70.7 | 57.5 | 42.8 | 65.6 | 62.2 | 53.8 | 15.5 |
| | Target Features | 66.3 | 77.5 | 63.2 | 19.9 | 72.8 | 23.6 | 46.8 | 47.2 | 42.4 | 23.1 | 53.7 | 37.2 | 14.2 | 60.4 | 48.9 | 30.8 | 61.2 | 60.5 | 47.2 | 18.5 |
| | Target Logits | 72.0 | 78.1 | 60.8 | 19.8 | 75.3 | 30.2 | 53.4 | 53.8 | 51.0 | 38.7 | 63.3 | 52.6 | 19.2 | 62.9 | 56.1 | 41.3 | 65.5 | 62.4 | 53.1 | 16.9 |
| DEV | Features | -47.2 | -79.0 | 38.7 | 33.9 | 59.1 | -3.8 | 19.1 | 38.1 | 28.2 | -14.3 | -8.9 | 7.8 | -10.9 | -11.7 | -42.6 | -7.1 | -15.3 | -0.8 | -3.9 | 33.3 |
| | Logits | -23.6 | -24.4 | 62.5 | 44.5 | 72.3 | 17.2 | 52.7 | 58.2 | 66.9 | 21.3 | 30.3 | 27.1 | 28.1 | 50.2 | 32.8 | 11.1 | 42.3 | 40.9 | 33.9 | 26.4 |
| | Preds | -45.2 | -45.5 | -39.0 | -42.7 | -42.3 | -53.0 | -8.8 | -12.8 | -12.1 | -29.4 | 64.2 | 59.0 | -49.7 | 53.1 | 54.8 | -44.8 | 60.2 | 64.6 | -3.9 | 46.4 |
| DEVN | Features, max normalization | 81.5 | 84.5 | 74.5 | 53.3 | 72.1 | 59.0 | 96.1 | 92.3 | 91.9 | 76.8 | 84.2 | 78.2 | 91.5 | 88.6 | 89.8 | 84.5 | 90.1 | 85.5 | 81.9 | 11.1 |
| | Features, standardization | 78.4 | 82.0 | 74.5 | 40.0 | 76.1 | 57.3 | 86.9 | 84.9 | 83.6 | 71.9 | 73.7 | 71.8 | 85.9 | 71.4 | 84.0 | 81.8 | 83.5 | 78.6 | 75.9 | 11.2 |
| | Logits, max normalization | 77.5 | 80.6 | 71.9 | 53.1 | 70.8 | 56.5 | 96.0 | 91.9 | 91.4 | 73.6 | 83.6 | 75.5 | 90.6 | 87.2 | 89.0 | 82.0 | 89.3 | 85.2 | 80.3 | 11.5 |
| | Logits, standardization | 66.4 | 71.2 | 71.4 | 41.8 | 76.5 | 47.3 | 88.7 | 82.2 | 82.3 | 59.5 | 71.3 | 63.6 | 82.2 | 72.4 | 81.3 | 76.6 | 83.0 | 79.8 | 72.1 | 12.2 |
| | Preds, max normalization | 76.9 | 79.8 | 68.4 | 52.3 | 63.4 | 57.0 | 96.5 | 92.7 | 93.3 | 74.1 | 76.1 | 65.7 | 90.7 | 90.4 | 84.9 | 81.0 | 91.6 | 85.9 | 78.9 | 12.9 |
| | Preds, standardization | 65.2 | 71.9 | 57.2 | 37.4 | 54.8 | 45.8 | 89.0 | 84.6 | 85.8 | 53.1 | 67.4 | 51.9 | 81.6 | 87.7 | 72.8 | 71.2 | 85.2 | 75.4 | 69.1 | 15.0 |
| Entropy | Source Train | 56.2 | 56.5 | 58.3 | 24.8 | 47.1 | 22.9 | 36.5 | 25.8 | 25.2 | 26.1 | 39.5 | 23.8 | 51.0 | 56.8 | 37.6 | 27.2 | 43.5 | 29.3 | 38.2 | 12.9 |
| | Source Train + Target | 70.2 | 77.4 | 68.1 | 24.4 | 79.3 | 29.5 | 73.0 | 61.2 | 65.6 | 53.0 | 65.3 | 51.0 | 65.4 | 78.2 | 68.2 | 56.6 | 75.4 | 66.0 | 62.7 | 14.9 |
| | Source Val | 61.9 | 64.7 | 65.2 | 28.7 | 59.8 | 27.5 | 66.0 | 52.6 | 61.9 | 24.6 | 44.3 | 27.4 | 55.5 | 74.3 | 57.7 | 51.7 | 71.4 | 65.7 | 53.4 | 15.7 |
| | Source Val + Target | 68.3 | 78.6 | 70.2 | 23.6 | 79.3 | 28.6 | 72.1 | 58.2 | 64.7 | 46.8 | 60.1 | 44.6 | 64.4 | 78.2 | 66.1 | 56.0 | 75.3 | 66.9 | 61.2 | 15.7 |
| | Target | 71.6 | 75.5 | 61.9 | 25.7 | 80.2 | 31.0 | 74.8 | 63.1 | 67.1 | 56.0 | 68.2 | 54.5 | 66.7 | 78.1 | 69.1 | 59.2 | 76.8 | 67.5 | 63.7 | 14.4 |
| SND | Features, τ = 0.05 | -89.2 | -88.4 | -71.8 | -82.7 | -68.3 | -85.1 | -86.5 | -92.5 | -92.1 | -86.4 | -89.7 | -86.2 | -79.8 | -83.7 | -91.4 | -90.2 | -89.2 | -92.5 | -85.9 | 6.6 |
| | Features, τ = 0.1 | -83.0 | -85.0 | -63.7 | -74.8 | -53.3 | -86.2 | -72.3 | -77.8 | -84.2 | -75.8 | -77.7 | -75.7 | -64.9 | -75.5 | -77.4 | -81.1 | -75.3 | -84.4 | -76.0 | 8.2 |
| | Features, τ = 0.5 | -71.5 | -85.0 | -53.8 | -50.9 | -49.4 | -65.2 | -76.0 | -72.2 | -80.4 | -73.7 | -74.7 | -76.8 | -62.3 | -78.2 | -74.0 | -77.7 | -78.2 | -81.5 | -71.2 | 10.3 |
| | Logits, τ = 0.05 | -91.6 | -90.3 | -83.0 | -86.5 | -86.7 | -88.8 | -96.2 | -96.5 | -96.7 | -92.0 | -95.2 | -94.6 | -90.3 | -92.4 | -95.8 | -93.7 | -95.4 | -94.4 | -92.2 | 3.8 |
| | Logits, τ = 0.1 | -91.8 | -90.5 | -79.2 | -88.6 | -81.1 | -94.8 | -92.7 | -96.6 | -96.8 | -90.3 | -94.4 | -92.9 | -84.1 | -88.9 | -95.5 | -93.0 | -93.8 | -95.0 | -91.1 | 5.0 |
| | Logits, τ = 0.5 | -81.4 | -86.3 | -78.2 | -66.7 | -64.9 | -74.9 | -83.8 | -86.9 | -90.8 | -84.2 | -84.7 | -87.6 | -77.8 | -86.6 | -88.4 | -86.2 | -86.0 | -89.4 | -82.7 | 6.6 |
| | Preds, τ = 0.05 | -83.6 | -87.8 | -54.2 | -80.0 | -55.9 | -82.8 | -87.7 | -91.3 | -90.9 | -89.4 | -89.6 | -88.8 | -78.7 | -84.6 | -89.9 | -89.6 | -89.2 | -89.1 | -83.5 | 10.7 |
| | Preds, τ = 0.1 | -83.2 | -88.8 | -47.8 | -84.7 | -47.9 | -91.1 | -84.4 | -92.7 | -94.9 | -89.2 | -93.6 | -91.9 | -74.0 | -86.7 | -94.1 | -88.9 | -92.1 | -92.5 | -84.4 | 13.8 |
| | Preds, τ = 0.5 | -82.3 | -88.3 | -48.2 | -74.4 | -53.0 | -80.6 | -80.4 | -86.8 | -85.2 | -87.0 | -86.4 | -85.1 | -72.9 | -82.6 | -87.0 | -84.3 | -85.8 | -87.8 | -79.9 | 11.2 |

Table 20: The weighted Spearman correlation of each validator/task pair for **IM**.

| | | AD | AW | DA | DW | WA | WD | AC | AP | AR | CA | CP | CR | PA | PC | PR | RA | RC | RP | Mean | Std |
|---|---|---|---|---|---|---|---|---|---|---|---|---|---|---|---|---|---|---|---|---|---|
| Accuracy | Source Train | 68.4 | 86.9 | 89.2 | 82.1 | 84.5 | 65.3 | 67.1 | 70.5 | 69.3 | 42.0 | 46.2 | 57.0 | 65.6 | 71.0 | 79.9 | 52.7 | 71.7 | 63.3 | 68.5 | 12.9 |
| | Source Val | 82.4 | 93.0 | 84.6 | 83.7 | 82.9 | 75.2 | 97.1 | 97.0 | 97.8 | 75.6 | 85.1 | 86.1 | 88.6 | 92.9 | 95.6 | 94.9 | 96.8 | 95.4 | 89.1 | 7.2 |
| BNM | Source Train | 71.1 | 86.8 | 83.2 | 69.0 | 81.0 | 49.1 | 65.5 | 73.7 | 45.5 | 45.2 | 39.2 | 54.6 | 70.9 | 73.4 | 73.0 | 35.4 | 66.8 | 55.0 | 63.2 | 15.1 |
| | Source Train + Target | 66.9 | 86.9 | 89.5 | 79.6 | 87.2 | 36.4 | 72.5 | 82.4 | 68.8 | 54.0 | 60.9 | 70.8 | 69.0 | 82.9 | 76.1 | 46.0 | 74.9 | 68.8 | 70.7 | 13.9 |
| | Source Val | 64.9 | 82.0 | 84.0 | 74.3 | 85.7 | 56.7 | 82.0 | 85.5 | 76.2 | 44.7 | 44.8 | 58.6 | 79.4 | 84.2 | 79.9 | 54.5 | 80.5 | 73.9 | 71.8 | 13.6 |
| | Source Val + Target | 65.7 | 84.1 | 88.0 | 81.5 | 89.1 | 47.7 | 79.0 | 84.8 | 75.2 | 51.9 | 54.6 | 67.1 | 72.1 | 84.1 | 78.3 | 49.5 | 77.9 | 72.2 | 72.4 | 13.1 |
| | Target | 66.3 | 85.8 | 89.3 | 78.9 | 86.7 | 35.5 | 67.5 | 81.6 | 69.3 | 54.0 | 63.2 | 72.0 | 68.5 | 81.8 | 75.9 | 46.3 | 73.5 | 69.2 | 70.3 | 13.7 |
| ClassAMI | Source + Target Features | 74.6 | 91.3 | 94.4 | 83.8 | 93.5 | 62.5 | 92.3 | 92.8 | 90.6 | 71.2 | 78.0 | 85.2 | 83.1 | 94.4 | 92.5 | 79.4 | 92.6 | 85.9 | 85.4 | 8.9 |
| | Source + Target Logits | 73.4 | 90.8 | 93.0 | 83.7 | 92.2 | 62.9 | 89.7 | 91.6 | 88.5 | 66.3 | 75.9 | 83.0 | 79.2 | 94.2 | 90.0 | 75.1 | 91.7 | 84.9 | 83.7 | 9.3 |
| | Target Features | -14.0 | -40.4 | -13.5 | -0.1 | -19.7 | -25.4 | -12.0 | -50.9 | -35.1 | -30.2 | -42.3 | -38.9 | -45.6 | -37.6 | -51.0 | -18.8 | -35.6 | -31.1 | -30.1 | 14.1 |
| | Target Logits | -6.5 | -45.9 | -6.3 | -3.6 | -6.8 | -17.4 | -21.6 | -51.4 | -11.8 | -26.0 | -33.7 | -29.5 | -48.3 | -42.2 | -49.2 | -4.8 | -38.5 | -19.1 | -25.7 | 16.7 |
| ClassSS | Source + Target Features | 29.0 | 46.5 | 55.7 | 67.7 | 38.6 | 24.5 | -22.9 | -16.8 | 7.9 | -30.1 | -22.0 | -1.9 | -6.8 | 11.4 | 14.9 | 3.0 | -13.5 | -5.7 | 10.0 | 27.8 |
| | Source + Target Logits | 12.9 | 14.3 | 37.1 | 48.8 | 17.7 | 7.8 | -22.5 | -19.7 | -6.5 | -31.4 | -24.7 | -11.5 | -8.5 | 6.6 | 2.6 | -11.1 | -25.3 | -13.8 | -1.5 | 21.3 |
| | Target Features | 42.3 | 54.5 | 49.6 | 68.1 | 28.0 | 33.3 | -50.7 | -23.2 | -6.2 | -41.4 | -34.7 | -7.3 | -40.8 | -42.9 | -8.1 | -4.9 | -47.3 | -6.2 | -2.1 | 37.7 |
| | Target Logits | 26.3 | 64.0 | 32.1 | 73.6 | 10.9 | 14.7 | -46.3 | -21.7 | -19.9 | -41.6 | -36.6 | -18.1 | -37.4 | -33.1 | -20.5 | -17.4 | -44.7 | -15.2 | -7.3 | 35.3 |
| DEV | Features | -26.0 | -9.5 | 27.0 | 42.5 | 51.6 | 44.7 | 26.4 | 45.4 | 44.3 | -3.7 | 0.6 | 2.4 | -15.1 | 1.2 | 6.3 | -0.4 | 0.9 | 16.3 | 14.1 | 23.1 |
| | Logits | 37.6 | 57.9 | 52.0 | 53.4 | 66.1 | 65.2 | 54.0 | 65.9 | 80.3 | 39.6 | 47.5 | 57.7 | 44.9 | 48.7 | 59.0 | 35.5 | 65.2 | 70.0 | 55.6 | 11.7 |
| | Preds | -11.0 | -19.1 | -33.9 | -36.9 | -39.5 | -35.7 | -33.9 | -9.3 | -23.9 | -18.0 | 58.7 | 75.2 | -20.2 | 77.2 | 60.3 | -51.3 | 75.5 | 57.4 | 4.0 | 46.2 |
| DEVN | Features, max normalization | 82.7 | 90.8 | 86.4 | 74.9 | 83.0 | 67.2 | 97.1 | 97.9 | 97.2 | 80.5 | 85.7 | 89.1 | 91.9 | 94.5 | 95.1 | 94.1 | 96.7 | 95.8 | 88.9 | 8.4 |
| | Features, standardization | 74.9 | 61.2 | 82.4 | 65.5 | 79.6 | 64.1 | 92.9 | 95.1 | 89.4 | 84.3 | 84.1 | 88.0 | 89.9 | 89.8 | 89.1 | 87.3 | 91.3 | 90.6 | 83.3 | 10.0 |
| | Logits, max normalization | 79.4 | 92.0 | 84.7 | 74.6 | 81.5 | 66.9 | 97.2 | 97.8 | 97.2 | 78.8 | 84.6 | 88.2 | 89.9 | 93.9 | 94.6 | 94.0 | 96.1 | 95.6 | 88.2 | 8.7 |
| | Logits, standardization | 67.6 | 83.8 | 80.8 | 64.8 | 80.6 | 61.3 | 92.8 | 95.3 | 91.8 | 78.6 | 81.1 | 83.8 | 87.2 | 87.2 | 89.0 | 86.2 | 90.7 | 89.0 | 82.9 | 9.4 |
| | Preds, max normalization | 80.4 | 92.2 | 83.8 | 74.6 | 78.9 | 66.8 | 96.9 | 97.5 | 97.1 | 76.1 | 83.0 | 88.0 | 88.8 | 93.5 | 93.3 | 92.8 | 96.3 | 94.9 | 87.5 | 8.9 |
| | Preds, standardization | 70.4 | 85.7 | 76.9 | 55.2 | 75.8 | 56.9 | 91.0 | 94.0 | 92.0 | 67.6 | 81.6 | 88.4 | 84.3 | 87.6 | 89.4 | 85.1 | 89.1 | 92.3 | 81.3 | 11.5 |
| Entropy | Source Train | 62.8 | 78.1 | 73.5 | 68.3 | 74.0 | 46.1 | 64.7 | 65.8 | 46.7 | 39.5 | 37.5 | 53.3 | 59.8 | 69.5 | 65.7 | 32.5 | 65.6 | 44.1 | 58.2 | 13.5 |
| | Source Train + Target | 64.4 | 74.9 | 65.3 | 79.8 | 70.7 | 64.6 | 62.8 | 67.2 | 69.5 | 48.5 | 55.2 | 67.6 | 54.5 | 71.4 | 63.2 | 52.9 | 66.2 | 55.9 | 64.1 | 7.9 |
| | Source Val | 3.6 | 33.2 | 59.4 | 59.3 | 69.8 | 49.4 | 51.4 | 47.5 | 68.8 | 32.0 | 38.5 | 52.2 | 59.5 | 72.2 | 61.2 | 50.2 | 61.0 | 53.1 | 51.2 | 16.1 |
| | Source Val + Target | 41.3 | 50.7 | 61.9 | 75.5 | 70.0 | 58.8 | 54.4 | 57.4 | 69.8 | 43.6 | 49.4 | 63.8 | 54.3 | 70.7 | 61.1 | 52.3 | 61.7 | 54.4 | 58.4 | 9.2 |
| | Target | 57.9 | 65.4 | 47.5 | 77.2 | 63.8 | 65.6 | 55.2 | 63.3 | 69.5 | 47.5 | 55.8 | 68.4 | 47.6 | 67.3 | 59.6 | 54.8 | 61.0 | 55.5 | 60.2 | 8.0 |
| SND | Features, $\tau = 0.05$ | -61.3 | -62.7 | -61.4 | -88.5 | -60.8 | -80.3 | -54.4 | -50.4 | -90.1 | -56.5 | -60.4 | -69.9 | -60.4 | -67.5 | -72.2 | -83.2 | -56.8 | -63.0 | -66.7 | 11.4 |
| | Features, $\tau = 0.1$ | -61.9 | -62.7 | -61.9 | -86.4 | -60.4 | -80.4 | -56.0 | -50.9 | -87.3 | -55.9 | -60.8 | -69.4 | -61.9 | -64.4 | -72.2 | -79.9 | -56.4 | -63.0 | -66.2 | 10.5 |
| | Features, $\tau = 0.5$ | -67.1 | -69.1 | -68.8 | -83.3 | -69.7 | -81.9 | -72.0 | -61.9 | -85.5 | -68.8 | -69.0 | -75.9 | -74.5 | -81.0 | -77.7 | -83.1 | -75.4 | -74.3 | -74.4 | 6.4 |
| | Logits, $\tau = 0.05$ | -63.6 | -64.1 | -62.7 | -88.6 | -61.9 | -81.4 | -54.4 | -51.5 | -91.8 | -59.6 | -61.3 | -73.7 | -62.4 | -72.4 | -76.1 | -84.5 | -57.6 | -64.9 | -68.5 | 11.5 |
| | Logits, $\tau = 0.1$ | -63.8 | -63.9 | -62.0 | -87.7 | -61.2 | -82.1 | -55.4 | -51.6 | -91.5 | -60.0 | -61.6 | -73.7 | -63.3 | -71.7 | -75.7 | -84.1 | -57.8 | -65.0 | -68.4 | 11.3 |
| | Logits, $\tau = 0.5$ | -66.9 | -67.3 | -63.1 | -74.2 | -62.8 | -78.9 | -68.0 | -57.5 | -87.0 | -67.2 | -65.7 | -75.4 | -72.4 | -77.5 | -78.8 | -82.6 | -72.8 | -71.8 | -71.7 | 7.4 |
| | Preds, $\tau = 0.05$ | -66.0 | -66.4 | -64.2 | -89.7 | -63.4 | -81.7 | -50.4 | -50.3 | -89.3 | -59.6 | -60.9 | -73.3 | -62.8 | -72.7 | -76.3 | -81.4 | -56.4 | -65.2 | -68.3 | 11.5 |
| | Preds, $\tau = 0.1$ | -66.3 | -66.7 | -64.6 | -90.8 | -64.0 | -82.7 | -50.7 | -50.2 | -90.8 | -59.6 | -60.9 | -73.5 | -62.9 | -72.6 | -76.4 | -84.1 | -57.0 | -65.5 | -68.9 | 11.9 |
| | Preds, $\tau = 0.5$ | -70.4 | -70.4 | -68.0 | -77.7 | -65.4 | -83.5 | -66.8 | -63.1 | -82.6 | -71.0 | -67.4 | -74.0 | -70.9 | -65.6 | -75.3 | -83.0 | -68.1 | -74.2 | -72.1 | 6.1 |

Table 21: The weighted Spearman correlation of each validator/task pair for **MCC**.

| | | AD | AW | DA | DW | WA | WD | AC | AP | AR | CA | CP | CR | PA | PC | PR | RA | RC | RP | Mean | Std |
|---|---|---|---|---|---|---|---|---|---|---|---|---|---|---|---|---|---|---|---|---|---|
| Accuracy | Source Train | 78.2 | 79.3 | 88.7 | 79.3 | 87.2 | 68.3 | 54.3 | 65.3 | 60.7 | 56.4 | 67.6 | 63.1 | 73.7 | 65.8 | 76.9 | 64.5 | 73.0 | 70.2 | 70.7 | 9.4 |
| | Source Val | 83.3 | 89.0 | 82.5 | 81.9 | 87.1 | 76.8 | 95.2 | 95.0 | 97.2 | 85.6 | 91.4 | 89.2 | 92.0 | 90.9 | 95.3 | 92.3 | 92.9 | 93.8 | 89.5 | 5.4 |
| BNM | Source Train | 77.0 | 81.2 | 81.7 | 61.5 | 80.3 | 40.3 | 53.0 | 55.7 | 43.1 | 32.5 | 48.4 | 45.6 | 63.4 | 57.3 | 70.4 | 49.5 | 56.5 | 55.7 | 58.5 | 14.3 |
| | Source Train + Target | 65.5 | 81.6 | 83.4 | 66.8 | 82.8 | -5.0 | 78.6 | 59.7 | 67.7 | 43.5 | 63.9 | 60.7 | 55.0 | 69.2 | 69.7 | 48.4 | 67.8 | 66.7 | 62.6 | 19.5 |
| | Source Val | 79.9 | 84.2 | 79.2 | 65.8 | 83.8 | 53.7 | 88.3 | 75.1 | 75.5 | 34.5 | 56.2 | 50.7 | 74.9 | 78.8 | 80.1 | 74.6 | 80.1 | 77.6 | 71.8 | 13.7 |
| | Source Val + Target | 74.8 | 83.7 | 82.4 | 72.4 | 85.8 | 32.1 | 85.2 | 70.4 | 73.8 | 42.1 | 62.0 | 57.7 | 60.4 | 72.3 | 74.7 | 58.0 | 74.2 | 73.2 | 68.6 | 14.1 |
| | Target | 63.9 | 78.7 | 82.1 | 65.7 | 81.8 | -7.8 | 76.1 | 56.0 | 66.3 | 43.3 | 64.7 | 61.7 | 54.3 | 67.9 | 68.1 | 48.0 | 67.0 | 65.2 | 61.3 | 19.6 |
| ClassAMI | Source + Target Features | 76.5 | 82.5 | 88.0 | 81.7 | 88.6 | 77.6 | 90.4 | 84.6 | 85.0 | 73.4 | 82.7 | 85.3 | 80.5 | 91.6 | 87.7 | 74.0 | 91.4 | 84.4 | 83.7 | 5.5 |
| | Source + Target Logits | 78.4 | 85.2 | 87.5 | 81.4 | 89.3 | 76.5 | 89.8 | 80.6 | 83.0 | 69.2 | 80.5 | 82.3 | 78.1 | 90.6 | 85.9 | 71.4 | 90.3 | 83.3 | 82.4 | 6.0 |
| | Target Features | -9.3 | -34.0 | 1.2 | 7.6 | -26.8 | 17.1 | -59.3 | -49.1 | -43.5 | -35.2 | -44.3 | -30.1 | -23.9 | -28.7 | -37.5 | -45.9 | -15.7 | -42.2 | -27.7 | 20.2 |
| | Target Logits | -12.2 | -33.0 | 1.9 | 4.6 | -28.8 | 18.2 | -61.1 | -54.0 | -44.5 | -36.4 | -44.5 | -30.0 | -31.4 | -39.5 | -47.2 | -18.9 | -43.9 | -29.2 | -29.2 | 20.3 |
| ClassSS | Source + Target Features | 50.2 | 60.7 | 65.4 | 63.6 | 71.8 | 41.3 | 27.4 | 13.5 | 35.0 | 13.3 | 40.9 | 29.9 | 21.2 | 42.8 | 30.0 | 27.7 | 41.0 | 46.1 | 40.1 | 16.8 |
| | Source + Target Logits | 54.0 | 56.5 | 59.5 | 59.3 | 73.1 | 40.6 | 28.3 | 21.9 | 40.0 | 15.7 | 47.8 | 33.2 | 26.7 | 44.9 | 27.6 | 29.3 | 46.4 | 51.7 | 42.0 | 15.0 |
| | Target Features | 66.5 | 68.2 | 62.7 | 68.8 | 70.7 | 51.6 | 11.3 | 10.4 | 29.4 | 16.3 | 45.8 | 30.5 | 12.5 | 29.6 | 24.0 | 22.6 | 31.2 | 44.7 | 38.7 | 20.9 |
| | Target Logits | 72.4 | 72.8 | 56.6 | 72.4 | 71.5 | 52.3 | 18.7 | 17.9 | 31.2 | 19.7 | 52.9 | 31.9 | 20.5 | 32.5 | 24.6 | 24.9 | 38.2 | 52.1 | 42.4 | 20.0 |
| DEV | Features | -34.9 | -37.9 | 36.7 | 37.9 | 68.3 | 38.8 | 41.5 | 33.6 | 18.5 | -13.5 | -10.0 | 8.5 | -9.5 | 8.4 | -13.7 | -14.9 | 16.0 | 5.2 | 9.9 | 28.1 |
| | Logits | 22.3 | 26.7 | 59.4 | 50.1 | 74.3 | 57.6 | 55.3 | 68.5 | 75.4 | 35.2 | 39.0 | 51.3 | 26.0 | 40.2 | 64.2 | 40.4 | 54.2 | 73.6 | 50.8 | 16.6 |
| | Preds | -19.1 | -35.3 | -33.4 | -41.9 | -31.3 | -48.5 | -12.6 | -8.3 | -23.5 | -29.8 | 84.8 | 81.9 | -40.9 | 73.3 | 79.3 | -47.7 | 85.4 | 81.5 | 6.3 | 53.9 |
| DEVN | Features, max normalization | 79.1 | 86.0 | 82.2 | 65.9 | 81.1 | 65.5 | 93.8 | 90.2 | 95.3 | 80.9 | 87.2 | 85.2 | 87.2 | 85.9 | 92.0 | 88.0 | 88.0 | 90.7 | 84.8 | 8.0 |
| | Features, standardization | 63.2 | 63.2 | 84.7 | 59.1 | 81.3 | 62.6 | 90.0 | 90.1 | 87.3 | 86.7 | 85.7 | 85.4 | 84.4 | 79.2 | 83.2 | 84.2 | 85.8 | 84.7 | 80.0 | 10.0 |
| | Logits, max normalization | 77.8 | 86.8 | 79.2 | 66.1 | 80.0 | 65.0 | 93.9 | 92.1 | 95.3 | 80.1 | 87.2 | 84.7 | 85.5 | 84.4 | 91.7 | 85.7 | 86.7 | 90.4 | 84.0 | 8.2 |
| | Logits, standardization | 69.9 | 77.4 | 79.6 | 61.0 | 80.7 | 59.4 | 92.2 | 90.9 | 90.0 | 83.4 | 83.5 | 81.5 | 83.1 | 77.3 | 84.5 | 81.3 | 83.1 | 85.3 | 80.2 | 8.7 |
| | Preds, max normalization | 78.0 | 87.1 | 76.8 | 65.0 | 78.5 | 64.4 | 93.8 | 91.5 | 95.1 | 77.1 | 88.2 | 84.9 | 84.7 | 87.7 | 92.4 | 86.1 | 91.0 | 91.2 | 84.1 | 8.8 |
| | Preds, standardization | 73.2 | 80.7 | 69.6 | 45.5 | 72.6 | 54.2 | 90.1 | 85.5 | 87.3 | 62.2 | 85.4 | 83.6 | 76.7 | 83.3 | 89.4 | 81.9 | 91.5 | 90.9 | 78.0 | 12.7 |
| Entropy | Source Train | 72.6 | 74.9 | 80.9 | 64.4 | 82.8 | 44.9 | 56.8 | 57.7 | 45.6 | 27.6 | 44.5 | 44.4 | 61.9 | 58.4 | 70.0 | 47.1 | 54.9 | 56.4 | 58.1 | 14.1 |
| | Source Train + Target | 79.5 | 80.1 | 71.2 | 76.4 | 79.1 | 68.9 | 70.5 | 67.7 | 70.8 | 45.7 | 63.4 | 58.6 | 63.9 | 65.2 | 71.9 | 61.8 | 62.6 | 71.8 | 68.3 | 8.3 |
| | Source Val | 21.4 | 41.9 | 66.8 | 44.9 | 77.6 | 44.2 | 65.2 | 60.3 | 70.5 | 25.0 | 50.2 | 44.5 | 68.7 | 69.9 | 75.4 | 71.8 | 66.6 | 71.7 | 57.6 | 16.6 |
| | Source Val + Target | 65.4 | 63.4 | 67.2 | 66.2 | 78.7 | 54.7 | 65.8 | 63.0 | 70.1 | 41.2 | 60.4 | 54.4 | 64.7 | 64.7 | 72.5 | 65.1 | 62.5 | 71.0 | 63.9 | 7.9 |
| | Target | 78.1 | 71.8 | 59.3 | 73.7 | 75.1 | 69.6 | 65.8 | 65.8 | 68.9 | 48.0 | 64.6 | 59.8 | 62.8 | 61.9 | 69.1 | 61.9 | 60.4 | 70.4 | 65.8 | 6.9 |
| SND | Features, $\tau = 0.05$ | -77.2 | -74.6 | -70.6 | -91.1 | -75.7 | -82.3 | -85.6 | -88.9 | -93.9 | -85.1 | -85.7 | -84.6 | -82.8 | -84.3 | -87.1 | -88.3 | -79.8 | -88.8 | -83.7 | 5.9 |
| | Features, $\tau = 0.1$ | -76.4 | -74.1 | -69.0 | -88.4 | -73.4 | -86.6 | -82.4 | -84.7 | -87.2 | -79.9 | -79.4 | -77.7 | -76.7 | -79.9 | -83.3 | -82.1 | -70.8 | -83.7 | -79.8 | 5.4 |
| | Features, $\tau = 0.5$ | -81.4 | -81.9 | -77.9 | -77.5 | -84.4 | -69.7 | -86.6 | -88.9 | -82.7 | -83.6 | -86.5 | -82.5 | -85.2 | -82.3 | -86.0 | -80.4 | -73.2 | -81.0 | -81.8 | 4.7 |
| | Logits, $\tau = 0.05$ | -78.4 | -77.0 | -69.3 | -92.8 | -76.6 | -88.2 | -90.9 | -91.9 | -96.2 | -92.0 | -91.7 | -92.4 | -90.7 | -87.9 | -89.6 | -90.6 | -89.8 | -91.8 | -87.7 | 6.9 |
| | Logits, $\tau = 0.1$ | -78.1 | -76.4 | -66.5 | -90.6 | -74.4 | -89.3 | -87.7 | -90.1 | -93.7 | -89.6 | -88.4 | -89.5 | -87.4 | -82.7 | -86.9 | -88.2 | -79.7 | -88.5 | -84.9 | 6.9 |
| | Logits, $\tau = 0.5$ | -77.1 | -77.3 | -65.1 | -71.5 | -68.5 | -61.4 | -85.4 | -84.8 | -84.7 | -85.6 | -85.1 | -85.7 | -84.4 | -85.1 | -82.3 | -77.9 | -82.8 | -79.6 | -79.6 | 7.7 |
| | Preds, $\tau = 0.05$ | -83.1 | -79.8 | -84.3 | -93.9 | -87.4 | -83.8 | -92.2 | -92.2 | -90.8 | -91.4 | -92.7 | -94.6 | -92.0 | -90.9 | -91.3 | -88.6 | -94.4 | -87.2 | -89.5 | 4.2 |
| | Preds, $\tau = 0.1$ | -83.0 | -79.7 | -85.0 | -96.3 | -88.3 | -94.1 | -92.4 | -94.5 | -96.9 | -92.8 | -94.5 | -95.7 | -93.0 | -92.1 | -92.9 | -90.5 | -96.0 | -93.2 | -91.7 | 4.6 |
| | Preds, $\tau = 0.5$ | -83.6 | -82.1 | -68.5 | -82.5 | -73.1 | -84.7 | -85.7 | -88.2 | -91.1 | -88.0 | -89.4 | -89.6 | -82.2 | -73.6 | -86.1 | -87.7 | -75.9 | -88.3 | -83.4 | 6.3 |

Table 22: The weighted Spearman correlation of each validator/task pair for **MCD**.

| | | AD | AW | DA | DW | WA | WD | AC | AP | AR | CA | CP | CR | PA | PC | PR | RA | RC | RP | Mean | Std |
|---|---|---|---|---|---|---|---|---|---|---|---|---|---|---|---|---|---|---|---|---|---|
| Accuracy | Source Train | 83.1 | 79.7 | 80.0 | 74.2 | 80.7 | 79.5 | 82.1 | 79.1 | 63.0 | 79.4 | 73.7 | 72.6 | 87.9 | 79.4 | 82.7 | 77.1 | 88.5 | 80.9 | 79.1 | 5.6 |
| | Source Val | 86.2 | 84.4 | 80.9 | 77.4 | **83.9** | 81.2 | 96.7 | **98.4** | **98.4** | 89.8 | **94.5** | 93.4 | 95.8 | **93.2** | **95.3** | **96.0** | 94.4 | **96.2** | **90.9** | 6.5 |
| BNM | Source Train | 88.2 | 85.6 | 79.8 | 60.9 | 67.4 | 75.1 | 88.4 | 83.6 | 63.3 | 90.5 | 72.2 | 81.0 | 91.3 | 83.6 | 84.5 | 82.1 | 93.2 | 75.2 | 80.3 | 9.3 |
| | Source Train + Target | 90.9 | **91.3** | 58.7 | **81.9** | 60.3 | 77.2 | 89.8 | 91.5 | 90.6 | **93.7** | 92.3 | **93.8** | 91.1 | 87.3 | 94.1 | 83.0 | 94.3 | 90.6 | 86.3 | 10.5 |
| | Source Val | 88.9 | 84.6 | 75.6 | 75.9 | 64.1 | 71.3 | 91.4 | 92.9 | 91.9 | 92.0 | 86.7 | 89.2 | 92.8 | 92.2 | 93.3 | 92.7 | **96.1** | 93.9 | 87.0 | 8.8 |
| | Source Val + Target | **91.8** | 90.3 | 66.0 | 82.8 | 61.7 | 73.5 | 90.6 | 92.7 | 92.0 | 93.4 | 91.2 | 92.7 | 91.7 | 89.7 | 94.5 | 87.5 | **95.6** | 93.4 | 87.3 | 9.7 |
| | Target | 88.6 | **91.7** | 56.0 | **84.4** | 64.2 | 76.6 | 87.6 | 91.1 | 90.8 | **93.8** | 92.9 | **94.6** | 90.7 | 86.9 | 94.2 | 82.6 | 93.6 | 90.6 | 86.2 | 10.3 |
| ClassAMI | Source + Target Features | 86.5 | **90.4** | 45.1 | 60.6 | 57.5 | 79.9 | 74.7 | 83.7 | 76.0 | 89.6 | 83.5 | 88.6 | 91.5 | 65.9 | 87.6 | 91.8 | 89.8 | 91.9 | 79.7 | 13.4 |
| | Source + Target Logits | 86.7 | **89.5** | 46.2 | 61.1 | 57.5 | 80.0 | 81.2 | 85.5 | 78.1 | **90.5** | 84.6 | 89.2 | 92.8 | 68.7 | 88.1 | 93.2 | 91.2 | 93.1 | 80.9 | 13.3 |
| | Target Features | -21.2 | -54.2 | -55.9 | -9.3 | -58.2 | 17.6 | -82.0 | -70.4 | -76.9 | -73.5 | -70.3 | -74.1 | -65.2 | -61.8 | -69.3 | -71.0 | -72.8 | -64.9 | -57.4 | 25.6 |
| | Target Logits | -25.3 | -53.5 | -60.5 | -7.9 | -58.4 | 9.3 | -83.5 | -66.8 | -73.4 | -73.4 | -68.8 | -73.3 | -62.8 | -65.1 | -67.6 | -69.6 | -74.0 | -58.7 | -57.4 | 23.9 |
| ClassSS | Source + Target Features | 43.9 | 33.8 | 4.8 | 20.1 | 16.9 | 34.7 | -1.2 | 24.2 | 20.2 | 14.6 | 18.3 | 5.5 | 25.7 | 23.2 | 37.8 | 42.0 | 34.9 | 37.1 | 24.3 | 12.8 |
| | Source + Target Logits | 50.6 | 45.1 | 2.1 | 26.4 | -4.6 | 39.2 | -2.8 | 41.0 | 16.1 | 16.8 | 28.4 | 17.1 | 9.4 | 18.1 | 30.1 | 48.7 | 10.5 | 39.2 | 24.0 | 17.0 |
| | Target Features | 36.4 | 69.7 | -23.2 | 47.3 | -23.5 | 35.9 | -36.7 | 9.1 | 15.4 | -15.7 | 2.5 | -10.0 | -18.1 | -14.2 | 16.2 | 20.9 | -7.2 | 31.1 | 7.6 | 27.9 |
| | Target Logits | 43.1 | 52.4 | -27.6 | 35.1 | -34.3 | 40.9 | -31.9 | 19.7 | 8.6 | -18.3 | 10.4 | -6.1 | -31.1 | -6.3 | 8.7 | 23.4 | -5.6 | 29.0 | 6.1 | 27.1 |
| DEV | Features | -61.7 | -68.4 | -2.6 | 46.1 | 34.5 | 40.5 | 38.8 | 60.1 | 42.7 | -43.0 | -18.1 | -16.5 | -11.5 | 5.1 | -4.5 | -19.9 | 9.1 | -6.4 | 1.4 | 36.1 |
| | Logits | -9.1 | -8.0 | 22.4 | 57.9 | 52.8 | 60.0 | 68.3 | 75.2 | 67.6 | 41.7 | 61.8 | 67.8 | 63.4 | 66.8 | 66.9 | 51.2 | 81.3 | 65.2 | 53.0 | 25.2 |
| | Preds | -16.5 | -20.0 | -2.5 | -40.2 | -6.3 | -29.7 | 14.4 | 9.7 | 15.1 | 0.2 | 91.1 | 84.3 | -1.7 | 61.6 | 83.2 | -18.1 | 86.8 | 83.9 | 22.0 | 44.8 |
| DEVN | Features, max normalization | 87.6 | 86.4 | 81.6 | 59.7 | 82.1 | 65.0 | **97.1** | 97.9 | 96.6 | 92.2 | 94.1 | 93.4 | **96.6** | 92.5 | 91.3 | 94.9 | 94.5 | 92.5 | 88.7 | 10.4 |
| | Features, standardization | 83.1 | 78.7 | 71.0 | 52.2 | 71.9 | 63.7 | 89.3 | 88.2 | 86.4 | 85.9 | 78.8 | 79.7 | 92.8 | 80.2 | 88.2 | 90.5 | 88.7 | 82.2 | 80.6 | 10.1 |
| | Logits, max normalization | 87.3 | 84.3 | **82.1** | 60.7 | 82.8 | 65.2 | **97.3** | 98.1 | 97.2 | 89.6 | 93.1 | 92.9 | **96.1** | 92.2 | 90.6 | 94.3 | 94.8 | 93.6 | 88.4 | 10.2 |
| | Logits, standardization | 77.8 | 78.3 | 71.8 | 53.8 | 76.6 | 64.0 | 91.2 | 91.9 | 86.8 | 78.7 | 70.8 | 77.6 | 90.8 | 82.5 | 84.1 | 88.2 | 90.0 | 86.7 | 80.1 | 10.0 |
| | Preds, max normalization | 87.1 | 84.8 | 81.3 | 60.9 | 83.4 | 64.7 | 96.7 | 98.3 | 97.7 | 91.2 | 93.7 | 93.4 | **96.4** | 92.1 | 92.2 | 94.3 | **95.3** | 95.8 | 88.9 | 10.4 |
| | Preds, standardization | 83.0 | 82.1 | 79.6 | 51.7 | 77.5 | 58.7 | 92.2 | 95.6 | 91.7 | 85.2 | 91.2 | 91.7 | 93.4 | 92.3 | 90.6 | 91.4 | **95.1** | 95.3 | 85.5 | 12.0 |
| Entropy | Source Train | 79.8 | 71.3 | 21.2 | 55.0 | 29.8 | 71.3 | 81.9 | 82.4 | 61.3 | 78.4 | 67.4 | 75.7 | 91.0 | 83.7 | 83.6 | 81.3 | 91.1 | 74.3 | 71.7 | 18.9 |
| | Source Train + Target | 78.5 | 62.9 | -33.5 | 73.3 | -26.5 | 79.9 | 68.8 | 83.0 | 85.6 | **91.1** | 83.9 | 83.9 | 88.6 | 81.2 | 88.8 | 81.1 | 87.2 | 86.7 | 69.1 | 35.8 |
| | Source Val | 40.9 | 37.0 | -16.4 | 64.5 | -4.3 | 68.9 | 72.1 | 84.8 | 88.6 | 90.0 | 83.0 | 82.1 | 91.7 | 90.1 | 91.7 | 91.0 | 89.4 | 90.3 | 68.6 | 32.3 |
| | Source Val + Target | 70.9 | 44.6 | -34.6 | 75.0 | -27.3 | 74.4 | 70.7 | 83.6 | 88.1 | **91.1** | 83.9 | 83.4 | 88.9 | 82.1 | 89.5 | 83.5 | 88.0 | 88.8 | 68.0 | 36.6 |
| | Target | 67.4 | 58.5 | -35.0 | 66.0 | -31.8 | 77.5 | 70.4 | 81.2 | 85.6 | **91.2** | 81.9 | 83.7 | 88.9 | 79.0 | 87.9 | 80.8 | 86.8 | 86.5 | 67.0 | 36.5 |
| SND | Features, $\tau = 0.05$ | -77.3 | -75.1 | -71.1 | -95.2 | -74.3 | -91.1 | -85.6 | -88.7 | -90.7 | -74.9 | -87.1 | -84.3 | -84.6 | -81.3 | -85.7 | -84.8 | -78.1 | -86.3 | -82.1 | 7.5 |
| | Features, $\tau = 0.1$ | -76.0 | -73.9 | -69.6 | -91.6 | -70.8 | -88.9 | -85.5 | -85.9 | -88.3 | -75.2 | -84.2 | -82.3 | -67.4 | -80.3 | -82.9 | -81.5 | -79.0 | -82.4 | -80.3 | 6.7 |
| | Features, $\tau = 0.5$ | -82.8 | -82.1 | -78.7 | -78.2 | -74.6 | -79.2 | -90.0 | -84.6 | -84.9 | -80.6 | -82.7 | -82.2 | -79.9 | -87.3 | -82.6 | -81.8 | -87.3 | -81.5 | -82.3 | **3.6** |
| | Logits, $\tau = 0.05$ | -78.4 | -75.9 | -72.9 | -94.3 | -76.9 | -91.7 | -86.2 | -89.0 | -89.7 | -75.2 | -86.0 | -84.5 | -67.3 | -80.9 | -85.7 | -84.2 | -78.6 | -85.7 | -82.4 | 6.9 |
| | Logits, $\tau = 0.1$ | -76.8 | -74.8 | -72.1 | -87.7 | -75.5 | -87.3 | -85.9 | -86.2 | -87.4 | -75.7 | -81.7 | -82.5 | -68.7 | -79.5 | -82.7 | -79.2 | -79.4 | -79.8 | -80.2 | 5.4 |
| | Logits, $\tau = 0.5$ | -78.7 | -81.9 | -80.1 | -73.9 | -79.7 | -78.9 | -88.9 | -84.8 | -87.9 | -79.8 | -82.5 | -82.3 | -80.3 | -87.5 | -83.7 | -81.8 | -86.0 | -80.5 | -82.2 | **3.7** |
| | Preds, $\tau = 0.05$ | -76.0 | -74.6 | -69.2 | -94.6 | -70.7 | -89.4 | -83.6 | -87.4 | -90.6 | -74.8 | -86.1 | -84.5 | -79.9 | -86.0 | -83.7 | -77.8 | -85.4 | -81.1 | -81.1 | 7.8 |
| | Preds, $\tau = 0.1$ | -75.4 | -73.3 | -67.6 | -96.0 | -68.4 | -89.3 | -83.0 | -87.0 | -91.5 | -74.7 | -86.7 | -85.3 | -64.6 | -78.6 | -86.9 | -82.7 | -77.2 | -86.0 | -80.8 | 8.6 |
| | Preds, $\tau = 0.5$ | -78.3 | -78.5 | -75.1 | -86.5 | -70.1 | -89.2 | -87.7 | -87.2 | -88.4 | -78.6 | -84.6 | -84.3 | -75.8 | -87.5 | -86.1 | -83.5 | -85.4 | -84.0 | -82.8 | 5.3 |

Table 23: The weighted Spearman correlation of each validator/task pair for **MMD**.

| | | AD | AW | DA | DW | WA | WD | AC | AP | AR | CA | CP | CR | PA | PC | PR | RA | RC | RP | Mean | Std |
|---|---|---|---|---|---|---|---|---|---|---|---|---|---|---|---|---|---|---|---|---|---|
| Accuracy | Source Train | 62.4 | 67.7 | **86.4** | 71.0 | 78.1 | 65.5 | 51.9 | 62.6 | 55.4 | 42.9 | 59.2 | 53.3 | 64.6 | 64.9 | 75.7 | 55.6 | 62.6 | 65.1 | 63.6 | 9.9 |
| | Source Val | 76.7 | 86.7 | 81.5 | **73.8** | 82.7 | **75.1** | **97.2** | 96.6 | **97.0** | 82.0 | 90.3 | 87.6 | 93.2 | 94.4 | 96.0 | 96.0 | 97.1 | **96.6** | **88.9** | 8.1 |
| BNM | Source Train | 49.6 | 49.8 | 54.8 | 33.2 | 51.4 | 14.6 | 21.1 | 35.3 | 16.2 | 12.2 | 28.6 | 14.9 | 28.0 | 44.8 | 53.6 | 4.2 | 18.3 | 19.1 | 30.5 | 16.0 |
| | Source Train + Target | 45.8 | 61.7 | 79.8 | 63.1 | 73.4 | 18.7 | 69.0 | 69.4 | 67.8 | 29.2 | 56.2 | 54.5 | 37.1 | 67.8 | 68.7 | 34.1 | 60.3 | 57.3 | 56.3 | 16.3 |
| | Source Val | 59.1 | 65.2 | 57.1 | 56.7 | 61.4 | 38.4 | 85.1 | 82.2 | 81.0 | 18.7 | 37.1 | 24.3 | 58.6 | 68.1 | 75.5 | 60.7 | 76.0 | 78.2 | 60.2 | 19.0 |
| | Source Val + Target | 50.8 | 64.3 | 73.8 | 65.6 | 72.9 | 34.7 | 81.5 | 79.0 | 78.2 | 27.4 | 50.6 | 45.1 | 42.1 | 69.0 | 71.9 | 44.0 | 69.0 | 70.6 | 60.6 | 16.1 |
| | Target | 46.0 | 61.9 | 79.5 | 63.6 | 73.3 | 18.9 | 69.0 | 69.3 | 67.8 | 29.4 | 56.9 | 55.4 | 36.9 | 67.9 | 68.8 | 33.9 | 60.5 | 57.2 | 56.5 | 16.3 |
| ClassAMI | Source + Target Features | 63.4 | 75.7 | **89.5** | 53.5 | **83.0** | 54.2 | 85.3 | 81.3 | 83.5 | 71.8 | 83.4 | 82.5 | 75.0 | 89.4 | 86.7 | 68.9 | 88.3 | 85.1 | 77.8 | 11.0 |
| | Source + Target Logits | 60.2 | 74.3 | **88.3** | 54.7 | 79.6 | 54.4 | 84.7 | 79.7 | 81.9 | 68.3 | 80.3 | 79.5 | 72.5 | 89.2 | 86.0 | 65.4 | 87.6 | 84.7 | 76.2 | 10.9 |
| | Target Features | 63.6 | 78.3 | 76.5 | 54.2 | **83.0** | 50.3 | 66.1 | 67.7 | 64.3 | 35.1 | 49.4 | 71.2 | 59.0 | 64.7 | 83.4 | 55.1 | 82.4 | 69.9 | 65.3 | 12.7 |
| | Target Logits | 60.4 | 75.9 | **83.2** | 55.0 | 80.6 | 50.3 | 61.4 | 75.5 | 78.1 | 43.2 | 65.3 | 72.0 | 55.7 | 62.1 | 83.1 | 55.4 | 79.6 | 79.9 | 67.6 | 12.2 |
| ClassSS | Source + Target Features | 24.8 | 25.7 | 59.0 | 52.5 | 59.7 | 21.3 | -6.7 | 2.7 | 16.6 | -37.9 | 8.0 | -1.8 | 2.6 | 16.4 | 39.6 | 6.7 | 10.7 | 26.8 | 18.2 | 23.7 |
| | Source + Target Logits | 29.1 | 33.3 | 48.2 | 55.4 | 59.5 | 24.9 | -2.3 | 3.8 | 11.6 | -33.9 | 9.3 | -1.9 | 3.1 | 18.7 | 39.3 | 6.8 | 6.9 | 25.3 | 18.7 | 22.6 |
| | Target Features | 25.6 | 38.6 | 48.6 | 52.9 | 53.0 | 33.4 | -19.1 | -26.3 | 12.6 | -57.6 | 0.2 | -3.8 | -14.8 | 1.0 | 30.6 | -11.4 | -12.2 | 16.5 | 9.3 | 29.7 |
| | Target Logits | 29.4 | 41.1 | 38.2 | 54.2 | 47.8 | 37.1 | -15.8 | -26.8 | 3.6 | -60.0 | -2.6 | -3.0 | -16.6 | 2.0 | 25.6 | -15.7 | -12.6 | 7.2 | 7.4 | 29.5 |
| DEV | Features | -30.0 | -28.2 | 5.1 | 33.1 | 28.4 | 34.8 | 38.6 | 42.4 | 11.1 | 4.9 | 21.3 | 19.6 | 22.3 | 35.0 | 35.7 | 9.3 | 43.0 | 38.5 | 20.3 | 21.2 |
| | Logits | -12.2 | -14.5 | 56.1 | 36.5 | 61.3 | 45.5 | 30.3 | 32.3 | 36.8 | -7.0 | 21.8 | 16.0 | -6.4 | 35.1 | 28.2 | -3.8 | 29.5 | 22.4 | 22.7 | 22.3 |
| | Preds | -46.9 | -52.6 | -59.8 | -48.4 | -62.4 | -52.5 | -37.0 | -43.1 | -35.5 | -50.4 | 35.9 | 37.6 | -63.6 | 13.5 | 55.0 | -63.6 | 20.0 | 47.3 | -22.5 | 42.1 |
| DEVN | Features, max normalization | 72.9 | 81.6 | 72.5 | 51.9 | 57.3 | 53.0 | 95.4 | 94.6 | 94.6 | 70.7 | 80.9 | 74.2 | 87.5 | 90.5 | 91.5 | 89.8 | 94.3 | 92.5 | 80.3 | 14.3 |
| | Features, standardization | 65.5 | 70.8 | 78.0 | 48.0 | 56.0 | 51.1 | 90.8 | 92.0 | 91.4 | 75.2 | 75.4 | 72.2 | 81.8 | 83.8 | 83.5 | 79.7 | 86.4 | 85.9 | 76.0 | 13.0 |
| | Logits, max normalization | 68.7 | 80.5 | 67.5 | 50.7 | 61.4 | 53.4 | 95.3 | 94.6 | 94.6 | 70.1 | 80.0 | 74.0 | 86.2 | 90.7 | 89.9 | 94.0 | 92.7 | 91.7 | 79.7 | 14.3 |
| | Logits, standardization | 57.0 | 65.1 | 74.2 | 46.2 | 69.3 | 47.7 | 91.6 | 92.3 | 91.4 | 68.3 | 70.9 | 67.3 | 77.1 | 80.4 | 82.7 | 78.8 | 85.5 | 88.2 | 74.1 | 13.7 |
| | Preds, max normalization | 69.3 | 81.0 | 61.3 | 50.3 | 56.0 | 53.1 | 95.2 | 94.0 | 93.9 | 64.7 | 63.3 | 61.4 | 85.8 | 88.4 | 88.0 | 89.6 | 92.8 | 90.7 | 76.6 | 15.7 |
| | Preds, standardization | 57.4 | 68.0 | 53.0 | 37.9 | 50.6 | 44.1 | 87.6 | 87.6 | 87.2 | 46.8 | 51.9 | 54.2 | 68.9 | 74.1 | 80.0 | 76.1 | 76.4 | 81.2 | 65.7 | 15.9 |
| Entropy | Source Train | 48.8 | 48.7 | 50.3 | 31.3 | 44.1 | 13.5 | 13.2 | 28.5 | 9.0 | 11.5 | 29.6 | 17.3 | 20.0 | 40.8 | 46.8 | -2.4 | 14.8 | 11.9 | 26.5 | 16.2 |
| | Source Train + Target | 41.8 | 59.6 | 70.3 | 51.0 | 67.8 | 42.1 | 60.9 | 65.8 | 58.9 | 32.7 | 52.3 | 49.6 | 39.6 | 65.0 | 66.4 | 38.2 | 59.9 | 60.1 | 54.6 | 11.2 |
| | Source Val | 52.4 | 65.7 | 57.6 | 46.1 | 57.9 | 32.7 | 70.6 | 71.1 | 70.7 | 17.7 | 38.8 | 26.5 | 53.3 | 66.8 | 71.6 | 55.4 | 69.6 | 74.6 | 55.5 | 16.6 |
| | Source Val + Target | 50.9 | 63.8 | 70.8 | 50.5 | 68.0 | 40.3 | 65.0 | 68.4 | 65.1 | 31.2 | 50.2 | 45.8 | 42.3 | 65.8 | 68.0 | 42.9 | 63.1 | 66.4 | 56.6 | 11.9 |
| | Target | 41.8 | 59.7 | 67.1 | 51.4 | 67.2 | 43.3 | 61.1 | 65.7 | 58.8 | 33.7 | 52.9 | 50.4 | 39.4 | 65.2 | 66.5 | 38.3 | 60.2 | 60.2 | 54.6 | 10.8 |
| SND | Features, $\tau = 0.05$ | -89.3 | -80.8 | -74.0 | -72.3 | -79.0 | -70.5 | -77.4 | -84.7 | -80.2 | -88.0 | -79.2 | -74.2 | -81.0 | -69.6 | -69.8 | -88.9 | -77.8 | -87.6 | -79.1 | 6.4 |
| | Features, $\tau = 0.1$ | -80.3 | -59.3 | -61.6 | -50.2 | -51.3 | -65.5 | -67.0 | -73.1 | -57.1 | -83.1 | -57.8 | -59.5 | -57.1 | -51.8 | -40.9 | -69.8 | -61.5 | -59.7 | -61.5 | 10.2 |
| | Features, $\tau = 0.5$ | -36.4 | -35.4 | -46.3 | -2.0 | -28.2 | -1.1 | -20.6 | -61.8 | -46.5 | -47.7 | -36.2 | -22.7 | -1.1 | 4.4 | -16.5 | -16.5 | -3.2 | -43.8 | -26.8 | 20.0 |
| | Logits, $\tau = 0.05$ | -93.3 | -81.4 | -84.2 | -70.2 | -91.1 | -69.4 | -94.2 | -94.1 | -95.7 | -94.4 | -94.0 | -94.1 | -96.5 | -95.0 | -95.4 | -96.5 | -96.5 | -97.1 | -90.7 | 8.4 |
| | Logits, $\tau = 0.1$ | -95.2 | -88.1 | -77.5 | -84.3 | -86.3 | -92.1 | -89.6 | -93.5 | -90.7 | -94.1 | -91.3 | -92.8 | -92.6 | -87.7 | -89.2 | -92.6 | -91.2 | -95.9 | -90.2 | **4.3** |
| | Logits, $\tau = 0.5$ | -66.9 | -55.3 | -58.2 | -17.6 | -56.9 | -33.0 | -73.6 | -77.8 | -63.4 | -84.8 | -55.6 | -64.5 | -72.5 | -66.7 | -52.1 | -77.4 | -67.2 | -64.0 | -61.5 | 15.6 |
| | Preds, $\tau = 0.05$ | -90.7 | -72.1 | -82.6 | -66.0 | -86.7 | -56.9 | -72.5 | -81.0 | -77.9 | -85.9 | -72.5 | -71.9 | -82.5 | -66.3 | -72.6 | -75.3 | -74.8 | -71.8 | -75.6 | 8.1 |
| | Preds, $\tau = 0.1$ | -97.0 | -92.1 | -84.6 | -95.0 | -91.1 | -96.8 | -93.0 | -95.2 | -96.9 | -94.6 | -98.0 | -96.9 | -96.6 | -96.0 | -98.3 | -97.7 | -97.1 | -98.7 | -95.3 | **3.3** |
| | Preds, $\tau = 0.5$ | -88.1 | -88.2 | -43.4 | -63.2 | -42.2 | -71.5 | -55.6 | -81.8 | -68.4 | -87.4 | -80.6 | -71.6 | -65.3 | -60.7 | -70.5 | -79.2 | -67.7 | -79.9 | -70.3 | 13.4 |

## G.2 Weighted Spearman Correlation for MNIST

Table 24: The weighted Spearman correlation for the MNIST $\rightarrow$ MNISTM task, using the checkpoints of all algorithms simultaneously.

|          |                                | MM    |
|----------|--------------------------------|-------|
|          |                                | MM    |
| Accuracy | Source Train                   | 9.4   |
|          | Source Val                     | -7.1  |
| BNM      | Source Train                   | 3.5   |
|          | Source Train + Target          | 55.7  |
|          | Source Val                     | 2.5   |
|          | Source Val + Target            | 52.8  |
|          | Target                         | **56.1** |
| ClassAMI | Source + Target Features       | -29.4 |
|          | Source + Target Logits         | -25.8 |
|          | Target Features                | 2.2   |
|          | Target Logits                  | -21.5 |
| ClassSS  | Source + Target Features       | -68.1 |
|          | Source + Target Logits         | -66.1 |
|          | Target Features                | -60.5 |
|          | Target Logits                  | -53.8 |
| DEV      | Features                       | -11.0 |
|          | Logits                         | -5.1  |
|          | Preds                          | -24.0 |
| DEVN     | Features, max normalization    | -30.6 |
|          | Features, standardization      | -14.0 |
|          | Logits, max normalization      | -28.9 |
|          | Logits, standardization        | -14.2 |
|          | Preds, max normalization       | -45.2 |
|          | Preds, standardization         | -38.7 |
| Entropy  | Source Train                   | -4.1  |
|          | Source Train + Target          | -35.6 |
|          | Source Val                     | -5.8  |
|          | Source Val + Target            | -36.6 |
|          | Target                         | -42.2 |
| SND      | Features, $\tau = 0.05$        | -63.7 |
|          | Features, $\tau = 0.1$         | -62.9 |
|          | Features, $\tau = 0.5$         | -69.3 |
|          | Logits, $\tau = 0.05$          | -67.4 |
|          | Logits, $\tau = 0.1$           | -67.7 |
|          | Logits, $\tau = 0.5$           | -72.8 |
|          | Preds, $\tau = 0.05$           | -67.6 |
|          | Preds, $\tau = 0.1$            | -66.3 |
|          | Preds, $\tau = 0.5$            | -68.4 |

Table 25: The weighted Spearman correlation of each validator/algorithm pair, for the MNIST → MNISTM task.

| | | ATDOC | BNM | BSP | CDAN | DANN | GVB | IM | MCC | MCD | MMD | Mean | Std |
|---|---|---|---|---|---|---|---|---|---|---|---|---|---|
| Accuracy | Source Train | -7.5 | 34.8 | -1.8 | -12.4 | 6.2 | 17.8 | 21.2 | 16.5 | -8.0 | 27.6 | 9.4 | 15.6 |
| | Source Val | -8.1 | 34.2 | -6.4 | -4.4 | -4.6 | 11.8 | 24.3 | 2.4 | -24.1 | 30.4 | 5.5 | 18.0 |
| BNM | Source Train | -9.5 | 35.9 | 38.8 | -4.7 | 14.8 | 16.9 | 44.7 | 8.1 | -5.5 | 28.2 | 16.8 | 18.7 |
| | Source Train + Target | 81.2 | 56.4 | 72.8 | 36.2 | 61.7 | 30.6 | 48.2 | 50.7 | 92.5 | 23.9 | 55.4 | 21.0 |
| | Source Val | 12.6 | 50.7 | 52.2 | -0.9 | 11.3 | 14.8 | 65.0 | -27.7 | 7.4 | 25.7 | 21.1 | 26.6 |
| | Source Val + Target | 81.1 | 56.7 | 74.8 | 35.1 | 61.1 | 26.2 | 50.8 | 51.8 | 92.3 | 22.8 | 55.3 | 21.9 |
| | Target | 80.3 | 56.3 | 69.4 | 40.2 | 62.5 | 35.5 | 41.9 | 48.9 | 92.3 | 25.3 | 55.3 | 20.0 |
| ClassAMI | Source + Target Features | -21.6 | 33.5 | -78.8 | 43.6 | 58.3 | 15.8 | 56.3 | -32.9 | -58.2 | 7.7 | 2.4 | 45.8 |
| | Source + Target Logits | -0.8 | 37.4 | -78.0 | 60.3 | 57.7 | 19.6 | 58.3 | -42.6 | -56.8 | -10.7 | 4.4 | 48.2 |
| | Target Features | 45.7 | -19.2 | -69.6 | -43.4 | 21.5 | -52.9 | -34.8 | -16.6 | 4.7 | 23.7 | -14.1 | 35.4 |
| | Target Logits | 25.1 | 23.1 | -72.4 | -53.6 | -67.5 | -56.7 | 3.0 | -30.4 | -49.1 | 7.9 | -27.1 | 36.2 |
| ClassSS | Source + Target Features | -1.9 | -67.1 | -76.1 | -59.2 | -38.5 | -39.5 | -59.9 | -56.8 | -36.5 | 2.5 | -43.3 | 25.0 |
| | Source + Target Logits | -8.7 | -73.7 | -77.1 | -41.1 | -47.6 | -31.5 | -62.4 | -66.0 | -28.6 | 33.3 | -40.3 | 32.1 |
| | Target Features | -11.6 | -55.9 | -76.7 | -59.3 | -35.8 | -43.3 | -56.7 | -60.0 | -46.0 | -10.6 | -45.6 | 20.2 |
| | Target Logits | -20.4 | -67.1 | -80.2 | -41.3 | -36.8 | -40.7 | -60.5 | -61.4 | -53.4 | 27.7 | -43.4 | 28.8 |
| DEV | Features | 22.8 | -52.6 | -10.5 | 9.5 | -12.9 | -13.1 | -38.8 | -7.7 | -47.9 | 9.1 | -14.2 | 24.0 |
| | Logits | 9.5 | -18.3 | -16.1 | 1.6 | -4.4 | 8.1 | 20.2 | -3.4 | -73.2 | 8.1 | -6.8 | 24.8 |
| | Preds | 18.2 | -8.2 | 22.9 | 11.7 | 22.5 | 15.3 | -9.6 | -20.5 | -45.6 | 15.7 | 2.2 | 21.4 |
| DEVN | Features, max normalization | -0.9 | -13.3 | 5.4 | -5.0 | -10.1 | -14.7 | -26.3 | -13.1 | -11.8 | 9.3 | -8.1 | 10.0 |
| | Features, standardization | 32.3 | -13.9 | 27.3 | 11.1 | -7.6 | -16.9 | -24.5 | -12.5 | -23.4 | 18.4 | -1.0 | 20.2 |
| | Logits, max normalization | 6.2 | -16.6 | 13.7 | -5.9 | -7.3 | -16.9 | -28.7 | -12.6 | -12.9 | 7.7 | -7.3 | 12.4 |
| | Logits, standardization | 13.4 | -19.1 | 2.5 | 7.8 | 9.7 | -11.0 | -20.9 | 3.0 | -42.1 | 6.6 | -5.0 | 16.8 |
| | Preds, max normalization | -17.4 | -9.4 | -1.2 | -3.1 | -13.5 | -7.1 | -26.4 | -13.8 | -24.0 | 11.8 | -10.4 | 10.8 |
| | Preds, standardization | -29.3 | -51.0 | -6.5 | 22.0 | -2.0 | -4.4 | -36.5 | -9.1 | -45.2 | 5.8 | -15.6 | 22.5 |
| Entropy | Source Train | -5.8 | 35.9 | 45.8 | -30.1 | -22.9 | 16.3 | 45.8 | 7.5 | -5.3 | 29.1 | 11.6 | 26.1 |
| | Source Train + Target | -47.7 | -18.1 | 65.4 | -4.2 | 10.4 | 12.6 | -6.6 | 21.6 | -66.6 | 16.1 | -1.7 | 35.1 |
| | Source Val | 14.1 | 52.8 | 57.9 | -29.3 | -24.0 | 14.7 | 64.6 | -32.3 | 6.8 | 27.5 | 15.3 | 34.2 |
| | Source Val + Target | -47.6 | -19.5 | 64.9 | -3.4 | 11.2 | 13.2 | -7.8 | 16.5 | -69.8 | 15.9 | -2.6 | 35.4 |
| | Target | -46.1 | -46.3 | 57.0 | -0.5 | 4.2 | 20.1 | -47.0 | 14.4 | -84.2 | 16.3 | -11.2 | 40.6 |
| SND | Features, $\tau = 0.05$ | -34.1 | -41.9 | -73.8 | -78.3 | -62.9 | -80.5 | -42.6 | -66.5 | -22.9 | 39.0 | -46.4 | 34.1 |
| | Features, $\tau = 0.1$ | -36.5 | -40.7 | -73.8 | -82.3 | -60.6 | -80.0 | -43.5 | -62.0 | -25.1 | 39.8 | -46.5 | 34.1 |
| | Features, $\tau = 0.5$ | -27.4 | -68.7 | -76.4 | -82.4 | -65.1 | -80.3 | -69.7 | -58.3 | -37.2 | 37.9 | -52.8 | 34.7 |
| | Logits, $\tau = 0.05$ | -12.8 | -56.1 | -75.1 | -75.2 | -69.2 | -84.6 | -51.9 | -90.4 | -13.7 | -54.9 | -58.4 | 25.6 |
| | Logits, $\tau = 0.1$ | -16.8 | -51.8 | -75.4 | -84.5 | -69.1 | -84.4 | -49.0 | -88.7 | -15.7 | -60.1 | -59.5 | 25.2 |
| | Logits, $\tau = 0.5$ | -26.6 | -52.8 | -76.3 | -81.0 | -71.0 | -78.4 | -60.5 | -80.1 | -25.7 | -48.2 | -60.1 | 20.1 |
| | Preds, $\tau = 0.05$ | -5.0 | -61.5 | -75.9 | -80.2 | -75.5 | -87.6 | -52.7 | -85.6 | -36.7 | -43.2 | -60.4 | 24.9 |
| | Preds, $\tau = 0.1$ | -7.6 | -60.4 | -77.1 | -85.0 | -76.3 | -85.6 | -52.0 | -85.9 | -37.3 | -72.6 | -64.0 | 24.2 |
| | Preds, $\tau = 0.5$ | -16.7 | -56.9 | -80.6 | -82.6 | -76.4 | -83.6 | -51.7 | -85.5 | -43.2 | -40.1 | -61.7 | 22.4 |

# H  Summary of UDA Algorithms

The goal of unsupervised domain adaptation (UDA) is to adapt a model trained on labeled source data, for use on unlabeled target data. Applications of UDA include:

- semantic segmentation (Toldo et al., 2020)
- object detection (Oza et al., 2021)
- natural language processing (Ramponi & Plank, 2020)

There are also other types of domain adaptation, including:

- semi-supervised (Saito et al., 2019)
- multi-source (Peng et al., 2019)
- partial (Panareda Busto & Gall, 2017; Saito et al., 2018b; Cao et al., 2018)
- universal (You et al., 2019a)
- source-free (Liang et al., 2020)

In this paper, we focus on UDA for image classification, because it is well-studied and often used as a foundation for other domain adaptation subfields.

Here we provide a summary of UDA algorithms by category:

- **Adversarial** methods use a GAN where the generator outputs feature vectors. The discriminator's goal is to correctly classify features as coming from the source or target domain, while the generator tries to minimize the discriminator's accuracy. Examples:
  - DANN (Ganin et al., 2016)
  - Domain Confusion (Tzeng et al., 2015)
  - ADDA (Tzeng et al., 2017)
  - CDAN (Long et al., 2017a)
  - VADA (Shu et al., 2018)
- **Feature distance losses** encourage source and target features to have similar distributions. Examples:
  - MMD (Long et al., 2015)
  - CORAL (Sun et al., 2016)
  - JMMD (Long et al., 2017b)
- **Maximum classifier discrepancy** methods use a generator and multiple classifiers in an adversarial setup. The classifiers' goal is to maximize the difference between their prediction vectors (i.e. after softmax) for the target domain data, while the generator's goal is to minimize this discrepancy. Examples:
  - MCD (Saito et al., 2018a)
  - SWD (Lee et al., 2019)
  - STAR (Lu et al., 2020)
- **Information maximization** methods use the entropy or mutual information of prediction vectors. Examples:
  - ITL (Shi & Sha, 2012)
  - MCC (Jin et al., 2020)
  - SENTRY (Prabhu et al., 2021)
- **SVD losses** apply singular value decomposition to the source and/or target features. Examples:
  - BSP (Chen et al., 2019)
  - BNM (Cui et al., 2020a)
- **Image generation** methods use a decoder model to generate source/target -like images from feature vectors, usually as part of of an adversarial method. Examples:
  - DRCN (Ghifary et al., 2016)
  - GTA (Sankaranarayanan et al., 2018)

- **Pseudo-labeling** methods generate labels for the unlabeled target-domain data, to transform the problem from unsupervised to supervised. This is also known as self-supervised learning. Examples:
  - ATDA (Saito et al., 2017)
  - ATDOC (Liang et al., 2021)
- **Mixup augmentations** create training data and features that are a blend between source and target domains. Examples:
  - DM-ADA (Xu et al., 2020)
  - DMRL (Wu et al., 2020)
- Other notable methods that are more difficult to categorize include:
  - RTN (Long et al., 2016)
  - AFN (Xu et al., 2019)
  - DSBN (Chang et al., 2019)
  - SymNets (Zhang et al., 2019)
  - GVB (Cui et al., 2020b)

