# OpenReview forum: "Evaluating the Evaluators: Which UDA validation methods are most effective? Can they be improved?"
_TMLR — Rejected by TMLR_

### Review · Reviewer_5fUT · 2022-11-26

**Summary Of Contributions:**

This paper focuses on methods for evaluating the out of distribution (OOD) performance of machine learning models. The authors compare eight different methods, and rank their usefulness for model selection.

The main contribution of this paper is the experimental setup, which allows comparing multiple methods on many different datasets. The results, while not too insightful, are useful for researchers working with the methods compared here.


**Audience:**

Yes

**Broader Impact Concerns:**

I have no concerns about the ethical implications of this work that would require adding a Broader Impact Statement.



**Claims And Evidence:**

Yes

**Requested Changes:**


As mentioned in the weaknesses section, my main concerns are the lack of theoretical motivation and the missing literature. Therefore, I’m adding here a list of some potentially relevant papers that I recommend the authors address and compare in their analysis (this is a very small but representative sample):

1. Invariant risk minimization

2. Invariant causal prediction for nonlinear models

3. Distributionally robust neural networks for group shifts: On the importance of regularization for worst-case generalization

4. Wilds: A benchmark of in-the-wild distribution shifts

5. Are VQA systems rad? measuring robustness to augmented data with focused interventions

6. Domain generalization: A survey

7. Just train twice: Improving group robustness without training group information

8. Measuring robustness to natural distribution shifts in image classification

9. On calibration and out-of-domain generalization

In addressing all of this body of work, I believe that the analysis and evaluation results would be much more useful to the TMLR audience. If the authors seriously engage with this literature it will strengthen this work substantially.


**Strengths And Weaknesses:**

Strengths:

The paper discusses an important and timely problem - which model selection methods are able to choose models that will perform well out of distribution. The paper is very clearly written, and provides a valuable comparison between some potential methods. The experimental setup is well-motivated, and allows comparing the different approaches.

Weaknesses:

While I whole-heartedly share with the authors the belief that model selection is a crucially important problem for OOD generalization, I find the contributions of this paper very limited.

My two main concerns are: (1) This paper is oblivious to almost all of the work on OOD generalization from the last 5-7 years. This causes the analysis to be extremely partial and irrelevant. (2) The lack of any theoretical motivation in choosing the methods or in analyzing the results, leads to very limited insight from the provided analysis.

To be more concrete, here are my main concerns:

1. The paper feels quite shallow in its analysis, and is lacking any substantial theoretical motivation. This is crucial because it is otherwise unclear what is the insight that we are gaining from such analysis. In trying to use this paper to gather intuition about UDA and when it should work or fail, I find this paper only partially useful. One question that remains unanswered is what more do we understand about UDA following this paper?

2. One potential avenue where UDA and its evaluation methods could be motivated in a principled way is from a causal perspective. Building on well-known papers such as “Invariant Causal Predictions” (ICP) and “Invariant Risk Minimization” (IRM), these approaches provide methods and evaluation metrics for a UDA setting. The authors do not address this literature at all, leading to strong and potentially misleading claims such as: “To date, much UDA research has been focused on the algorithms that train models” (Page 1). Many papers in the invariant learning literature deal with evaluation metrics for out-of-distribution (OOD) settings. In natural language processing and computer vision, there are many papers that are mostly focused on evaluation for OOD (see potentially relevant references at the bottom).

This is not the only literature they completely fail to mention. There are entire branches of work that are focused on the exact problem discussed in this paper. Apart from the invariant learning literature, there is a lot of work on distributionally robust optimization for OOD generalization. There is also a lot of work on UDA in NLP and in computer vision that is completely missing here. I’m attaching many examples at the requested changes sections so the authors could read up on all of this illuminating work.

3. There are now multiple ambitious projects that include datasets for UDA, but none of them are mentioned here. For example, in WILDS, a very well-known project, there are more than 10 different datasets that compare many different models and evaluation methods (i.e. model selection strategies) in computer vision, NLP and more. Here, the authors choose to focus on a much more limited set of datasets and methods.

Minor Issues:

4. There are many redundant equations and explanations that could be discarded given that the target audience understands basic concepts in machine learning. For example, Equations 1-3 are well known to potential readers and are not needed in my opinion.

5. Why is DEV with normalization (DEVN) explained using coding syntax and not with math/words? In my opinion, this feels odd given the other explanations in Section 1.

---

> ### Author Response · Authors · 2023-01-08
> **Response to 5fUT: Part 1**
>
> Thank you for your review. Here is our response:
>
> Summary:
> - The papers you have suggested are irrelevant because they are focused on OOD-generalization, which is an entirely different field from UDA.
> - Our paper presents 3 new validation methods that outperform existing methods. This alone far exceeds the standard for publication in leading machine-learning research journals.
> - Our comparison and analysis of UDA validation methods is the most comprehensive to date. It will serve as an invaluable resource for researchers.
>
> Details:
>
> We are pleased to learn that you think our “experimental setup is well-motivated, and allows comparing the different approaches.”
>
> However, there seems to be a misunderstanding about the subject of our paper. You believe our field of research is OOD-generalization (OODG), but that is not true. Instead, our actual field of research is UDA. This misunderstanding has led you to suggest papers that are unrelated to our work. They are irrelevant because all of them focus on OODG, whereas our topic is UDA.
>
> While some aspects of OODG and UDA might seem tangentially related, they are actually quite different in terms of:
> - Train/test data:
>   - OODG uses labeled training data, whereas UDA uses both labeled and unlabeled training data.
>   - OODG uses test-time data that has a never-before-seen distribution. In contrast, UDA uses test-time data that has the same distribution as the unlabeled training data.
> - Validation:
>   - OODG can directly compute accuracy on labeled validation data. In contrast, UDA can only estimate accuracy, because the validation data is unlabeled.
>
> Given the above differences, OODG has no bearing on our research, and we were right to disregard it. Moreover, disregarding OODG did not make our analysis “extremely partial and irrelevant” as you claim. Instead, disregarding OODG helped focus our analysis on what was truly relevant. In fact, UDA is a distinct topic that can be studied independently of OODG, much as dogs can be studied independently of cats.
>
> Furthermore, you claim that there are “entire branches of work that are focused on the exact problem discussed in this paper”, yet none of the papers you suggested are about
> - UDA validation methods,
> - model selection using unlabeled data, or
> - accuracy estimation on unlabeled training data.
>
> Thus none of those papers are relevant.
>
> In addition, you mention that “there is a lot of work on distributionally robust optimization for OOD generalization”. This work is irrelevant to our paper because it does not relate to UDA, nor does it relate to model selection using unlabeled data, nor does it relate to accuracy estimation on unlabeled training data.
>
> Another point you make is that “there is also a lot of work on UDA in NLP and in computer vision that is completely missing here”. Yes indeed, innumerable UDA papers exist, but it would be impractical for us to cite them all, and doing so would overwhelm our readers. So we cited only those UDA papers that are most relevant. After all, our paper is not a survey.
>
> Regarding datasets in particular, you state that “there are now multiple ambitious projects that include datasets for UDA, but none of them are mentioned here”. Granted, but it is impossible to do a thorough evaluation on all possible datasets, due to computational constraints.
>
> Specifically, our experiments with our chosen datasets required many thousands of GPU-hours, and terabytes of disk space. Thousands of additional GPU-hours were spent running experiments on an even larger dataset, DomainNet126. We are planning to update our paper with the results from these experiments.
>
> Furthermore, the four datasets we chose have been widely used for UDA research. For some examples of papers that use these datasets, see the appendix to this response.
>
> Regarding the WILDS benchmark, it is intended for OODG, not for UDA. In the related paper, “Extending the WILDS Benchmark for Unsupervised Adaptation”, they use accuracy on labeled data as the model selection criterion. This is just one of the validation methods we evaluated (we call it “source domain accuracy”). Furthermore, in section 7 of the paper, the authors say: “Improved methods for hyperparameter tuning could significantly improve OOD performance. Such methods might make use of the unlabeled target data”. We agree, which is why our paper fills in this gap by benchmarking numerous methods that make use of *unlabeled* target data.
>
> You claim that our paper offers “limited insight” because it is based on empirical results rather than theory. However, empirical benchmark papers actually serve a useful purpose. That is why many such papers have been accepted to top-tier conferences and journals (including TMLR). These empirical, non-theoretical papers have also been cited widely – because they are useful. For some examples of recently-published empirical papers, see the appendix of this response.

---

> > ### Author Response · Authors · 2023-01-08
> > **Response to 5fUT: Part 2**
> >
> > In fact, our paper offers considerable value, including the following:
> >
> > - Our comparison and analysis of UDA validation methods is the most comprehensive to date. As such, it will serve as an invaluable resource for researchers who wish to improve the state-of-the-art.
> > - Our paper reveals that current UDA validation methods are not yet reliable enough to be used for training models.
> > - Our paper proposes three new validation methods that outperform existing methods. This contribution exceeds the standard in our field, which is to publish a single new validation method, without comparing it to a diverse set of existing methods on a very large dataset of checkpoints. Thus, the three validation methods that we propose are, on their own, a significant contribution that is well worthy of publication.
> >
> > Thank you for reviewing our paper.

---

> > > ### Author Response · Authors · 2023-01-08
> > > **Response to 5fUT: Appendix**
> > >
> > > Appendix:
> > >
> > > Recent UDA papers that use the Office-31, OfficeHome, and/or DomainNet126 datasets:
> > >
> > > - Reusing the Task-specific Classifier as a Discriminator: Discriminator-free Adversarial Domain Adaptation (CVPR 2022)
> > > - A Closer Look at Smoothness in Domain Adversarial Training (ICML 2022)
> > > - Adversarial Unsupervised Domain Adaptation with Conditional and Label Shift: Infer, Align and Iterate (ICCV 2021)
> > > - Gradient Distribution Alignment Certificates Better Adversarial Domain Adaptation (ICCV 2021)
> > > - Re-energizing Domain Discriminator with Sample Relabeling for Adversarial Domain Adaptation (ICCV 2021)
> > > - MetaAlign: Coordinating Domain Alignment and Classification for Unsupervised Domain Adaptation (CVPR 2021)
> > >
> > >
> > > Non-theoretical benchmark papers accepted to TMLR:
> > >
> > > - Benchmarking and Analyzing Unsupervised Network Representation Learning and the Illusion of Progress
> > > - Failure Detection in Medical Image Classification: A Reality Check and Benchmarking Testbed
> > > - Benchmarking Progress to Infant-Level Physical Reasoning in AI
> > >
> > >
> > > Non-theoretical benchmark papers accepted to other top-tier conferences:
> > >
> > > - OpenOOD: Benchmarking Generalized Out-of-Distribution Detection (NeurIPS 2022)
> > > - Beyond Supervised vs. Unsupervised: Representative Benchmarking and Analysis of Image Representation Learning (CVPR 2022)
> > > - Descending through a Crowded Valley - Benchmarking Deep Learning Optimizers (ICML 2021)
> > > - Measuring Robustness to Natural Distribution Shifts in Image Classification (NeurIPS 2020)
> > > - Benchmarking Neural Network Robustness to Common Corruptions and Perturbations (ICLR 2019)

---

### Review · Reviewer_NLeA · 2022-11-28

**Summary Of Contributions:**

This paper conducts empirical analysis to evaluate the unsupervised validation methods in the problem of unsupervised domain adaptation (UDA). Specifically, the authors study 8 validators paired with 10 UDA algorithms, the authors also propose 3 new validators that are modified variants of existing ones. The experiments are performed on 19 transfer tasks where each was run with 100 different hyperparameters. The authors finally draw several conclusions that could be helpful for practitioners to select the appropriate validator.

**Audience:**

Yes

**Claims And Evidence:**

Yes

**Requested Changes:**

According to the weaknesses above, I would appreciate it if the authors can:

1. improve the writing of the paper, particularly alter the layout of the paper to make it more compact
2. modify the findings drawn from the results and present them in a better way

**Strengths And Weaknesses:**

### Strengths

1. The authors conduct comprehensive empirical analysis on the validators in UDA. It is nice to see such a comparison including multiple UDA algorithms, validators, and training runs.
2. I think that related researchers could draw helpful conclusions from the results in this paper for them to pick the appropriate validators.

### Weaknesses

1. I feel this paper represents a very focused contribution to a very focused subproblem – the unsupervised validator selection in UDA. Given that the contributions are mostly from empirical analysis of this specific subproblem, the impact of this work is limited.
2. The writing of this paper is more like a technical report – I think the writing and layout could be improved, and the paper could be presented in a more compact format, for example, most figures are unnecessarily large.
3. I feel one of the main findings “BNM validator combined with the ATDOC algorithm comprises the best algorithm/validator pair” is a weird framing and not helpful since here “best” is in terms of weighted spearman correlation. However, in Table 5(b) such a combination actually underperforms others in many tasks in terms of performance. That being said, I think performance is a more instructive metric to judge algorithm/validation pairs, because it is useless if an algorithm/validator pair has high weighted spearman correlation but bad performance.

---

> ### Author Response · Authors · 2023-01-08
> **Response to NLeA**
>
> Thank you for your review.
>
> ---
>
> You have expressed concern that our paper is focused on a “specific subproblem”, so the impact of our work is limited. In other words, you implied that validator selection in UDA is an insignificant niche.
>
> However, UDA is actually a very active research field. There have been 158 papers submitted to Arxiv in 2022 with the phrase “Unsupervised Domain Adaptation” in the title.
>
> In fact, UDA has many applications, such as object detection for self-driving cars, segmentation in medical images, and natural language processing.
>
> Furthermore, anyone using a UDA algorithm in a real-world application will need to use a validation method. Moreover, the choice of validation method can have a much bigger effect on accuracy than the choice of algorithm.
>
> Thus, validation methods are not a trivial or niche subject matter. They are actually a crucial component of UDA, and UDA is a very active field of research with numerous applications.
>
> As for the empirical nature of our paper, this will not limit its impact. On the contrary, our paper makes the following significant contributions:
>
> - Our comparison and analysis of UDA validation methods is the most comprehensive to date. As such, it will serve as an invaluable resource for researchers who wish to improve the state-of-the-art.
> - Our paper reveals that current UDA validation methods are not yet reliable enough to be used for training models.
> - Our paper proposes three new validation methods that outperform existing methods. This contribution exceeds the standard in our field, which is to publish a single new validation method, without comparing it to a diverse set of existing methods on a very large dataset of checkpoints. Thus, the three validation methods that we propose are, on their own, a significant contribution that is well worthy of publication.
>
> Therefore, our benchmark of UDA validation methods is actually a high-impact contribution about a crucial subproblem (validation methods) in a widely-studied field (UDA).
>
> ---
>
> You have also suggested that most of the figures in our paper are unnecessarily large.
>
> However, there is a good reason why the figures are large. That is, the scatter plots have thousands of points, and we made them large so that all the points are visible. Likewise, the bar plots have many bars, and we made them large so that the label for each bar is legible.
>
> ---
>
> In addition, you have suggested that performance is a more instructive metric by which to judge algorithm/validation pairs, because even if an algorithm/validator pair has a high weighted spearman correlation, it is nonetheless useless if it performs poorly.
>
> This is a good point. We investigated this and found that:
> - When comparing algorithm/validator pairs, the best metric is accuracy (“performance”) as you have suggested.
> - When comparing validators independently of algorithms, or with a single algorithm, the best metric is weighted Spearman correlation.
>
> Please see the following anonymized document for details:
>
> https://docs.google.com/document/d/1FkrdW1fBTc9QI0RA93HgKxADTuMU3uRN1oN6cyMv4EI/edit?usp=sharing
>
> Since submitting our original paper, we have expanded the scope of our study to include the DomainNet126 dataset. This additional experiment has revealed that the algorithm/validator pair with the best average accuracy is MCC / ClassAMI. We are planning to revise our paper accordingly.

---

### Review · Reviewer_DiNn · 2023-01-06

**Summary Of Contributions:**

This paper provides empirical studies to investigate the effectiveness of various UDA validation methods. Such a largescale benchmark study focusing on model selection is the first work of its kind in the UDA community, which brings many new knowledge to the file:
1. Many UDA validators are not as good as we expected and there is still a huge rome for the improvement.

2. Existing UDA validators usually perform various on different UDA tasks and UDA agorithms. Therefore, for specific UDA setting, how to select a suitable validator faces challenges.

3. Even the best existing validators are selected by each algorithm, there still have a large gap when comparing to the oracle validator.

Besides, the paper also improves two existing validators and provides a new validator into the analysis.

**Audience:**

Yes

**Claims And Evidence:**

Yes

**Requested Changes:**

Note that the weaknesses listed in above are not neccessary to address, but I think the paper could be improved by considering the following suggestions:
1. In Figure 2, similar figure on MNIST could be provided. Since the results of this paper did not provide any conclusions related to the MNIST dataset (except Table 5), the readers may be curious about what is the purpose by doing experiments on MNIST. Therefore, I think the paper could provide addtional knowledge by doing such analysis.

2. The detail descriptions on the Weight Spearman Correlation in Page 19 (Equations  20-23) are unclear. What is the reason by such design? Why not just adding the weight (Equation12) to the Spearman correlation? I don't know the purpose of Equations  20-23.

3. In Table 5, I think it would be better to provided the mean accuracy across tasks for Office31 and OfficeHome.

**Strengths And Weaknesses:**

# Strengths:
1. Many UDA researches have been focused on the algorithms and usually reported their results based on oracle validator, which is really an issue that must be addressed. I think this empirical study is very insightful to the UDA community by showing that UDA validation methods indeed paly a much important role to the UDA problem itself.

2. The design of the empirical study is good and the reported results are clear presented and sufficient.

3. All conclusions in the paper are supported by the results.

# Weaknesses
1. There are still three validators that are not analysed in this paper. I understand that these three validators are hard to analyse due to the computing issue, but the reverse validation method looks like the best one in my mind.

2. The experiments on Office31 and OfficeHome show similar results but are different on MNIST dataset, e.g., the source validtion method is good on Office31 and OfficeHome but seems very bad on MNIST, as shown in Tables 24 and 25. Such phenomenon may let readers curious about wheather the validators are sentative to the datasets, but the authors did not provide such anaysis to the paper.

---

> ### Author Response · Authors · 2023-01-08
> **Response to DiNn**
>
> Thank you for your review
>
> You suggested showing more of the MNIST results in the main part of the paper. This is a good point. We highlighted Office31 and OfficeHome because they are more similar to what most ML researchers will work with. However, we agree that readers would be interested in the performance on MNIST, since it is such an outlier for many of the validators. We will update the paper to show MNIST along with the other two datasets.
>
> ---
>
> You asked why the weighted Spearman correlation is designed the way it is. The weighted Spearman correlation is based on the weighted Pearson correlation (equation 17-19 in the paper), which uses weighted means, weighted variances, and weighted covariances.
>
> The purpose of equations 20-23 is to compute the weighted rank of a sample. The weighted rank of sample j consists of two components:
> - The sum of weights of all samples with a lower rank than sample j (equation 21)
> - The mean of weights of all samples with the same rank as sample j, scaled by the number of tied samples (equation 22). This equation is constructed to make the weighted ranks start at 1 and be the average of the ranks in the case of ties. For details of the derivation, refer to the original paper “Weighted and Unweighted Correlation Methods for LargeScale Educational Assessment: wCorr Formulas”.
>
> ---
>
> You suggested showing the mean accuracy across tasks in table 5. This is a good idea. We will update table 5 to include a mean accuracy column.

---

### Decision · Action_Editors · 2023-02-10

**Recommendation:** Reject

**Comment:**

The paper focuses on an important issue in UDA research about the efficacy of validators for selection of UDA models. The paper
benchmarks 8 validation methods and also proposes 3 new variants which show some advantages over existing ones.

Three expert reviewers give comprehensive reviews. They can find merits of the paper, such as thorough empirical studies and some
useful suggestions for selection of UDA validators. However, they are also concerned with the impact of the paper, its lack of significant
technical contributions, and writing of the paper.

In the discussion phase, authors have responded to some of the issues raised by reviewers. For example, the issue raised by Reviewer NLeA about whether weighted spearman correlation is an appropriate metric for algorithm/validator pair, the authors responded with elaborate arguments, unfortunately, the reviewer is not fully convinced. Reviewer 5fUT evaluates the paper from a more general perspective of OOD generalization, which is closely related to UDA problem (with a subtle difference of setting).  The authors have argued that the paper should be judged within the UDA domain, and thus the comments from 5fUT are less relevant. While AE agrees with the authors to some extent, AE also thinks that inclusion of important insights, including some theoretical ones, from OOD literature can significantly improve the quality of the paper.

Based on the review comments and the discussion, it is unfortunate that the paper cannot be accepted at the current stage.

**Audience:**

The studied topic is an important issue in the UDA research field. More individuals would be interested in the paper if more consistent and insightful suggestions can be drawn from the empirical studies.

**Claims And Evidence:**

The claims made in the paper are not well supported by the empirical results. It would be better if more theoretical insights can be included, at least analyzing the empirical results from a more theoretical perspective. In addition, claims may be better supported if consistent conclusions can be drawn for different UDA algorithms.